# Safe Pontryagin Differentiable Programming

**Wanxin Jin**
University of Pennsylvania
wanxinjin@gmail.com

**Shaoshuai Mou**
Purdue University
mous@purdue.edu

**George J. Pappas**
University of Pennsylvania
pappasg@seas.upenn.edu

## Abstract

We propose a Safe Pontryagin Differentiable Programming (Safe PDP) methodology, which establishes a theoretical and algorithmic framework to solve a broad class of safety-critical learning and control tasks—problems that require the guarantee of safety constraint satisfaction at any stage of the learning and control progress. In the spirit of interior-point methods, Safe PDP handles different types of system constraints on states and inputs by incorporating them into the cost or loss through barrier functions. We prove three fundamentals of the proposed Safe PDP: first, both the solution and its gradient in the backward pass can be approximated by solving their more efficient unconstrained counterparts; second, the approximation for both the solution and its gradient can be controlled for arbitrary accuracy by a barrier parameter; and third, importantly, all intermediate results throughout the approximation and optimization strictly respect the constraints, thus guaranteeing safety throughout the entire learning and control process. We demonstrate the capabilities of Safe PDP in solving various safety-critical tasks, including safe policy optimization, safe motion planning, and learning MPCs from demonstrations, on different challenging systems such as 6-DoF maneuvering quadrotor and 6-DoF rocket powered landing.

## 1 Introduction

Safety is usually a priority in the deployment of a learning or control algorithm to real-world systems. For a physical system (agent), safety is normally given in various constraints on system states and inputs, which must not be violated by the algorithm at *any stage* of the learning and control process, otherwise will cause irrevocable or unacceptable failure/damage. Those systems are referred to as *safety-critical*. The constraints in a safety-critical system can include the immediate ones, which are directly imposed on the system state and input at certain or all-time instances, and the long-term ones, which are defined on the trajectory of system states and inputs over a long period.

Compared to the abundant results that focus on system optimality [1–3], systematic and principled treatments for safety-critical learning and control problems seem largely insufficient, particularly in the following gaps (detailed in Section 1.1). First, existing safety strategies are either too conservative, which may restrict the task performance, or violation-tolerable, which only pursues the near-constraint guarantee and thus are not strictly constraint-respecting. Second, a systematic safety paradigm capable of handling different types of constraints, including system state and input (or mixed), immediate, or/and long-term constraints, is still lacked. Third, some existing safety strategies suffer from huge computational- and data- complexity, difficult to be integrated into any differentiable programming frameworks to solve large-scale learning and continuous control tasks.

To address the above research gaps, this paper aims to develop a *safe differentiable programming* framework with the following key capabilities. First, the framework provides a systematic treatment for *different types of constraints* in a safety-critical problem; second, it attains *provable safety- and accuracy- guarantees* throughout the learning and control process; third, it is flexible to perform *safe learning* of any unknown aspects of a constrained decision-making system, including policy, dynamics, state and input constraints, and control cost; finally, it can be integrated to any *differentiable programming framework* to efficiently solve large-scale safe learning and control tasks.

35th Conference on Neural Information Processing Systems (NeurIPS 2021).

## 1.1 Related Work

In machine learning and control fields, safety has been defined by different criteria, such as worst-case [4, 5], risk-sensitive [6], ergodicity [7], robust [8, 9], etc., most of which are formulated by directly altering an objective function [10]. In this paper, we focus only on constrained learning and control problems, where constraints are *explicitly formulated and must be satisfied*. We categorize existing techniques into *at-convergence safety* methods, which only concern constraint satisfaction at convergence, or *in-progress safety* methods, which attempt to ensure constraint satisfaction during the entire optimization process.

**At-convergence safety methods**. In reinforcement learning (RL), a constrained agent is typically formulated as a Constrained Markov Decision Process (CMDP) [11], seeking a policy that not only optimizes a reward but also satisfies an upper bound for a cumulative cost. A common strategy [12–17] to solve CMDPs is to use the primal-dual method, by establishing the unconstrained Lagrangian and performing saddle-point optimization. In deep learning, the primal-dual method has been recently used [18] to train deep neural networks with constraints. In control, the primal-dual method has been used to solve constrained optimal control (constrained trajectory) problems [19–21]. While proved to satisfy constraints at convergence [22, 23], the primal-dual type methods cannot guarantee constraint satisfaction during optimization, as shown in [13, 24], thus are not suitable for safety-critical tasks.

**In-progress safety methods.** To enforce safety during training, [25] and [26] solve CMDPs by introducing additional constraints into the Trust Region Policy Optimization (TRPO) [27]. Since these methods only obtain the 'near constraint' guarantee, constraint violation is not fully eliminated. Another line of constrained RL [24, 28–30] leverages the Lyapunov theory [31] to bound behavior of an agent. But how to choose a valid Lyapunov function for general tasks is still an open problem to date [32], particularly for constrained RL, since it requires a Lyapunov function to be consistent with the constraints and to permit optimal policies [28]. Some other work also attempts to handle immediate constraints — the constraints imposed on agent state and input at any time. In [33], a safe exploration scheme is proposed to produce a safe reachable region; and it only considers finite state space. [34] develops a method that learns safety constraints and then optimizes a reward within the certified safe region; the method defines constraints purely on agent state and thus may not be readily applicable to mixed state-input constraints.

In control, in-progress safety can be achieved via two model-based frameworks: reachability theory [35] and control barrier functions [36, 37]. Safe control based on reachability theory [35, 38–40] explicitly considers adversarial factors and seeks a strategy that maintains the constraints despite the adversarial factors. This process typically requires solving the Hamilton-Jacobi-Isaacs equations [41], which become computationally difficult for high-dimensional systems [35]. Control barrier functions [36, 37] constrain a system only on safety boundaries, making it a less-conservative strategy for safety-critical tasks [42–44]. Most of the methods consider affine dynamics and directly use the given constraint function as a control barrier function. Such a choice could be problematic when a system is uncontrollable at the boundary of the sublevel set. Thus, how to find a valid control barrier function is still an ongoing research topic [45–47]. The above two control safety frameworks favorably focus on pure state constraints and cannot be readily extended to other constraints, such as mixed state-input constraints or the cumulative constraints defined on the system trajectory.

**Interior-point methods and control.** Interior-point methods (IPMs) [48–50] solve constrained optimization by sequentially finding solutions to unconstrained problems with the objective combining the original objective and a barrier that prevents from leaving the feasible regions. IPMs have been used for *constrained* linear quadratic regular (LQR) control in [51–57]. While IPMs for nonlinear constrained optimal control are studied in [58–62], they mostly focus on developing algorithms to solve the unconstrained approximation (from the perturbed KKT conditions) and lack of performance analysis. Most recently, [63] uses the IPM to develop a zero-th order non-convex optimization method; and [64] uses IPMs to solve reinforcement learning with only cumulative constraints. Despite the promise of the trend, the theoretical results and systematic algorithms regarding the *differentiability of general constrained control systems based on IPMs* have not been studied and established.

**Differentiable projection layer.** In machine learning, a recent line of work considers embedding a differentiable projection layer [65–67] into a general training process to ensure safety. Particularly, [67] and [66] enforce safety by constructing a dedicated projection layer, which projects the unsafe actions outputted from a neural policy into a safe region (satisfying safety constraints). This projection layer is a differentiable convex layer [68, 69], which can be trained end-to-end. In [65], safety is

defined as robustness in the case of the worst adversarial disturbance, and the set of robust policies is solved by classic robust control (solving LMIs). An RL neural policy with a differentiable projection layer is learned such that the action from the neural policy lies in the robust policy set. Different from the above work, Safe PDP does not enforce safety by projection; instead, Safe PDP uses *barrier functions* to guarantee safety constraint satisfaction. More importantly, we have shown, in both theory and experiments, that *with barrier functions, differentiability can also be attained.*

**Sensitivity analysis and differentiable MPCs.**    Other work related to Safe PDP includes the recent results for sensitivity analysis [70], which focuses on differentiation of a solution to a general nonlinear program, and differentiable MPCs [68], which is based on differentiable quadratic programming. In long-horizon control settings, directly applying [70] and [68] can be inefficient: the complexity of [68, 70] for differentiating a solution to a general optimal control system is at least $\mathcal{O}(T^2)$ ($T$ is the time horizon) due to computing the inverse of Jacobian of the KKT conditions. Since an optimal control system has more sparse structures than general nonlinear or quadratic programs, by exploiting those structures and proposing the *Auxiliary Control System*, Safe PDP enjoys the complexity of $\mathcal{O}(T)$ for differentiating a solution to a general control system. Such advantages have been discussed and shown in the foundational PDP work [71] and will also be shown later (in Section 8) in this paper.

## 1.2   Paper Contributions

We propose a safe differentiable programming methodology named as *Safe Pontryagin Differentiable Programming* (Safe PDP). Safe PDP provides a systematic treatment of different types of system constraints, including state and inputs (or mixed), immediate, and long-term constraints, with provable safety- and performance-guarantee. Safe PDP is also a unified differentiable programming framework, which can be used to efficiently solve a broad class of safety-critical learning and control tasks.

In the spirit of interior-point methods, Safe PDP incorporates different types of system constraints into control cost and loss through barrier functions, approximating a constrained control system and task using their unconstrained counterparts. Contributions of Safe PDP are theoretical and algorithmic. Theoretically, we prove in Theorem 2 and Theorem 3 that (I) not only a solution but also the gradient of the solution can be safely approximated by solving a more efficient unconstrained counterpart; (II) any intermediate results throughout the approximation and optimization are strictly safe, that is, never violating the original system/task constraints; and (III) the approximations for both solution and its gradient can be controlled for arbitrary accuracy by barrier parameters. Arithmetically, (IV) we prove in Theorem 1 that if a constrained control system is differentiable, the gradient of its trajectory is a globally unique solution to an Auxiliary Control System [71], which can be solved efficiently with the complexity of only $\mathcal{O}(T)$, $T$ is control horizon; (V) in Section 7, we experimentally demonstrate the capability of Safe PDP for efficiently solving various safety-critical learning and control problems, including safe neural policy optimization, safe motion planning, learning MPCs from demonstrations.

## 2   Safe PDP Problem Formulation

Consider a class of constrained optimal control systems (models) $\boldsymbol{\Sigma}(\boldsymbol{\theta})$, which are parameterized by a tunable parameter $\boldsymbol{\theta} \in \mathbb{R}^r$ in its control cost function, dynamics, initial condition, and constraints:

$$
\boldsymbol{\Sigma}(\boldsymbol{\theta}) : \quad
\begin{aligned}
\textit{control cost:} \quad & J(\boldsymbol{\theta}) = \sum\nolimits_{t=0}^{T-1} c_t(\boldsymbol{x}_t, \boldsymbol{u}_t, \boldsymbol{\theta}) + c_T(\boldsymbol{x}_T, \boldsymbol{\theta}) \\
\textit{subject to} \quad & \\
\textit{dynamics:} \quad & \boldsymbol{x}_{t+1} = \boldsymbol{f}(\boldsymbol{x}_t, \boldsymbol{u}_t, \boldsymbol{\theta}) \quad \text{with} \quad \boldsymbol{x}_0 = \boldsymbol{x}_0(\boldsymbol{\theta}), \quad \forall t, \\
\textit{terminal constraints:} \quad & \boldsymbol{g}_T(\boldsymbol{x}_T, \boldsymbol{\theta}) \leq \boldsymbol{0}, \quad \boldsymbol{h}_T(\boldsymbol{x}_T, \boldsymbol{\theta}) = \boldsymbol{0}, \\
\textit{path constraints:} \quad & \boldsymbol{g}_t(\boldsymbol{x}_t, \boldsymbol{u}_t, \boldsymbol{\theta}) \leq \boldsymbol{0}, \quad \boldsymbol{h}_t(\boldsymbol{x}_t, \boldsymbol{u}_t, \boldsymbol{\theta}) = \boldsymbol{0}, \quad \forall t.
\end{aligned}
\tag{1}
$$

Here, $\boldsymbol{x}_t \in \mathbb{R}^n$ is the system state; $\boldsymbol{u}_t \in \mathbb{R}^m$ is the control input; $c_t : \mathbb{R}^n \times \mathbb{R}^m \times \mathbb{R}^r \to \mathbb{R}$ and $c_T : \mathbb{R}^n \times \mathbb{R}^r \to \mathbb{R}$ are the stage and final costs, respectively; $\boldsymbol{f} : \mathbb{R}^n \times \mathbb{R}^m \times \mathbb{R}^r \to \mathbb{R}^n$ is the dynamics with initial state $\boldsymbol{x}_0 = \boldsymbol{x}_0(\boldsymbol{\theta}) \in \mathbb{R}^n$; $t = 0, 1, ..., T$ is the time step with $T$ the time horizon; $\boldsymbol{g}_T : \mathbb{R}^n \times \mathbb{R}^r \to \mathbb{R}^{q_\mathrm{T}}$ and $\boldsymbol{h}_T : \mathbb{R}^n \times \mathbb{R}^r \to \mathbb{R}^{s_\mathrm{T}}$ are the final inequality and equality constraints, respectively; $\boldsymbol{g}_t : \mathbb{R}^n \times \mathbb{R}^m \times \mathbb{R}^r \to \mathbb{R}^{q_t}$ and $\boldsymbol{h}_t : \mathbb{R}^n \times \mathbb{R}^m \times \mathbb{R}^r \to \mathbb{R}^{s_t}$ are the immediate inequality and equality constraints at time $t$, respectively. All inequalities (here and below) are entry-wise. We consider that all functions in $\boldsymbol{\Sigma}(\boldsymbol{\theta})$ are three-times continuously differentiable (i.e., $C^3$ with respect to its arguments. Although we here have parameterized all aspects of $\boldsymbol{\Sigma}(\boldsymbol{\theta})$, for a specific application (see Section 7), one only needs to parameterize the unknown aspects in $\boldsymbol{\Sigma}(\boldsymbol{\theta})$ and keep others given.

Any unknown aspects in $\boldsymbol{\Sigma}(\boldsymbol{\theta})$ can be implemented by differentiable neural networks. For a given $\boldsymbol{\theta}$, $\boldsymbol{\Sigma}(\boldsymbol{\theta})$ produces a trajectory $\boldsymbol{\xi}_{\boldsymbol{\theta}} = \{\boldsymbol{x}_{0:T}^{\boldsymbol{\theta}}, \boldsymbol{u}_{0:T-1}^{\boldsymbol{\theta}}\}$ by solving the following Problem $\mathrm{B}(\boldsymbol{\theta})$:

$$\boldsymbol{\xi}_{\boldsymbol{\theta}} = \{\boldsymbol{x}_{0:T}^{\boldsymbol{\theta}}, \boldsymbol{u}_{0:T-1}^{\boldsymbol{\theta}}\} \in \arg\min_{\{\boldsymbol{x}_{0:T}, \boldsymbol{u}_{0:T-1}\}} \quad J(\boldsymbol{\theta})$$

$$\text{subject to} \quad \boldsymbol{x}_{t+1} = \boldsymbol{f}(\boldsymbol{x}_t, \boldsymbol{u}_t, \boldsymbol{\theta}) \quad \text{with} \quad \boldsymbol{x}_0 = \boldsymbol{x}_0(\boldsymbol{\theta}), \quad \forall t,$$
$$\boldsymbol{g}_T(\boldsymbol{x}_T, \boldsymbol{\theta}) \leq \boldsymbol{0} \quad \text{and} \quad \boldsymbol{h}_T(\boldsymbol{x}_T, \boldsymbol{\theta}) = \boldsymbol{0}, \qquad \mathrm{B}(\boldsymbol{\theta})$$
$$\boldsymbol{g}_t(\boldsymbol{x}_t, \boldsymbol{u}_t, \boldsymbol{\theta}) \leq \boldsymbol{0} \quad \text{and} \quad \boldsymbol{h}_t(\boldsymbol{x}_t, \boldsymbol{u}_t, \boldsymbol{\theta}) = \boldsymbol{0}, \quad \forall t.$$

Here, we use $\in$ since $\boldsymbol{\xi}_{\boldsymbol{\theta}}$ to Problem $\mathrm{B}(\boldsymbol{\theta})$ may not be unique in general, thus constituting a solution set $\{\boldsymbol{\xi}_{\boldsymbol{\theta}}\}$. We will discuss the existence and uniqueness of $\{\boldsymbol{\xi}_{\boldsymbol{\theta}}\}$ in Section 4.

For a specific task, we aim to find a specific model $\boldsymbol{\Sigma}(\boldsymbol{\theta}^*)$, i.e, searching for a specific $\boldsymbol{\theta}^*$, such that its trajectory $\boldsymbol{\xi}_{\boldsymbol{\theta}^*}$ from $\mathrm{B}(\boldsymbol{\theta}^*)$ meets the following two *given* requirements. First, $\boldsymbol{\xi}_{\boldsymbol{\theta}^*}$ minimizes a *given task loss* $\ell(\boldsymbol{\xi}_{\boldsymbol{\theta}}, \boldsymbol{\theta})$; and second, $\boldsymbol{\xi}_{\boldsymbol{\theta}^*}$ satisfies the *given task constraints* $R_i(\boldsymbol{\xi}_{\boldsymbol{\theta}}, \boldsymbol{\theta}) \leq 0$, $i = 1, 2, ..., l$. Note that, we need to distinguish between the two types of objectives: task loss $\ell(\boldsymbol{\xi}_{\boldsymbol{\theta}}, \boldsymbol{\theta})$ and control cost $J(\boldsymbol{\theta})$, and also the two types of constraints: task constraints $R_i(\boldsymbol{\xi}_{\boldsymbol{\theta}}, \boldsymbol{\theta})$ and model constraints $\boldsymbol{g}_t(\boldsymbol{\theta})$. In fact, $J(\boldsymbol{\theta})$ and $\boldsymbol{g}_t(\boldsymbol{\theta})$ in $\boldsymbol{\Sigma}(\boldsymbol{\theta})$ are *unknown and parameterized by $\boldsymbol{\theta}$* and can represent the unknown inherent aspects of a physical agent, while $\ell(\boldsymbol{\xi}_{\boldsymbol{\theta}}, \boldsymbol{\theta})$ and $R_i(\boldsymbol{\xi}_{\boldsymbol{\theta}}, \boldsymbol{\theta})$ are *given and known* depending on the specific task (they also explicitly depend on $\boldsymbol{\theta}$ since $\boldsymbol{\theta}$ needs to be regularized in some learning cases). Assume $\ell(\boldsymbol{\xi}_{\boldsymbol{\theta}}, \boldsymbol{\theta})$ and $R_i(\boldsymbol{\xi}_{\boldsymbol{\theta}}, \boldsymbol{\theta})$ are both twice-continuously differentiable. The problem of searching for $\boldsymbol{\theta}^*$ can be formally written as:

$$\boldsymbol{\theta}^* = \arg\min_{\boldsymbol{\theta}} \quad \ell(\boldsymbol{\xi}_{\boldsymbol{\theta}}, \boldsymbol{\theta})$$

$$\text{subject to} \quad R_i(\boldsymbol{\xi}_{\boldsymbol{\theta}}, \boldsymbol{\theta}) \leq 0, \quad i = 1, 2, ..., l, \qquad \mathrm{P}$$
$$\boldsymbol{\xi}_{\boldsymbol{\theta}} \text{ solves Problem } \mathrm{B}(\boldsymbol{\theta}) .$$

For a specific learning and control task, one only needs to specify the details of $\boldsymbol{\Sigma}(\boldsymbol{\theta})$ and give a task loss $\ell(\boldsymbol{\xi}_{\boldsymbol{\theta}}, \boldsymbol{\theta})$ and constraints $R_i(\boldsymbol{\xi}_{\boldsymbol{\theta}}, \boldsymbol{\theta}) \leq 0$. Section 7 will give representative examples.

## 3 Challenges to Solve Problem P

Problem P belongs to bi-level optimization [72]—each time $\boldsymbol{\theta}$ is updated in the outer-level (including task loss $\ell$ and task constraint $R_i$) of Problem P, the corresponding trajectory $\boldsymbol{\xi}_{\boldsymbol{\theta}}$ needs to be solved from the inner-level Problem $\mathrm{B}(\boldsymbol{\theta})$. Similar to PDP [71], one could approach Problem P using gradient-based methods by ignoring the process of solving inner-level Problem $\mathrm{B}(\boldsymbol{\theta})$ and just viewing $\boldsymbol{\xi}_{\boldsymbol{\theta}}$ as an explicit differentiable function of $\boldsymbol{\theta}$. Then, based on interior-point methods [48], one can introduce a logarithmic barrier function for each task constraint, $-\ln(-R_i)$, and a barrier parameter $\epsilon > 0$. This leads to solving the following unconstrained Problem $\mathrm{SP}(\epsilon)$ sequentially

$$\boldsymbol{\theta}^*(\epsilon) = \arg\min_{\boldsymbol{\theta}} \ \ell(\boldsymbol{\xi}_{\boldsymbol{\theta}}, \boldsymbol{\theta}) - \epsilon \sum_{i=1}^{l} \ln\left(-R_i(\boldsymbol{\xi}_{\boldsymbol{\theta}}, \boldsymbol{\theta})\right) \qquad \mathrm{SP}(\epsilon)$$

for a fixed $\epsilon$. By controlling $\epsilon \to 0$, $\boldsymbol{\theta}^*(\epsilon)$ is expected to converge to the solution $\boldsymbol{\theta}^*$ to Problem P. Although plausible, the above process has the following technical challenges to be addressed:

(1) As $\boldsymbol{\xi}_{\boldsymbol{\theta}}$ is a solution to the constrained optimal control Problem $\mathrm{B}(\boldsymbol{\theta})$, is $\boldsymbol{\xi}_{\boldsymbol{\theta}}$ differentiable? Does the auxiliary control system [71] exist for solving $\frac{\partial \boldsymbol{\xi}_{\boldsymbol{\theta}}}{\partial \boldsymbol{\theta}}$?

(2) Since we want to obtain $\boldsymbol{\xi}_{\boldsymbol{\theta}}$ and $\frac{\partial \boldsymbol{\xi}_{\boldsymbol{\theta}}}{\partial \boldsymbol{\theta}}$ at as low cost as possible, instead of solving the constrained Problem $\mathrm{B}(\boldsymbol{\theta})$, can we use an *unconstrained* system to approximate both $\boldsymbol{\xi}_{\boldsymbol{\theta}}$ and $\frac{\partial \boldsymbol{\xi}_{\boldsymbol{\theta}}}{\partial \boldsymbol{\theta}}$? Importantly, can the accuracy of the approximations for $\boldsymbol{\xi}_{\boldsymbol{\theta}}$ and $\frac{\partial \boldsymbol{\xi}_{\boldsymbol{\theta}}}{\partial \boldsymbol{\theta}}$ be arbitrarily and safely controlled?

(3) Can we guarantee that the approximation for $\boldsymbol{\xi}_{\boldsymbol{\theta}}$ is safe in a sense that the approximation always respects the system original inequality constraints $\boldsymbol{g}_t \leq \boldsymbol{0}$ and $\boldsymbol{g}_T \leq \boldsymbol{0}$?

(4) With the safe approximations for both $\boldsymbol{\xi}_{\boldsymbol{\theta}}$ and $\frac{\partial \boldsymbol{\xi}_{\boldsymbol{\theta}}}{\partial \boldsymbol{\theta}}$, can accuracy of the solution to the outer-level unconstrained optimization $\mathrm{SP}(\epsilon)$ be arbitrarily controlled towards $\boldsymbol{\theta}^*$?

(5) With the safe approximations for both $\boldsymbol{\xi}_{\boldsymbol{\theta}}$ and $\frac{\partial \boldsymbol{\xi}_{\boldsymbol{\theta}}}{\partial \boldsymbol{\theta}}$, can we guarantee the safety of the outer-level inequality constraints $R_i \leq 0$ during the optimization for the outer-level $\mathrm{SP}(\epsilon)$?

The following paper will address the above challenges. For reference, we give a quick overview: Challenge (1) will be addressed in Section 4 and the result is in Theorem 1; Challenges (2) and (3) will be addressed in Section 5 and the result is in Theorem 2; Challenges (4) and (5) will be addressed in Section 6 and the result is in Theorem 3; and Section 7 gives some representative applications.

# 4 Differentiability for $\Sigma(\boldsymbol{\theta})$ and its Auxiliary Control System

## 4.1 Differentiability of $\boldsymbol{\xi}_{\boldsymbol{\theta}}$

For the constrained optimal control system $\Sigma(\boldsymbol{\theta})$ in (1), we define the following Hamiltonian $L_t$ for $t = 0, 1, ..., T-1$ and $L_T$, respectively,

$$L_t = c_t(\boldsymbol{x}_t, \boldsymbol{u}_t, \boldsymbol{\theta}) + \boldsymbol{\lambda}'_{t+1} \boldsymbol{f}(\boldsymbol{x}_t, \boldsymbol{u}_t, \boldsymbol{\theta}) + \boldsymbol{v}'_t \boldsymbol{g}_t(\boldsymbol{x}_t, \boldsymbol{u}_t, \boldsymbol{\theta}) + \boldsymbol{w}'_t \boldsymbol{h}_t(\boldsymbol{x}_t, \boldsymbol{u}_t, \boldsymbol{\theta}), \qquad (2a)$$

$$L_T = c_T(\boldsymbol{x}_T, \boldsymbol{\theta}) + \boldsymbol{v}'_T \boldsymbol{g}_T(\boldsymbol{x}_T, \boldsymbol{\theta}) + \boldsymbol{w}'_T \boldsymbol{h}_T(\boldsymbol{x}_T, \boldsymbol{\theta}), \qquad (2b)$$

where $\boldsymbol{\lambda}_t \in \mathbb{R}^n$ is the costate, $\boldsymbol{v}_t \in \mathbb{R}^{q_t}$ and $\boldsymbol{w}_t \in \mathbb{R}^{s_t}$ are multipliers for the inequality and equality constraints, respectively. The well-known second-order condition for $\boldsymbol{\xi}_{\boldsymbol{\theta}}$ to be *a local isolated (locally unique) minimizing trajectory* to $\Sigma(\boldsymbol{\theta})$ in Problem B($\boldsymbol{\theta}$) has been well-established in [73]. For completeness, we present it in Lemma A.2 in Appendix A. Lemma A.2 states that there exist costate sequence $\boldsymbol{\lambda}^{\boldsymbol{\theta}}_{1:T}$, and multiplier sequences $\boldsymbol{v}^{\boldsymbol{\theta}}_{0:T}$ and $\boldsymbol{w}^{\boldsymbol{\theta}}_{0:T}$, such that $(\boldsymbol{\xi}_{\boldsymbol{\theta}}, \boldsymbol{\lambda}^{\boldsymbol{\theta}}_{1:T}, \boldsymbol{v}^{\boldsymbol{\theta}}_{0:T}, \boldsymbol{w}^{\boldsymbol{\theta}}_{0:T})$ satisfies the well-known Constrained Pontryagin Minimum Principle (C-PMP) given in (S.5) in Lemma A.2. Based on the above, one can have the following result for the differentiability of $\boldsymbol{\xi}_{\boldsymbol{\theta}}$.

**Lemma 1** (Differentiability of $\boldsymbol{\xi}_{\boldsymbol{\theta}}$). *Given a fixed $\bar{\boldsymbol{\theta}}$, assume the following conditions hold for $\Sigma(\bar{\boldsymbol{\theta}})$:*

*(i) the second-order condition (Lemma A.2) is satisfied for $\Sigma(\bar{\boldsymbol{\theta}})$;*

*(ii) the gradients of all binding constraints at $\boldsymbol{\xi}_{\bar{\boldsymbol{\theta}}}$ are linearly independent (binding constraints include all equality constraints and all active inequality constraints);*

*(iii) strict complementarity holds at $\boldsymbol{\xi}_{\bar{\boldsymbol{\theta}}}$, i.e., active inequality constraint has positive multiplier.*

*Then, for all $\boldsymbol{\theta}$ in a neighborhood of $\bar{\boldsymbol{\theta}}$, there exists a unique once-continuously differentiable function $(\boldsymbol{\xi}_{\boldsymbol{\theta}}, \boldsymbol{\lambda}^{\boldsymbol{\theta}}_{1:T}, \boldsymbol{v}^{\boldsymbol{\theta}}_{0:T}, \boldsymbol{w}^{\boldsymbol{\theta}}_{0:T})$ that satisfies the second-order condition (Lemma A.2) for the constrained optimal control system $\Sigma(\boldsymbol{\theta})$ with $(\boldsymbol{\xi}_{\boldsymbol{\theta}}, \boldsymbol{\lambda}^{\boldsymbol{\theta}}_{1:T}, \boldsymbol{v}^{\boldsymbol{\theta}}_{0:T}, \boldsymbol{w}^{\boldsymbol{\theta}}_{0:T}) = (\boldsymbol{\xi}_{\bar{\boldsymbol{\theta}}}, \boldsymbol{\lambda}^{\bar{\boldsymbol{\theta}}}_{1:T}, \boldsymbol{v}^{\bar{\boldsymbol{\theta}}}_{0:T}, \boldsymbol{w}^{\bar{\boldsymbol{\theta}}}_{0:T})$ at $\boldsymbol{\theta} = \bar{\boldsymbol{\theta}}$. Hence, $\boldsymbol{\xi}_{\boldsymbol{\theta}}$ is a local isolated minimizing trajectory to $\Sigma(\boldsymbol{\theta})$. Further, for all $\boldsymbol{\theta}$ near $\bar{\boldsymbol{\theta}}$, the strict complementarity is preserved, and the linear independence of the gradients of all binding constraints at $\boldsymbol{\xi}_{\boldsymbol{\theta}}$ hold.*

The proof of Lemma 1 can directly follow the well-known first-order sensitivity result in Theorem 2.1 in [74]. Here, conditions (i)-(iii) are the sufficient conditions to guarantee the applicability of the well-known implicit function theorem [75] to the C-PMP. Condition (ii) is well-known and serves as a sufficient condition for the constraint qualification to establish the C-PMP (see Corollary 3, pp. 22, [48]). Condition (iii) is necessary to ensure that the Jacobian matrix in the implicit function theorem is invertible, and it also leads to the persistence of strict complementarity, saying that the inactive inequalities remain inactive and active ones remain active and there is no 'switching' between them near $\bar{\boldsymbol{\theta}}$. Both our practice and previous works [68, 69, 71, 74, 76] show that the conditions (i)-(iii) are very mild and the differentiability of $\boldsymbol{\xi}_{\boldsymbol{\theta}}$ can be attained almost everywhere in the space of $\boldsymbol{\theta}$.

## 4.2 Auxiliary Control System to Solve $\frac{\partial \boldsymbol{\xi}_{\boldsymbol{\theta}}}{\partial \boldsymbol{\theta}}$

If the conditions (i)-(iii) in Lemma 1 for differentiability of $\boldsymbol{\xi}_{\boldsymbol{\theta}}$ hold, we next show that $\frac{\partial \boldsymbol{\xi}_{\boldsymbol{\theta}}}{\partial \boldsymbol{\theta}}$ can also be efficiently solved by an auxiliary control system, which is originally proposed in the foundational work [71]. First, we define the new state and input (matrix) variables $X_t \in \mathbb{R}^{n \times r}$ and $U_t \in \mathbb{R}^{m \times r}$, respectively. Then, we introduce the following *auxiliary control system*,

$$\overline{\Sigma}(\boldsymbol{\xi}_{\boldsymbol{\theta}}):$$

$$\begin{aligned}
\textit{control cost:} \quad & \bar{J} = \text{Tr} \sum_{t=0}^{T-1} \left( \frac{1}{2} \begin{bmatrix} X_t \\ U_t \end{bmatrix}' \begin{bmatrix} L_t^{xx} & L_t^{xu} \\ L_t^{ux} & L_t^{uu} \end{bmatrix} \begin{bmatrix} X_t \\ U_t \end{bmatrix} + \begin{bmatrix} L_t^{x\theta} \\ L_t^{u\theta} \end{bmatrix}' \begin{bmatrix} X_t \\ U_t \end{bmatrix} \right) \\
& + \text{Tr} \left( \frac{1}{2} X_T' L_T^{xx} X_T + (L_T^{x\theta})' X_T \right) \\
\textit{subject to} \\
\textit{dynamics:} \quad & X_{t+1} = F_t^x X_t + F_t^u U_t + F_t^\theta \quad \text{with} \quad X_0 = X_0^{\boldsymbol{\theta}} \\
\textit{terminal constraint:} \quad & \bar{G}_T^x X_T + \bar{G}_T^\theta = \mathbf{0}, \qquad H_T^x X_T + H_T^\theta = \mathbf{0}, \\
\textit{path constraint:} \quad & \bar{G}_t^x X_t + \bar{G}_t^u U_t + \bar{G}_t^\theta = \mathbf{0}, \qquad H_t^x X_t + H_t^u U_t + H_t^\theta = \mathbf{0}.
\end{aligned} \qquad (3)$$

Here, $L_t^x$ and $L_t^{xx}$ denote the first- and second- order derivatives, respectively, of the Hamiltonian $L_t$ in (2) with respect to $\boldsymbol{x}$; $F_t^x$, $H_t^x$, $\bar{G}_t$ denote the first-order derivatives of $\boldsymbol{f}_t$, $\boldsymbol{h}_t$, $\bar{\boldsymbol{g}}_t$ with respect

to $\boldsymbol{x}$, respectively, where $\bar{\boldsymbol{g}}_t$ is the vector of stacking all active inequality constraints in $\boldsymbol{g}_t$; and the similar convention applies to the other notations. All derivative matrices defining $\overline{\Sigma}(\boldsymbol{\xi}_{\boldsymbol{\theta}})$ are evaluated at $(\boldsymbol{\xi}_{\boldsymbol{\theta}}, \boldsymbol{\lambda}^{\boldsymbol{\theta}}_{1:T}, \boldsymbol{v}^{\boldsymbol{\theta}}_{0:T}, \boldsymbol{w}^{\boldsymbol{\theta}}_{0:T})$, where $\boldsymbol{\lambda}^{\boldsymbol{\theta}}_{1:T}$, $\boldsymbol{v}^{\boldsymbol{\theta}}_{0:T}$, and $\boldsymbol{w}^{\boldsymbol{\theta}}_{0:T}$ are usually the byproducts of a constrained optimal control solver [77] or can be easily solved from the C-PMP given $\boldsymbol{\xi}_{\boldsymbol{\theta}}$, as done in [71]. We note that $\overline{\Sigma}(\boldsymbol{\xi}_{\boldsymbol{\theta}})$ is a *Equality-constrained Linear Quadratic Regulator (LQR)* system, as its control cost function is quadratic and dynamics and constraints are linear. For the above $\overline{\Sigma}(\boldsymbol{\xi}_{\boldsymbol{\theta}})$, we have the following important result *without additional assumptions*.

**Theorem 1** ($\frac{\partial \boldsymbol{\xi}_{\boldsymbol{\theta}}}{\partial \boldsymbol{\theta}}$ is a globally unique minmizing trajectory to $\overline{\Sigma}(\boldsymbol{\xi}_{\boldsymbol{\theta}})$)**.** *Let the conditions (i)-(iii) in Lemma 1 for differentiability of $\boldsymbol{\xi}_{\boldsymbol{\theta}}$ hold. Then, the auxiliary control system $\overline{\Sigma}(\boldsymbol{\xi}_{\boldsymbol{\theta}})$ in (3) has a globally unique minimizing trajectory, denoted as $\left\{X^{\boldsymbol{\theta}}_{0:T}, U^{\boldsymbol{\theta}}_{0:T-1}\right\}$, which is exactly $\frac{\partial \boldsymbol{\xi}_{\boldsymbol{\theta}}}{\partial \boldsymbol{\theta}}$, i.e.,*

$$\left\{X^{\boldsymbol{\theta}}_{0:T}, U^{\boldsymbol{\theta}}_{0:T-1}\right\} = \frac{\partial \boldsymbol{\xi}_{\boldsymbol{\theta}}}{\partial \boldsymbol{\theta}} \quad with \quad X^{\boldsymbol{\theta}}_t = \frac{\partial \boldsymbol{x}^{\boldsymbol{\theta}}_t}{\partial \boldsymbol{\theta}} \quad and \quad U^{\boldsymbol{\theta}}_t = \frac{\partial \boldsymbol{u}^{\boldsymbol{\theta}}_t}{\partial \boldsymbol{\theta}}. \tag{4}$$

The proof of the above theorem is in Appendix B. Theorem 1 states that as long as the conditions (i)-(iii) in Lemma 1 for differentiability of $\boldsymbol{\xi}_{\boldsymbol{\theta}}$ are satisfied, without additional assumptions, the auxiliary control system $\overline{\Sigma}(\boldsymbol{\xi}_{\boldsymbol{\theta}})$ always has a globally unique minimizing trajectory, which is exactly $\frac{\partial \boldsymbol{\xi}_{\boldsymbol{\theta}}}{\partial \boldsymbol{\theta}}$. Thus, obtaining $\frac{\partial \boldsymbol{\xi}_{\boldsymbol{\theta}}}{\partial \boldsymbol{\theta}}$ is equivalent to solving $\overline{\Sigma}(\boldsymbol{\xi}_{\boldsymbol{\theta}})$, which be efficiently done thanks to the recent development of the equality-constrained LQR algorithms [78–80], all of which have a complexity of $\mathcal{O}(T)$. The algorithm that implements Theorem 1 is given in Algorithm 1 in Appendix E.1.

# 5 Safe Unconstrained Approximations for $\boldsymbol{\xi}_{\boldsymbol{\theta}}$ and $\frac{\partial \boldsymbol{\xi}_{\boldsymbol{\theta}}}{\partial \boldsymbol{\theta}}$

From Section 4, we know that one can solve the constrained system $\Sigma(\boldsymbol{\theta})$ to obtain $\boldsymbol{\xi}_{\boldsymbol{\theta}}$ and solve its auxiliary control system $\overline{\Sigma}(\boldsymbol{\xi}_{\boldsymbol{\theta}})$ to obtain $\frac{\partial \boldsymbol{\xi}_{\boldsymbol{\theta}}}{\partial \boldsymbol{\theta}}$. Although theoretically appealing, there are several difficulties in implementation. First, solving a constrained optimal control Problem B($\boldsymbol{\theta}$) is not as easy as solving an unconstrained optimal control, for which many trajectory optimization algorithms, e.g., iLQR [81] and DDP [82], are available. Second, establishing $\overline{\Sigma}(\boldsymbol{\xi}_{\boldsymbol{\theta}})$ requires the values of the multipliers $\boldsymbol{v}^{\boldsymbol{\theta}}_{0:T}$ and $\boldsymbol{w}^{\boldsymbol{\theta}}_{0:T}$. And third, to construct $\overline{\Sigma}(\boldsymbol{\xi}_{\boldsymbol{\theta}})$, one also needs to identify all active inequality constraints $\bar{\boldsymbol{g}}_t$, which can be numerically difficult due to numerical error (we will show this in later experiments). All those difficulties motivate us to develop a more efficient paradigm to obtain both $\boldsymbol{\xi}_{\boldsymbol{\theta}}$ and $\frac{\partial \boldsymbol{\xi}_{\boldsymbol{\theta}}}{\partial \boldsymbol{\theta}}$, which is the goal of this section.

To proceed, we first convert the constrained system $\Sigma(\boldsymbol{\theta})$ to an unconstrained system $\Sigma(\boldsymbol{\theta}, \gamma)$ by adding all constraints to its control cost via barrier functions. Here, we use quadratic barrier function for each equality constraint and logarithm barrier functions for each inequality constraint; and all barrier functions are associated with the same barrier parameter $\gamma > 0$. This leads to $\Sigma(\boldsymbol{\theta}, \gamma)$ to be

$$\Sigma(\boldsymbol{\theta}, \gamma): \quad \begin{aligned} \textit{control cost:} \quad J(\boldsymbol{\theta}, \gamma) &= \sum_{t=0}^{T-1} \Bigg( c_t(\boldsymbol{x}_t, \boldsymbol{u}_t, \boldsymbol{\theta}) - \gamma \sum_{i=1}^{q_t} \ln\left(-g_{t,i}(\boldsymbol{x}_t, \boldsymbol{u}_t, \boldsymbol{\theta})\right) + \\ &\quad \frac{1}{2\gamma} \sum_{i=1}^{s_t} \left(h_{t,i}(\boldsymbol{x}_t, \boldsymbol{u}_t, \boldsymbol{\theta})\right)^2 \Bigg) + c_T(\boldsymbol{x}_T, \boldsymbol{\theta}) - \\ &\quad \gamma \sum_{i=1}^{q_T} \ln\left(-g_{T,i}(\boldsymbol{x}_T, \boldsymbol{\theta})\right) + \frac{1}{2\gamma} \sum_{i=1}^{s_T} \left(h_{T,i}(\boldsymbol{x}_T, \boldsymbol{\theta})\right)^2, \\[4pt] \textit{subject to} \\ \textit{dynamics:} \quad &\boldsymbol{x}_{t+1} = \boldsymbol{f}(\boldsymbol{x}_t, \boldsymbol{u}_t, \boldsymbol{\theta}) \quad with \quad \boldsymbol{x}_0 = \boldsymbol{x}_0(\boldsymbol{\theta}), \quad \forall t. \end{aligned} \tag{5}$$

The trajectory $\boldsymbol{\xi}_{(\boldsymbol{\theta}, \gamma)} = \left\{\boldsymbol{x}^{(\boldsymbol{\theta}, \gamma)}_{0:T}, \boldsymbol{u}^{(\boldsymbol{\theta}, \gamma)}_{0:T-1}\right\}$ produced by the above unconstrained system $\Sigma(\boldsymbol{\theta}, \gamma)$ is

$$\boldsymbol{\xi}_{(\boldsymbol{\theta}, \gamma)} = \left\{\boldsymbol{x}^{(\boldsymbol{\theta}, \gamma)}_{0:T}, \boldsymbol{u}^{(\boldsymbol{\theta}, \gamma)}_{0:T-1}\right\} \in \arg\min_{\{\boldsymbol{x}_{0:T}, \boldsymbol{u}_{0:T-1}\}} \quad J(\boldsymbol{\theta}, \gamma) \\ \text{s.t.} \quad \boldsymbol{x}_{t+1} = \boldsymbol{f}(\boldsymbol{x}_t, \boldsymbol{u}_t, \boldsymbol{\theta}) \quad with \quad \boldsymbol{x}_0 = \boldsymbol{x}_0(\boldsymbol{\theta}), \tag{SB($\boldsymbol{\theta}, \gamma$)}$$

that is, $\boldsymbol{\xi}_{(\boldsymbol{\theta}, \gamma)}$ is minimizing the new control cost $J(\boldsymbol{\theta}, \gamma)$ subject to only dynamics. Then we have the following important result about the safe unconstrained approximation for $\boldsymbol{\xi}_{\boldsymbol{\theta}}$ and $\frac{\partial \boldsymbol{\xi}_{\boldsymbol{\theta}}}{\partial \boldsymbol{\theta}}$ using $\Sigma(\boldsymbol{\theta}, \gamma)$.

**Theorem 2.** *Let conditions (i)-(iii) in Lemma 1 for differentiability of $\xi_\theta$ hold. For any small $\gamma > 0$,*

(a) *there exists a local isolated minimizing trajectory $\xi_{(\theta,\gamma)}$ that solves Problem $SB(\theta, \gamma)$, and $\Sigma(\theta, \gamma)$ is well-defined at $\xi_{(\theta,\gamma)}$, i.e., $g_t(x_t^{(\theta,\gamma)}, u_t^{(\theta,\gamma)}, \theta) < 0$ and $g_T(x_T^{(\theta,\gamma)}, \theta) < 0$;*

(b) *$\xi_{(\theta,\gamma)}$ is once-continuously differentiable with respect to $(\theta, \gamma)$, and*

$$\xi_{(\theta,\gamma)} \to \xi_\theta \quad and \quad \frac{\partial \xi_{(\theta,\gamma)}}{\partial \theta} \to \frac{\partial \xi_\theta}{\partial \theta} \quad as \quad \gamma \to 0; \qquad (6)$$

(c) *the trajectory derivative $\frac{\partial \xi_{(\theta,\gamma)}}{\partial \theta}$ is a globally unique minimizing trajectory to the auxiliary control system $\overline{\Sigma}(\xi_{(\theta,\gamma)})$ corresponding to $\Sigma(\theta, \gamma)$.*

The proof of the above theorem is given in Appendix C. It is worth noting that the above assertions require no additional assumption except the same conditions (i)-(iii) for differentiability of $\xi_\theta$. We make the following comments on the above results, using an illustrative cartpole example in Fig. 1.

First, assertion (b) states that by choosing a small $\gamma > 0$, one can simply use $\xi_{(\theta,\gamma)}$ and $\frac{\partial \xi_{(\theta,\gamma)}}{\partial \theta}$ of the *unconstrained optimal control system* $\Sigma(\theta, \gamma)$ to approximate $\xi_\theta$ and $\frac{\partial \xi_\theta}{\partial \theta}$ of the original constrained system $\Sigma(\theta)$, respectively. Second, notably, assertion (b) also states that the above approximations can be controlled for arbitrary accuracy by simply letting $\gamma \to 0$, as illustrated in the upper panels in Fig. 1. Third, more importantly, assertion (a) states that the above approximations are always safe in a sense that the approximation $\xi_{(\theta,\gamma)}$ with any small $\gamma > 0$ is guaranteed to satisfy all inequality constraints in the original $\Sigma(\theta)$, as illustrated in the bottom panels in Fig. 1. Finally, similar to Theorem 1, assertion (c) states that the derivative $\frac{\partial \xi_{(\theta,\gamma)}}{\partial \theta}$ for $\Sigma(\theta, \gamma)$ is a globally unique minimizing trajectory to its corresponding auxiliary control system $\overline{\Sigma}(\xi_{(\theta,\gamma)})$, thus PDP [71] directly applies here.

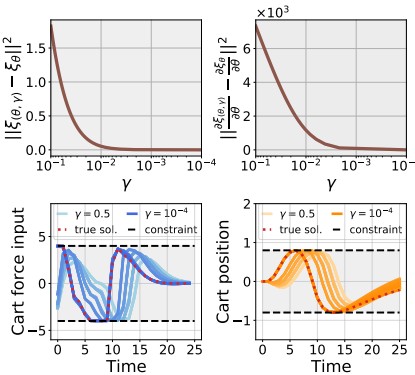

Figure 1: $\xi_{(\theta,\gamma)}$ and $\frac{\partial \xi_{(\theta,\gamma)}}{\partial \theta}$ approximate $\xi_\theta$ and $\frac{\partial \xi_\theta}{\partial \theta}$ under different $\gamma > 0$.

In addition to the theoretical importance of Theorem 2, we also summarize its algorithmic advantage compared to directly handling the original constrained system $\Sigma(\theta)$ and its auxiliary control system $\overline{\Sigma}(\xi_\theta)$. First, solving the unconstrained $\Sigma(\theta, \gamma)$ is easier than solving the constrained $\Sigma(\theta)$ as more off-the-shelf algorithms are available for unconstrained trajectory optimization than for constrained one. Second, when solving $\frac{\partial \xi_{(\theta,\gamma)}}{\partial \theta}$ using $\overline{\Sigma}(\xi_{(\theta,\gamma)})$, there is no need to identify the inactive and active inequality constraints, as opposed to solving $\frac{\partial \xi_\theta}{\partial \theta}$ using $\overline{\Sigma}(\xi_\theta)$; thus it is easier to implement and more numerically stable (we will show this later in experiments). Third, in contrast to Theorem 1, the unconstrained $\Sigma(\theta, \gamma)$ and $\overline{\Sigma}(\xi_{(\theta,\gamma)})$ avoid dealing with the multipliers $v_{0:T}$ and $w_{0:T}$. Finally, by absorbing hard inequality constraints into the control cost through barrier functions, $\Sigma(\theta, \gamma)$ introduces the 'softness' of constraints and mitigates the discontinuous 'switching' between inactive/active inequalities over a large range of $\theta$. This leads to a more numerically stable algorithm, as we will show in later experiments. Implementation of Theorem 2 is given in Algorithm 2 in Appendix E.2.

## 6   Safe PDP to Solve Problem P

According to Theorem 2, we use the safe unconstrained approximation system $\Sigma(\theta, \gamma)$ in (5) to replace the original inner-level constrained system $\Sigma(\theta)$ in (1). Then, we give the following important result for solving Problem P, which addresses the Challenges (4) and (5) in Section 3.

**Theorem 3.** *Consider all functions defining the constrained optimal control system $\Sigma(\theta)$ are at least three-times continuously differentiable, and let the conditions (i)-(iii) in Lemma 1 for differentiability of $\xi_\theta$ hold in a neighborhood of $\theta^*$. Suppose that the second-order condition for a local isolated minimizor $\theta^*$ to Problem P is satisfied, that the gradients $\nabla_\theta R_i(\xi_{\theta^*}, \theta^*)$ of all binding constraints $R_i(\xi_\theta, \theta) = 0$ are linearly independent at $\theta^*$, and that the strict complementary holds at $\theta^*$. Then, for any small $\epsilon > 0$ and any small $\gamma > 0$, the following outer-level unconstrained approximation*

$$\theta^*(\epsilon, \gamma) = \arg\min_\theta \ \ell(\xi_{(\theta,\gamma)}, \theta) - \epsilon \sum_{i=1}^l \ln\left(-R_i(\xi_{(\theta,\gamma)}, \theta)\right), \qquad SP(\epsilon, \gamma)$$

with $\boldsymbol{\xi}_{(\boldsymbol{\theta},\gamma)}$ being the optimal trajectory to the inner-level safe unconstrained approximation system $\boldsymbol{\Sigma}(\boldsymbol{\theta},\gamma)$ in (5), has the following assertions:

(a) there exists a local isolated minimizor $\boldsymbol{\theta}^*(\epsilon,\gamma)$ to the above SP$(\epsilon,\gamma)$, and the corresponding trajectory $\boldsymbol{\xi}_{(\boldsymbol{\theta}^*(\epsilon,\gamma),\gamma)}$ from the inner-level approximation system $\boldsymbol{\Sigma}\big(\boldsymbol{\theta}^*(\epsilon,\gamma),\gamma\big)$ is safe with respect the original outer-level constraints, i.e., $R_i\big(\boldsymbol{\xi}_{(\boldsymbol{\theta}^*(\epsilon,\gamma),\gamma)},\boldsymbol{\theta}^*(\epsilon,\gamma)\big) < 0$, $i = 1,2,...,l$;

(b) $\boldsymbol{\theta}^*(\epsilon,\gamma)$ is once-continuously differentiable with respect to both $\epsilon$ and $\gamma$, and

$$\boldsymbol{\theta}^*(\epsilon,\gamma) \to \boldsymbol{\theta}^* \qquad as \qquad (\epsilon,\gamma) \to (0,0); \tag{7}$$

(c) for any $\boldsymbol{\theta}$ near $\boldsymbol{\theta}^*(\epsilon,\gamma)$, $\boldsymbol{\xi}_{(\boldsymbol{\theta},\gamma)}$ from the inner-level approximation system $\boldsymbol{\Sigma}(\boldsymbol{\theta},\gamma)$ is safe with respect to the original outer-level constraints, i.e., $R_i\big(\boldsymbol{\xi}_{(\boldsymbol{\theta},\gamma)},\boldsymbol{\theta}\big) < 0$, $i = 1,2,...,l$.

The proof of the above theorem is given in Appendix D. The above result says that instead of solving the original constrained Problem P with the inner-level constrained system $\boldsymbol{\Sigma}(\boldsymbol{\theta})$ in (1), one can solve an *unconstrained approximation* Problem SP$(\epsilon,\gamma)$ with the inner-level *safe unconstrained approximation system* $\boldsymbol{\Sigma}(\boldsymbol{\theta},\gamma)$ in (5). Particularly, we make the following comments on the importance of the above theorem. First, claim (a) affirms that although the inner-level trajectory $\boldsymbol{\xi}_{(\boldsymbol{\theta},\gamma)}$ is an approximation (recall Theorem 2), the outer-level unconstrained Problem SP$(\epsilon,\gamma)$ always has a locally unique solution $\boldsymbol{\theta}^*(\epsilon,\gamma)$; furthermore, at $\boldsymbol{\theta}^*(\epsilon,\gamma)$, the corresponding inner-level trajectory $\boldsymbol{\xi}_{(\boldsymbol{\theta}^*(\epsilon,\gamma),\gamma)}$ is safe with respect to the original outer-level constraints, i.e., $R_i\big(\boldsymbol{\xi}_{(\boldsymbol{\theta}^*(\epsilon,\gamma),\gamma)},\boldsymbol{\theta}^*(\epsilon,\gamma)\big) < 0$, $i = 1,2,...,l$. Second, claim (b) asserts that the accuracy of the solution $\boldsymbol{\theta}^*(\epsilon,\gamma)$ to the outer-level approximation Problem SP$(\epsilon,\gamma)$ is controlled jointly by the inner-level barrier parameter $\gamma$ and outer-level barrier parameter $\epsilon$: as both barrier parameters approach zero, $\boldsymbol{\theta}^*(\epsilon,\gamma)$ is converging to the true solution $\boldsymbol{\theta}^*$ to the original Problem P. Third, claim (c) says that during the local search of the outer-level solution $\boldsymbol{\theta}^*(\epsilon,\gamma)$, the corresponding inner-level trajectory $\boldsymbol{\xi}_{(\boldsymbol{\theta},\gamma)}$ is always safe with respect to the original outer-level constraints, i.e., $R_i\big(\boldsymbol{\xi}_{(\boldsymbol{\theta},\gamma)},\boldsymbol{\theta}\big) < 0$, $i = 1,2,...,l$. The above Theorem 3, together with Theorem 2 provide the safety- and accuracy- guarantees for the whole Safe PDP framework. Then entire Safe PDP algorithm is given in Algorithm 3 in Appendix E.3.

## 7   Applications to Different Safety-Critical Tasks

We apply Safe PDP to solve some representative safety-critical learning/control tasks. For a specific task, one only needs to specify the parameterization detail of $\boldsymbol{\Sigma}(\boldsymbol{\theta})$, a task loss $\ell(\boldsymbol{\xi}_{\boldsymbol{\theta}},\boldsymbol{\theta})$, and task constraints $R_i(\boldsymbol{\xi}_{\boldsymbol{\theta}},\boldsymbol{\theta})$ in Problem P. The experiments are performed on the systems of different complexities in Table 1. All codes are available at https://github.com/wanxinjin/Safe-PDP.

Table 1: Experimental environments [71]

| System $\boldsymbol{\Sigma}(\boldsymbol{\theta})$ | Dynamics $\boldsymbol{f}(\boldsymbol{\theta}_{\text{dyn}})$ | Control cost $J(\boldsymbol{\theta}_{\text{obj}})$ | Constraints $\boldsymbol{g}(\boldsymbol{\theta}_{\text{cstr}})$ |
|---|---|---|---|
| Cartpole | cart & pole masses and length | $c_t = \|\boldsymbol{u}\|_2^2 +$ | $\boldsymbol{g}_x(\boldsymbol{x}) \leq X_{\max}$, |
| Two-link Robot arm | length and mass of links | $\|\boldsymbol{\theta}'_{\text{obj}}(\boldsymbol{x}-\boldsymbol{x}_{\text{goal}})\|_2^2$, | $\|\boldsymbol{u}\|_{2/\infty} \leq U_{\max}$, |
| 6-DoF quadrotor | mass, wing length, inertia | $c_T =$ | |
| 6-DoF rocket landing | rocket mass, length, inertia | $\|\boldsymbol{\theta}'_{\text{obj}}(\boldsymbol{x}-\boldsymbol{x}_{\text{goal}})\|_2^2$ | $\boldsymbol{\theta}_{\text{cstr}} = \{X_{\max},U_{\max}\}$ |

Note that for each system, $\boldsymbol{g}(\boldsymbol{\theta}_{\text{cstr}})$ includes the immediate constraints on system input $\boldsymbol{u}$ and state $\boldsymbol{x}$ at any time instance; $\boldsymbol{g}_x$ is known; $\|\cdot\|_{2/\infty}$ is the 2 or $\infty$ norm; and time horizon $T$ is around 50 for all systems.

**Problem I: Safe Policy Optimization** aims to find a policy that minimizes a control cost $J$ subject to constraints $\boldsymbol{g}$ while guaranteeing that *any intermediate policy during optimization should never violate the constraints*. To apply Safe PDP to solve such a problem for the systems in Table 1, we set:

$$\boldsymbol{\Sigma}(\boldsymbol{\theta}): \qquad \begin{aligned} \textit{dynamics:} \quad & \boldsymbol{x}_{t+1} = \boldsymbol{f}(\boldsymbol{x}_t,\boldsymbol{u}_t) \quad \text{with} \quad \boldsymbol{x}_0, \\ \textit{policy:} \quad & \boldsymbol{u}_t = \boldsymbol{\pi}_t(\boldsymbol{x}_t,\boldsymbol{\theta}), \end{aligned} \tag{8}$$

where dynamics $\boldsymbol{f}$ is learned from demonstrations in Problem III, and $\boldsymbol{\pi}(\boldsymbol{\theta})$ is represented by a (deep) feedforward neural network (NN) with $\boldsymbol{\theta}$ the NN parameter. In Problem P, the task loss $\ell(\boldsymbol{\xi}_{\boldsymbol{\theta}},\boldsymbol{\theta})$ is set as $J(\boldsymbol{\theta}_{\text{obj}})$, and task constraints $R_i(\boldsymbol{\xi}_{\boldsymbol{\theta}},\boldsymbol{\theta})$ as $\boldsymbol{g}(\boldsymbol{\theta}_{\text{cstr}})$, with both $\boldsymbol{\theta}_{\text{obj}}$ and $\boldsymbol{\theta}_{\text{cstr}}$ known. Then, safe policy optimization is to solve Problem P using Safe PDP. The results for the robot arm and 6-DoF maneuvering quadrotor are in Fig. 2, and the other results and details are in Appendix F.1.

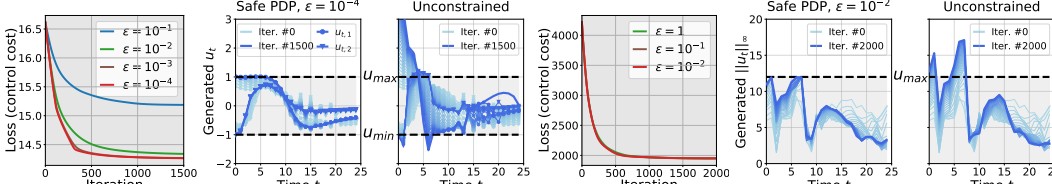

(a) Robot-arm loss  (b) Constraint violation during opt.  (c) Quadrotor loss  (d) Constraint violation during opt.

Figure 2: Safe neural policy optimization for robot-arm (a)-(b) and 6-DoF quadrotor (c)-(d).

Fig. 2a and 2c plot loss (control cost) versus gradient-descent iteration under different $\epsilon$, showing that the NN policy achieves a good convergence when $\epsilon \leq 10^{-2}$ (as asserted by Theorem 3). Fig. 2b and 2d show all indeterminate control trajectories generated from the NN policy during entire iterations; we also mark the constraints $U_{\max}$ and compare with the unconstrained policy optimization under the same settings. The results confirm that Safe PDP enables *to achieve an optimal policy while guaranteeing that any intermediate policy throughout optimization is safe*.

**Problem II: Safe Motion Planning** searches for a dynamics-feasible trajectory that optimizes a criterion and avoids unsafe regions (obstacles), meanwhile guaranteeing that any intermediate motion trajectory during search must avoid the unsafe regions. To apply Safe PDP to solve such problem, we specialize $\Sigma(\theta)$ as (8) except that policy here is $u_t = u(t, \theta)$, which is represented by Lagrangian polynomial [83] with $\theta$ the parameters (pivots). In Problem P, task loss is set as $J(\theta_{\text{obj}})$, and task constraints as $g(\theta_{\text{cstr}})$, with $\theta_{\text{obj}}$ and $\theta_{\text{cstr}}$ known in Table 1. The safe planning results using Safe PDP for cartpole and 6-DoF rocket landing are in Fig. 2, in comparison with ALTRO, a state-of-the-art constrained trajectory optimization method [21]. Other results and more details are in Appendix F.2.

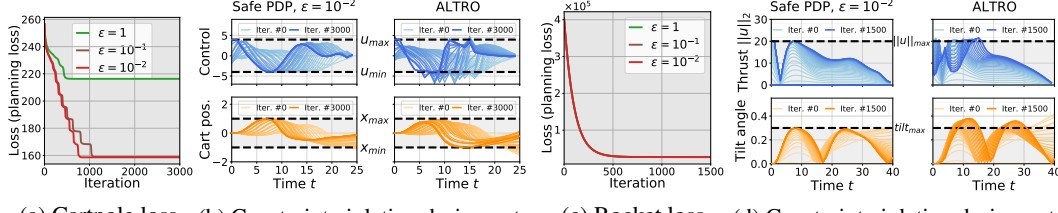

(a) Cartpole loss  (b) Constraint violation during opt.  (c) Rocket loss  (d) Constraint violation during opt.

Figure 3: Safe motion planning for cartpole (a)-(b) and 6-DoF rocket powered landing (c)-(d).

Fig. 3a and 3c plot the task loss versus gradient-descent iteration, showing that the trajectory achieves a good convergence with $\epsilon \leq 10^{-2}$. Fig. 3b and 3d show all intermediate motion trajectories during entire optimization, with constraints marked. The results confirm that Safe PDP can find an optimal trajectory while always respecting constraints throughout planning process.

**Problem III: Learning MPC from Demonstrations.** Suppose for all systems in Table 1, the control cost $J(\theta_{\text{cost}})$, dynamics $f(\theta_{\text{dyn}})$ and constraints $g_t(\theta_{\text{cstr}})$ are all unknown and parameterized as in Table 1. We aim to jointly learn $\theta = \{\theta_{\text{cost}}, \theta_{\text{dyn}}, \theta_{\text{cstr}}\}$ from demonstrations $\xi^{\text{demo}} = \{x_{0:T}^{\text{demo}}, u_{0:T-1}^{\text{demo}}\}$ of a true expert system. In Problem P, set $\Sigma(\theta)$ as (1), consisting of $J(\theta_{\text{cost}})$, $f(\theta_{\text{dyn}})$, and $g_t(\theta_{\text{cstr}})$ parameterized; set task loss $\ell(\xi_\theta, \theta) = \|\xi^{\text{demo}} - \xi_\theta\|^2$, which quantifies the *reproducing loss* between $\xi^{\text{demo}}$ and $\xi_\theta$; and there is no task constraints. By solving Problem P, we can learn $\Sigma(\theta)$ such that its reproduced $\xi_\theta$ is closest to given $\xi^{\text{demo}}$. The demonstrations $\xi^{\text{demo}}$ here are generated with $\theta$ known (two episode trajectories for each system with time horizon $T=50$). The plots of the loss versus gradient-descent iteration are in Fig. 4, and more details and results are in Appendix F.3.

In Fig. 4a-4d, for each system, we use three strategies to obtain $\xi_\theta$ and $\frac{\partial \xi_\theta}{\partial \theta}$ for $\Sigma(\theta)$: (A) use a solver [77] to obtain $\xi_\theta$ and use Theorem 1 to obtain $\frac{\partial \xi_\theta}{\partial \theta}$; (B) use Theorem 2 to approximate both $\xi_\theta$ and $\frac{\partial \xi_\theta}{\partial \theta}$ by $\xi_{(\theta,\gamma)}$ and $\frac{\partial \xi_{(\theta,\gamma)}}{\partial \theta}$, respectively, $\gamma = 10^{-2}$; and (C) use a solver to obtain $\xi_\theta$ and Theorem 2 only for $\frac{\partial \xi_\theta}{\partial \theta}$. Fig. 4a-4d show that for Strategies (B) and (C), the reproducing loss quickly converges to zeros, indicating that the dynamics, constraints, and control cost are successfully learned to reproduce the demonstrations. Fig. 4a-4d also show numerical instability for strategy (A); this is due to the discontinuous 'switching' of active inequalities between iterations, and also the error in correctly identifying active inequalities (we identify them by checking $g_{t,i} > -\delta$ with $\delta > 0$ a small

threshold), as analyzed in Section 5. More analysis is given in Appendix F.3. Note that we are not aware of any existing methods that can handle jointly learning of cost, dynamics, and constraints here, and thus we have not given benchmark comparison. Fig. 4e gives timing results of Safe PDP.

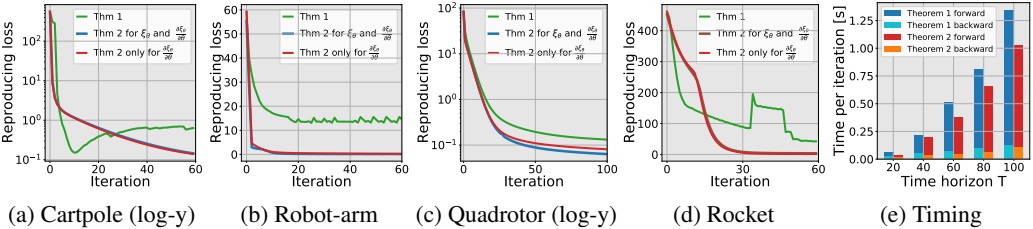

(a) Cartpole (log-y) (b) Robot-arm (c) Quadrotor (log-y) (d) Rocket (e) Timing

Figure 4: Jointly learning dynamics, constraints, and control cost from demonstrations.

## 8 Discussion

**Comparisons with other differentiable frameworks.** Fig. 5 compares Safe PDP, CasADi [70], and Differentiable MPC [68] for the computational efficiency of differentiating an optimal trajectory of a constrained optimal control system with different control horizons $T$. The results show a significantly computational advantage of Safe PDP over CasADi and Differentiable MPC. Specifically, Safe PDP has a complexity of $\mathcal{O}(T)$, while CasADi and Differentiable MPC have at least $\mathcal{O}(T^2)$. This is because both CasADi and differentiable MPC are based on the implicit function theorem [75] and need to compute the inverse of a Hessian matrix of the size proportional to $T \times T$. In contrast, Safe PDP solves the gradient of a trajectory by constructing an Auxiliary Control System, which can be solved using the Riccati equation.

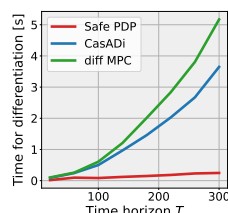

Figure 5: Time for differentiating optimal trajectory with different $T$.

**Limitation of Safe PDP.** Safe PDP requires a safe (feasible) initialization such that the log-barrier control cost or loss is well-defined. While restrictive, safe initialization is common in safe learning [63, 84]. We have the following empiricism on how to provide safe initializations for different types of problems, as adopted in our experiments in Section 7. In safe policy optimization, one could first use supervised learning to learn a safe policy from some safe trajectories/demonstrations (not necessarily be optimal) and then use the learned safe policy to initialize Safe PDP. In safe motion planning, one could arbitrarily provide a safe trajectory (not necessarily optimal) to initialize Safe PDP. In learning MPCs, the goal includes learning of constraint itself, and there is no such requirement.

**Strategies to accelerate forward pass of Safe PDP.** There are many strategies to accelerate a long-horizon trajectory optimization (optimal control) in the forward pass of Safe PDP. (I) One effective way is to scale the (continuous) long-horizon problem into a smaller one (e.g., a unit) by applying a time-warping function to the dynamics and cost function [85]. After solving the scaled short-horizon problem, re-scale the trajectory back. (II) There are also 'warm-up' tricks, e.g., one can initialize the trajectory at the next iteration using the result of the previous iteration. (III) One can also use a hierarchical strategy to solve trajectory optimization from coarse to fine resolutions. We have tested and provided the comparison for the above three acceleration strategies in Appendix G.2.

Please refer to Appendix G for more discussion, which includes G.1: comparison between Safe-PDP and non-safe PDP; G.2: comparison of different strategies for accelerating long-horizon trajectory optimization; G.3: trade-offs between accuracy and computational efficiency using barrier penalties; G.4: learning MPCs from non-optimal data; and G.5: detailed discussion on limitation of Safe PDP.

## 9 Conclusions

This paper proposes a Safe Pontryagin Differentiable Programming methodology, which establishes a provable and systematic safe differentiable framework to solve a broad class of safety-critical control and learning tasks with different types of safety constraints. For a constrained system and task, Safe PDP approximates both the solution and its gradient in backward pass by solving their more efficient unconstrained counterparts. Safe PDP has established two results: one is the *controlled accuracy guarantee* for approximations of the solution and its gradient, and the other is the *safety guarantee* for constraint satisfaction throughout the control and learning process. We envision the potential of Safe PDP for addressing various safety-critical problems in machine learning, control, and robotics fields.

## Acknowledgments and Disclosure of Funding

This work is supported by the NASA University Leadership Initiative (ULI) under grant number 80NSSC20M0161. The research of Prof. George J. Pappas is supported by the AFOSR Assured Autonomy in Congested Environments under grant number FA9550-19-1-0265. This work has been done primarily in the last semester of Wanxin Jin's Ph.D. study at Purdue University. Wanxin Jin thanks Prof. Zhaoran Wang for some discussion about this work.

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
