Appendix to the Safe Pontryagin Differentiable Programming paper

# A    Second-order Sufficient Condition

Before presenting the second-order condition for the optimal control Problem B($\boldsymbol{\theta}$), we present the second-order condition for a general constrained nonlinear programming. The interested reader can find the details in Theorem 4 in [48].

**Lemma A.1** (Second-order sufficient condition [48]). *If all functions defining a constrained optimization*

$$
\begin{aligned}
\min_{\boldsymbol{x}} \quad & f(\boldsymbol{x}) \\
\text{subject to} \quad & g_i(\boldsymbol{x}) \leq 0 \quad i = 1, 2, \cdots, m, \\
& h_j(\boldsymbol{x}) = 0 \quad j = 1, 2, \cdots, p,
\end{aligned}
\tag{S.1}
$$

*are twice-continuous differentiable, the second-order sufficient condition for $\boldsymbol{x}^*$ to be a local isolated minimizing solution to (S.1) is that there exist vectors $\boldsymbol{v}^*$ and $\boldsymbol{w}^*$ such that $(\boldsymbol{x}^*, \boldsymbol{v}^*, \boldsymbol{w}^*)$ satisfies*

$$
\begin{aligned}
g_i(\boldsymbol{x}^*) &\leq 0, \quad i = 1, 2, \cdots, m, \\
h_i(\boldsymbol{x}^*) &= 0, \quad j = 1, 2, \cdots, p, \\
v_i g_i(\boldsymbol{x}^*) &= 0, \quad i = 1, 2, \cdots, m, \\
u_i &\geq 0, \quad i = 1, 2, \cdots, m, \\
\nabla L(\boldsymbol{x}^*, \boldsymbol{v}^*, \boldsymbol{w}^*) &= 0,
\end{aligned}
\tag{S.2}
$$

*with*

$$
L(\boldsymbol{x}, \boldsymbol{v}, \boldsymbol{w}) = f(\boldsymbol{x}) + \sum_{i=1}^{m} v_i g_i(\boldsymbol{x}) + \sum_{i=1}^{p} w_i h_i(\boldsymbol{x}),
\tag{S.3}
$$

*and $\nabla L$ being the derivative of $L$ with respect to $\boldsymbol{x}$; and further for any nonzero $\boldsymbol{y} \neq \boldsymbol{0}$ satisfying $\boldsymbol{y}' \nabla g_i(\boldsymbol{x}^*) = 0$ for all $i$ with $v_i^* > 0$, $\boldsymbol{y}' \nabla g_i(\boldsymbol{x}^*) \leq 0$ for all $i$ with $v_i^* \geq 0$, and $\boldsymbol{y}' \nabla h_j(\boldsymbol{x}^*) = 0$ for all $j = 1, 2, \cdots, p$, it follows that*

$$
\boldsymbol{y}' \nabla^2 L(\boldsymbol{x}^*, \boldsymbol{v}^*, \boldsymbol{w}^*) \boldsymbol{y} > 0.
\tag{S.4}
$$

The above second-order sufficient condition for nonlinear programming is well-known. The proof for Lemma A.1 can be found in Theorem 4 in [48]. Similarly, we can establish the second-order sufficient condition for a general constrained optimal control system $\boldsymbol{\Sigma}(\boldsymbol{\theta})$ in (1), as below.

**Lemma A.2** (Second-order sufficient condition for $\boldsymbol{\Sigma}(\boldsymbol{\theta})$ to have a local isolated minimizing trajectory $\boldsymbol{\xi}_{\boldsymbol{\theta}}$ [73]). *Given $\boldsymbol{\theta}$, if all functions defining the constrained optimal control system $\boldsymbol{\Sigma}(\boldsymbol{\theta})$ are twice continuously differentiable in a neighborhood (tube) of $\boldsymbol{\xi}_{\boldsymbol{\theta}} = \{\boldsymbol{x}_{0:T}^{\boldsymbol{\theta}}, \boldsymbol{u}_{0:T-1}^{\boldsymbol{\theta}}\}$, $\boldsymbol{\xi}_{\boldsymbol{\theta}}$ is a local isolated minimizing trajectory to Problem B($\boldsymbol{\theta}$) if there exist sequences $\boldsymbol{\lambda}_{1:T}^{\boldsymbol{\theta}}$, $\boldsymbol{v}_{0:T}^{\boldsymbol{\theta}}$, and $\boldsymbol{w}_{0:T}^{\boldsymbol{\theta}}$ such that the following Constrained Pontryagin Maximum/Minimum Principle (C-PMP) conditions hold,*

$$
\begin{aligned}
\boldsymbol{x}_{t+1}^{\boldsymbol{\theta}} &= \boldsymbol{f}(\boldsymbol{x}_t^{\boldsymbol{\theta}}, \boldsymbol{u}_t^{\boldsymbol{\theta}}, \boldsymbol{\theta}) \quad \text{and} \quad \boldsymbol{x}_0^{\boldsymbol{\theta}} = \boldsymbol{x}_0(\boldsymbol{\theta}), \\
\boldsymbol{\lambda}_t^{\boldsymbol{\theta}} &= L_t^x(\boldsymbol{x}_t^{\boldsymbol{\theta}}, \boldsymbol{u}_t^{\boldsymbol{\theta}}, \boldsymbol{\lambda}_{t+1}^{\boldsymbol{\theta}}, \boldsymbol{v}_t^{\boldsymbol{\theta}}, \boldsymbol{w}_t^{\boldsymbol{\theta}}, \boldsymbol{\theta}) \quad \text{and} \quad \boldsymbol{\lambda}_T^{\boldsymbol{\theta}} = L_T^x(\boldsymbol{x}_T^{\boldsymbol{\theta}}, \boldsymbol{v}_t^{\boldsymbol{\theta}}, \boldsymbol{w}_t^{\boldsymbol{\theta}}, \boldsymbol{\theta}), \\
\boldsymbol{0} &= L_t^u(\boldsymbol{x}_t^{\boldsymbol{\theta}}, \boldsymbol{u}_t^{\boldsymbol{\theta}}, \boldsymbol{\lambda}_{t+1}^{\boldsymbol{\theta}}, \boldsymbol{v}_t^{\boldsymbol{\theta}}, \boldsymbol{w}_t^{\boldsymbol{\theta}}, \boldsymbol{\theta}), \\
h_{t,j}(\boldsymbol{x}_t^{\boldsymbol{\theta}}, \boldsymbol{u}_t^{\boldsymbol{\theta}}, \boldsymbol{\theta}) &= 0, \quad j = 1, 2, \cdots, s_t, \\
h_{T,j}(\boldsymbol{x}_T^{\boldsymbol{\theta}}, \boldsymbol{\theta}) &= 0, \quad j = 1, 2, \cdots, s_T, \\
g_{t,i}(\boldsymbol{x}_t^{\boldsymbol{\theta}}, \boldsymbol{u}_t^{\boldsymbol{\theta}}, \boldsymbol{\theta}) &\leq 0, \quad v_{t,i} g_{t,i}(\boldsymbol{x}_t^{\boldsymbol{\theta}}, \boldsymbol{u}_t^{\boldsymbol{\theta}}, \boldsymbol{\theta}) = 0, \quad v_{t,i} \geq 0, \quad i = 1, 2, \cdots, q_t, \\
g_{T,i}(\boldsymbol{x}_T^{\boldsymbol{\theta}}, \boldsymbol{\theta}) &\leq 0, \quad v_{T,i} g_{T,i}(\boldsymbol{x}_T^{\boldsymbol{\theta}}, \boldsymbol{\theta}) = 0, \quad v_{T,i} \geq 0, \quad i = 1, 2, \cdots, q_T,
\end{aligned}
\tag{S.5}
$$

*and further if*

$$
\sum_{t=0}^{T-1} \begin{bmatrix} \boldsymbol{x}_t \\ \boldsymbol{u}_t \end{bmatrix}' \begin{bmatrix} L_t^{xx} & L_t^{xu} \\ L_t^{ux} & L_t^{uu} \end{bmatrix} \begin{bmatrix} \boldsymbol{x}_t \\ \boldsymbol{u}_t \end{bmatrix} + \boldsymbol{x}_T' L_T^{xx} \boldsymbol{x}_T > 0
\tag{S.6}
$$

*for any non-zero trajectory $\{\boldsymbol{x}_{0:T}, \boldsymbol{u}_{0:T-1}\} \neq \boldsymbol{0}$ satisfying*

$$
\begin{aligned}
&\boldsymbol{x}_{t+1} = F_t^x \boldsymbol{x}_t + F_t^u \boldsymbol{u}_t \qquad \boldsymbol{x}_0 = \boldsymbol{0}, \\
&H_t^x \boldsymbol{x}_t + H_t^u \boldsymbol{u}_t = \boldsymbol{0} \quad and \quad H_T^x \boldsymbol{x}_T = \boldsymbol{0}, \\
&\check{G}_t^x \boldsymbol{x}_t + \check{G}_t^u \boldsymbol{u}_t = \boldsymbol{0} \quad and \quad \check{G}_T^x \boldsymbol{x}_T = \boldsymbol{0}, \\
&\bar{G}_t^x \boldsymbol{x}_t + \bar{G}_t^u \boldsymbol{u}_t \leq \boldsymbol{0} \quad and \quad \bar{G}_T^x \boldsymbol{x}_T \leq \boldsymbol{0}.
\end{aligned}
\tag{S.7}
$$

*Here, $t = 0, 1, ..., T-1$; $L_t^x$ is the first-order derivative of the Hamiltonian $L_t$ in (2) with respect to $\boldsymbol{x}$, and $L_t^{xx}$ is the second-derivative of $L_t$ with respect to $\boldsymbol{x}$, and similar notation convention applies to $L_T^x$, $L_t^u$, $L_t^{xu} = (L_t^{ux})'$, and $L_t^{uu}$; $H_t^x$ is the first-order derivative of $\boldsymbol{h}_t$ with respect to $\boldsymbol{x}$ and the similar convention applies to $H_T^x$, $H_t^u$, $F_t^x$ and $F_t^u$ for $\boldsymbol{f}$, $\check{G}_t^x$ and $\check{G}_t^u$ for $\check{\boldsymbol{g}}_t$, $\check{G}_T^x$ for $\check{\boldsymbol{g}}_T$, $\bar{G}_t^x$ and $\bar{G}_t^u$ for $\bar{\boldsymbol{g}}_t$, $\bar{G}_T^x$ for $\bar{\boldsymbol{g}}_T$, where*

$$
\begin{aligned}
\check{\boldsymbol{g}}_t(\boldsymbol{x}_t, \boldsymbol{u}_t, \boldsymbol{\theta}) &= \mathrm{col}\{g_{t,i}(\boldsymbol{x}_t, \boldsymbol{u}_t, \boldsymbol{\theta}) \,|\, v_{t,i}^{\boldsymbol{\theta}} > 0, i = 1, ..., q_t\}, \\
\check{\boldsymbol{g}}_T(\boldsymbol{x}_T, \boldsymbol{\theta}) &= \mathrm{col}\{g_{T,i}(\boldsymbol{x}_T, \boldsymbol{\theta}) \,|\, v_{t,i}^{\boldsymbol{\theta}} > 0, i = 1, ..., q_T\}, \\
\bar{\boldsymbol{g}}_t(\boldsymbol{x}_t, \boldsymbol{u}_t, \boldsymbol{\theta}) &= \mathrm{col}\{g_{t,i}(\boldsymbol{x}_t, \boldsymbol{u}_t, \boldsymbol{\theta}) \,|\, g_{t,i}(\boldsymbol{x}_t^{\boldsymbol{\theta}}, \boldsymbol{u}_t^{\boldsymbol{\theta}}, \boldsymbol{\theta}) = 0, i = 1, ..., q_t\} \in \mathbb{R}^{\bar{q}_t}, \\
\bar{\boldsymbol{g}}_T(\boldsymbol{x}_T, \boldsymbol{\theta}) &= \mathrm{col}\{g_{T,i}(\boldsymbol{x}_T, \boldsymbol{\theta}) \,|\, g_{T,i}(\boldsymbol{x}_T^{\boldsymbol{\theta}}, \boldsymbol{\theta}) = 0, i = 1, ..., q_T\} \in \mathbb{R}^{\bar{q}_T},
\end{aligned}
\tag{S.8}
$$

*i.e., $\bar{\boldsymbol{g}}_t$ and $\bar{\boldsymbol{g}}_T$ are the vector functions formed by stacking all active inequality constraints at $\boldsymbol{\xi_\theta}$. All the above first- and second-order derivatives are evaluated at $(\boldsymbol{\xi_\theta}, \boldsymbol{\lambda}_{1:T}^{\boldsymbol{\theta}}, \boldsymbol{v}_{0:T}^{\boldsymbol{\theta}}, \boldsymbol{w}_{0:T}^{\boldsymbol{\theta}})$.*

The above second-order sufficient condition for the constrained optimal control system $\Sigma(\boldsymbol{\theta})$ is well-known and has been well-established since [73]. The conditions in (S.5) is referred to as discrete-time Constrained Pontryagin Maximum/Minimum Principle (C-PMP) [73]. Note that in the case of strict complementarity, one has $\check{\boldsymbol{g}}_t(\boldsymbol{x}_t, \boldsymbol{u}_t, \boldsymbol{\theta}) = \bar{\boldsymbol{g}}_t(\boldsymbol{x}_t, \boldsymbol{u}_t, \boldsymbol{\theta})$ and $\check{\boldsymbol{g}}_T(\boldsymbol{x}_T, \boldsymbol{\theta}) = \bar{\boldsymbol{g}}_T(\boldsymbol{x}_T, \boldsymbol{\theta})$ in (S.8).

## B Proof of Theorem 1

To prove Theorem 1, in the first part, we need to derive the Differential Constrained Pontryagin Maximum/Minimum Principle (Differential C-PMP), which $\frac{\partial \boldsymbol{\xi_\theta}}{\partial \boldsymbol{\theta}} = \{X_{0:T}^{\boldsymbol{\theta}}, U_{0:T}^{\boldsymbol{\theta}}\}$ must satisfy. Then, in the second part, we formally present the proof for Theorem 1.

### B.1 Differential Constrained Pontryagin Maximum/Minimum Principle

From Lemma 1, for the constrained optimal control system $\Sigma(\boldsymbol{\theta})$ with any $\boldsymbol{\theta}$ in a neighborhood of $\bar{\boldsymbol{\theta}}$, $(\boldsymbol{\xi_\theta}, \boldsymbol{\lambda}_{1:T}^{\boldsymbol{\theta}}, \boldsymbol{v}_{0:T}^{\boldsymbol{\theta}}, \boldsymbol{w}_{0:T}^{\boldsymbol{\theta}})$ satisfies the C-PMP conditions in (S.5). Since $(\boldsymbol{\xi_\theta}, \boldsymbol{\lambda}_{1:T}^{\boldsymbol{\theta}}, \boldsymbol{v}_{0:T}^{\boldsymbol{\theta}}, \boldsymbol{w}_{0:T}^{\boldsymbol{\theta}})$ is also once-continuously differentiable with respect to $\boldsymbol{\theta}$ from Lemma 1, one can differentiate the C-PMP conditions in (S.5) on both sides with respect to $\boldsymbol{\theta}$, as below.

Differentiating the first five lines in (S.5) is straightforward, yielding

$$
\begin{aligned}
\frac{\partial \boldsymbol{x}_{t+1}^{\boldsymbol{\theta}}}{\partial \boldsymbol{\theta}} &= F_t^x \frac{\partial \boldsymbol{x}_t^{\boldsymbol{\theta}}}{\partial \boldsymbol{\theta}} + F_t^u \frac{\partial \boldsymbol{u}_t^{\boldsymbol{\theta}}}{\partial \boldsymbol{\theta}} + F_t^\theta \quad and \quad X_0^{\boldsymbol{\theta}} = \frac{\partial \boldsymbol{x}_0^{\boldsymbol{\theta}}}{\partial \boldsymbol{\theta}} = \frac{\partial \boldsymbol{x}_0(\boldsymbol{\theta})}{\partial \boldsymbol{\theta}}, \\
\frac{\partial \boldsymbol{\lambda}_t^{\boldsymbol{\theta}}}{\partial \boldsymbol{\theta}} &= L_t^{xx} \frac{\partial \boldsymbol{x}_t^{\boldsymbol{\theta}}}{\partial \boldsymbol{\theta}} + L_t^{xu} \frac{\partial \boldsymbol{u}_t^{\boldsymbol{\theta}}}{\partial \boldsymbol{\theta}} + (F_t^x)' \frac{\partial \boldsymbol{\lambda}_{t+1}^{\boldsymbol{\theta}}}{\partial \boldsymbol{\theta}} + (G_t^x)' \frac{\partial \boldsymbol{v}_t^{\boldsymbol{\theta}}}{\partial \boldsymbol{\theta}} + (H_t^x)' \frac{\partial \boldsymbol{w}_t^{\boldsymbol{\theta}}}{\partial \boldsymbol{\theta}} + L_t^{x\theta} \quad and \\
\frac{\partial \boldsymbol{\lambda}_T^{\boldsymbol{\theta}}}{\partial \boldsymbol{\theta}} &= L_T^{xx} \frac{\partial \boldsymbol{x}_T^{\boldsymbol{\theta}}}{\partial \boldsymbol{\theta}} + (G_T^x)' \frac{\partial \boldsymbol{v}_T^{\boldsymbol{\theta}}}{\partial \boldsymbol{\theta}} + (H_T^x)' \frac{\partial \boldsymbol{w}_T^{\boldsymbol{\theta}}}{\partial \boldsymbol{\theta}} + L_T^{x\theta}, \\
\boldsymbol{0} &= L_t^{ux} \frac{\partial \boldsymbol{x}_t^{\boldsymbol{\theta}}}{\partial \boldsymbol{\theta}} + L_t^{uu} \frac{\partial \boldsymbol{u}_t^{\boldsymbol{\theta}}}{\partial \boldsymbol{\theta}} + (F_t^u)' \frac{\partial \boldsymbol{\lambda}_{t+1}^{\boldsymbol{\theta}}}{\partial \boldsymbol{\theta}} + (G_t^u)' \frac{\partial \boldsymbol{v}_t^{\boldsymbol{\theta}}}{\partial \boldsymbol{\theta}} + (H_t^u)' \frac{\partial \boldsymbol{w}_t^{\boldsymbol{\theta}}}{\partial \boldsymbol{\theta}} + L_t^{u\theta}, \\
H_t^x \frac{\partial \boldsymbol{x}_t^{\boldsymbol{\theta}}}{\partial \boldsymbol{\theta}} &+ H_t^u \frac{\partial \boldsymbol{u}_t^{\boldsymbol{\theta}}}{\partial \boldsymbol{\theta}} + H_t^\theta = \boldsymbol{0} \quad and \quad H_T^x \frac{\partial \boldsymbol{x}_t^{\boldsymbol{\theta}}}{\partial \boldsymbol{\theta}} + H_T^\theta = \boldsymbol{0}.
\end{aligned}
\tag{S.9}
$$

We now consider to differentiate the two last equations (i.e., complementarity conditions) in the last two lines in (S.5). We start with

$$
v_{t,i}^{\boldsymbol{\theta}} \, g_{t,i}(\boldsymbol{x}_t^{\boldsymbol{\theta}}, \boldsymbol{u}_t^{\boldsymbol{\theta}}, \boldsymbol{\theta}) = 0, \qquad i = 1, 2, \cdots, q_t.
\tag{S.10}
$$

Differentiating the above (S.10) on both sides with respect to $\boldsymbol{\theta}$ yields

$$\frac{\partial v_{t,i}^{\boldsymbol{\theta}}}{\partial \boldsymbol{\theta}} \, g_{t,i}(\boldsymbol{x}_t^{\boldsymbol{\theta}}, \boldsymbol{u}_t^{\boldsymbol{\theta}}, \boldsymbol{\theta}) + \mu_{t,i}^{\boldsymbol{\theta}} \, \frac{\partial g_{t,i}(\boldsymbol{x}_t^{\boldsymbol{\theta}}, \boldsymbol{u}_t^{\boldsymbol{\theta}}, \boldsymbol{\theta})}{\partial \boldsymbol{\theta}} = \boldsymbol{0}, \quad i = 1, 2, \cdots, q_t. \tag{S.11}$$

In the above, we consider two following cases. If $g_{t,i}(\boldsymbol{x}_t^{\boldsymbol{\theta}}, \boldsymbol{u}_t^{\boldsymbol{\theta}}, \boldsymbol{\theta}) = 0$, i.e., $g_{t,i}$ is an active inequality constraint, then, $\mu_{t,i}^{\boldsymbol{\theta}} > 0$ according to strict complementarity (condition (iii) in Lemma 1). From (S.11), one thus has

$$\frac{\partial g_{t,i}(\boldsymbol{x}_t^{\boldsymbol{\theta}}, \boldsymbol{u}_t^{\boldsymbol{\theta}}, \boldsymbol{\theta})}{\partial \boldsymbol{\theta}} = \boldsymbol{0}. \tag{S.12}$$

If $g_{t,i}(\boldsymbol{x}_t^{\boldsymbol{\theta}}, \boldsymbol{u}_t^{\boldsymbol{\theta}}, \boldsymbol{\theta}) < 0$, i.e., $g_{t,i}$ is an inactive constraint, then $v_{t,i}^{\boldsymbol{\theta}} = 0$ and one has

$$\frac{\partial v_{t,i}^{\boldsymbol{\theta}}}{\partial \boldsymbol{\theta}} = \boldsymbol{0} \quad \text{for} \quad v_{t,i}^{\boldsymbol{\theta}} = 0. \tag{S.13}$$

Stacking (S.12) for all active inequality constraints defined in (S.8) will lead to

$$\boldsymbol{0} = \bar{G}_t^x \frac{\partial \boldsymbol{x}_t^{\boldsymbol{\theta}}}{\partial \boldsymbol{\theta}} + \bar{G}_t^u \frac{\partial \boldsymbol{u}_t^{\boldsymbol{\theta}}}{\partial \boldsymbol{\theta}} + \bar{G}_t^{\theta}. \tag{S.14}$$

Similarly, we can show that differentiating $v_{T,i} g_{T,i}(\boldsymbol{x}_T^{\boldsymbol{\theta}}, \boldsymbol{\theta}) = 0, i = 1, 2, \cdots, q_T$, will lead to

$$\bar{G}_T^x \frac{\partial \boldsymbol{x}_T^{\boldsymbol{\theta}}}{\partial \boldsymbol{\theta}} + \bar{G}_T^{\theta} = \boldsymbol{0}. \tag{S.15}$$

If we further define

$$\bar{\boldsymbol{v}}_t^{\boldsymbol{\theta}} = \text{col}\{ v_{t,i}^{\boldsymbol{\theta}} \mid v_{t,i}^{\boldsymbol{\theta}} > 0, i = 1, ..., q_t \} \in \mathbb{R}^{\bar{q}_t}, \tag{S.16}$$

then, due to (S.13), the following terms in the second, third, and fourth lines in (S.9) can be written in an equivalent way:

$$(G_t^x)' \frac{\partial \boldsymbol{v}_t^{\boldsymbol{\theta}}}{\partial \boldsymbol{\theta}} = (\bar{G}_t^x)' \frac{\partial \bar{\boldsymbol{v}}_t^{\boldsymbol{\theta}}}{\partial \boldsymbol{\theta}} \quad \text{and} \quad (G_t^u)' \frac{\partial \boldsymbol{v}_t^{\boldsymbol{\theta}}}{\partial \boldsymbol{\theta}} = (\bar{G}_t^u)' \frac{\partial \bar{\boldsymbol{v}}_t^{\boldsymbol{\theta}}}{\partial \boldsymbol{\theta}}. \tag{S.17}$$

In sum, combining (S.9), (S.14), (S.15), and (S.17), one can finally write the Differential C-PMP:

$$
\begin{aligned}
&\frac{\partial \boldsymbol{x}_{t+1}^{\boldsymbol{\theta}}}{\partial \boldsymbol{\theta}} = F_t^x \frac{\partial \boldsymbol{x}_t^{\boldsymbol{\theta}}}{\partial \boldsymbol{\theta}} + F_t^u \frac{\partial \boldsymbol{u}_t^{\boldsymbol{\theta}}}{\partial \boldsymbol{\theta}} + F_t^{\theta} \quad \text{and} \quad X_0^{\boldsymbol{\theta}} = \frac{\partial \boldsymbol{x}_0^{\boldsymbol{\theta}}}{\partial \boldsymbol{\theta}}, \\
&\frac{\partial \boldsymbol{\lambda}_t^{\boldsymbol{\theta}}}{\partial \boldsymbol{\theta}} = L_t^{xx} \frac{\partial \boldsymbol{x}_t^{\boldsymbol{\theta}}}{\partial \boldsymbol{\theta}} + L_t^{xu} \frac{\partial \boldsymbol{u}_t^{\boldsymbol{\theta}}}{\partial \boldsymbol{\theta}} + (F_t^x)' \frac{\partial \boldsymbol{\lambda}_{t+1}^{\boldsymbol{\theta}}}{\partial \boldsymbol{\theta}} + (\bar{G}_t^x)' \frac{\partial \bar{\boldsymbol{v}}_t^{\boldsymbol{\theta}}}{\partial \boldsymbol{\theta}} + (H_t^x)' \frac{\partial \boldsymbol{w}_t^{\boldsymbol{\theta}}}{\partial \boldsymbol{\theta}} + L_t^{x\theta} \quad \text{and} \\
&\frac{\partial \boldsymbol{\lambda}_T^{\boldsymbol{\theta}}}{\partial \boldsymbol{\theta}} = L_T^{xx} \frac{\partial \boldsymbol{x}_T^{\boldsymbol{\theta}}}{\partial \boldsymbol{\theta}} + (\bar{G}_T^x)' \frac{\partial \bar{\boldsymbol{v}}_T^{\boldsymbol{\theta}}}{\partial \boldsymbol{\theta}} + (H_T^x)' \frac{\partial \boldsymbol{w}_T^{\boldsymbol{\theta}}}{\partial \boldsymbol{\theta}} + L_T^{x\theta}, \\
&\boldsymbol{0} = L_t^{ux} \frac{\partial \boldsymbol{x}_t^{\boldsymbol{\theta}}}{\partial \boldsymbol{\theta}} + L_t^{uu} \frac{\partial \boldsymbol{u}_t^{\boldsymbol{\theta}}}{\partial \boldsymbol{\theta}} + (F_t^u)' \frac{\partial \boldsymbol{\lambda}_{t+1}^{\boldsymbol{\theta}}}{\partial \boldsymbol{\theta}} + (\bar{G}_t^u)' \frac{\partial \bar{\boldsymbol{v}}_t^{\boldsymbol{\theta}}}{\partial \boldsymbol{\theta}} + (H_t^u)' \frac{\partial \boldsymbol{w}_t^{\boldsymbol{\theta}}}{\partial \boldsymbol{\theta}} + L_t^{u\theta}, \\
&H_t^x \frac{\partial \boldsymbol{x}_t^{\boldsymbol{\theta}}}{\partial \boldsymbol{\theta}} + H_t^u \frac{\partial \boldsymbol{u}_t^{\boldsymbol{\theta}}}{\partial \boldsymbol{\theta}} + H_t^{\theta} = \boldsymbol{0} \quad \text{and} \quad H_T^x \frac{\partial \boldsymbol{x}_T^{\boldsymbol{\theta}}}{\partial \boldsymbol{\theta}} + H_T^{\theta} = \boldsymbol{0}, \\
&\bar{G}_t^x \frac{\partial \boldsymbol{x}_t^{\boldsymbol{\theta}}}{\partial \boldsymbol{\theta}} + \bar{G}_t^u \frac{\partial \boldsymbol{u}_t^{\boldsymbol{\theta}}}{\partial \boldsymbol{\theta}} + \bar{G}_t^{\theta} = \boldsymbol{0} \quad \text{and} \quad \bar{G}_T^x \frac{\partial \boldsymbol{x}_T^{\boldsymbol{\theta}}}{\partial \boldsymbol{\theta}} + \bar{G}_T^{\theta} = \boldsymbol{0}.
\end{aligned}
\tag{S.18}
$$

With the above Differential C-PMP, we next prove the claims in Theorem 1.

## B.2 Proof of Theorem 1

We prove Theorem 1 by two steps. We first prove that the trajectory in (4), rewritten below,

$$\{ X_{0:T}^{\boldsymbol{\theta}}, U_{0:T-1}^{\boldsymbol{\theta}} \} \quad \text{with} \quad X_t^{\boldsymbol{\theta}} = \frac{\partial \boldsymbol{x}_t^{\boldsymbol{\theta}}}{\partial \boldsymbol{\theta}} \quad \text{and} \quad U_t^{\boldsymbol{\theta}} = \frac{\partial \boldsymbol{u}_t^{\boldsymbol{\theta}}}{\partial \boldsymbol{\theta}},$$

is the *local isolated* minimizing trajectory to the auxiliary control system $\overline{\Sigma}(\boldsymbol{\xi}_{\boldsymbol{\theta}})$ in (3); and second, we prove that such a local minimizing trajectory is also a *global* minimizing trajectory.

First, we prove that $\{X_{0:T}^{\boldsymbol{\theta}}, U_{0:T-1}^{\boldsymbol{\theta}}\}$ is a *local isolated* minimizing trajectory to $\overline{\Sigma}(\boldsymbol{\xi_\theta})$.

To show that $\{X_{0:T}^{\boldsymbol{\theta}}, U_{0:T-1}^{\boldsymbol{\theta}}\}$ is a local isolated minimizing trajectory to $\overline{\Sigma}(\boldsymbol{\xi_\theta})$, we only need to check whether it satisfies the second-order sufficient condition for the constrained optimal control system $\overline{\Sigma}(\boldsymbol{\xi_\theta})$, as stated in Lemma A.2. To that end, we define the following Hamiltonian for $\overline{\Sigma}(\boldsymbol{\xi_\theta})$:

$$
\begin{aligned}
\bar{L}_t = {} & \mathrm{Tr}\left(\frac{1}{2}\begin{bmatrix} X_t \\ U_t \end{bmatrix}'\begin{bmatrix} L_t^{xx} & L_t^{xu} \\ L_t^{ux} & L_t^{uu} \end{bmatrix}\begin{bmatrix} X_t \\ U_t \end{bmatrix} + \begin{bmatrix} L_t^{x\theta} \\ L_t^{ue} \end{bmatrix}'\begin{bmatrix} X_t \\ U_t \end{bmatrix}\right) + \mathrm{Tr}\left(\Lambda_{t+1}'(F_t^x X_t + F_t^u U_t + F_t^\theta)\right) \\
& + \mathrm{Tr}\left(\bar{V}_t'(\bar{G}_t^x X_t + \bar{G}_t^u U_t + \bar{G}_t^\theta)\right) + \mathrm{Tr}\left(W_t'(H_t^x X_t + H_t^u U_t + H_t^\theta)\right), \quad t = 0, .., T{-}1, \\
\bar{L}_T = {} & \mathrm{Tr}\left(\frac{1}{2}X_T' L_T^{xx} X_T + (L_T^{x\theta})' X_T\right) + \mathrm{Tr}\left(\bar{M}_T'(\bar{G}_T^x X_T + \bar{G}_T^\theta)\right) \\
& + \mathrm{Tr}\left(N_T'(H_T^x X_T + H_T^\theta)\right), \quad t = T.
\end{aligned}
$$

(S.19)

Here, $\Lambda_t \in \mathbb{R}^{n \times r}$, $t = 1, 2, ..., T$, denotes the costate (matrix) variables for $\overline{\Sigma}(\boldsymbol{\xi_\theta})$; $\bar{V}_t \in \mathbb{R}^{\bar{q}_t \times r}$ and $W_t \in \mathbb{R}^{s_t \times r}$, $t = 0, 1, ..., T$, are the multipliers for the constraints in $\overline{\Sigma}(\boldsymbol{\xi_\theta})$. Further define

$$
\Lambda_t^{\boldsymbol{\theta}} = \frac{\partial \boldsymbol{\lambda}_t^{\boldsymbol{\theta}}}{\partial \boldsymbol{\theta}}, \quad W_t^{\boldsymbol{\theta}} = \frac{\partial \boldsymbol{w}_t^{\boldsymbol{\theta}}}{\partial \boldsymbol{\theta}}, \quad \bar{V}_t^{\boldsymbol{\theta}} = \frac{\partial \bar{\boldsymbol{v}}_t^{\boldsymbol{\theta}}}{\partial \boldsymbol{\theta}},
$$

(S.20)

with $\bar{v}_t^{\boldsymbol{\theta}}$ in (S.16). Then, the Differential C-PMP in (S.18) is exactly the Constrained Pontryagin Minimal Principle (C-PMP) for the auxiliary control system $\overline{\Sigma}(\boldsymbol{\xi_\theta})$ because

$$
\begin{aligned}
& X_{t+1}^{\boldsymbol{\theta}} = \frac{\partial \bar{L}_t}{\partial \Lambda_{t+1}^{\boldsymbol{\theta}}} = F_t^x X_t^{\boldsymbol{\theta}} + F_t^u U_t^{\boldsymbol{\theta}} + F_t^{\boldsymbol{\theta}} \quad \text{and} \quad X_0 = X_0^{\boldsymbol{\theta}}, \\
& \Lambda_t^{\boldsymbol{\theta}} = \frac{\partial \bar{L}_t}{\partial X_t^{\boldsymbol{\theta}}} = L_t^{xx} X_t^{\boldsymbol{\theta}} + L_t^{xu} U_t^{\boldsymbol{\theta}} + L_t^{x\theta} + (F_t^x)' \Lambda_{t+1}^{\boldsymbol{\theta}} + (\bar{G}_t^x)' \bar{V}_t^{\boldsymbol{\theta}} + (H_t^x)' W_t^{\boldsymbol{\theta}} \quad \text{and} \\
& \Lambda_T^{\boldsymbol{\theta}} = \frac{\partial \bar{L}_t}{\partial X_T^{\boldsymbol{\theta}}} = H_T^{xx} X_T^{\boldsymbol{\theta}} + H_T^{xe} + (\bar{G}_T^x)' \bar{V}_T^{\boldsymbol{\theta}} + (H_T^x)' W_T^{\boldsymbol{\theta}}, \\
& \boldsymbol{0} = \frac{\partial \bar{L}_t}{\partial U_t^{\boldsymbol{\theta}}} = L_t^{uu} U_t^{\boldsymbol{\theta}} + L_t^{ux} X_t^{\boldsymbol{\theta}} + L_t^{u\theta} + (F_t^u)' \Lambda_{t+1}^{\boldsymbol{\theta}} + (\bar{G}_t^u)' \bar{V}_t^{\boldsymbol{\theta}} + (H_t^u)' W_t^{\boldsymbol{\theta}}, \\
& H_t^x X_t^{\boldsymbol{\theta}} + H_t^u U_t^{\boldsymbol{\theta}} + H_t^{\boldsymbol{\theta}} = \boldsymbol{0} \quad \text{and} \quad H_T^x X_T^{\boldsymbol{\theta}} + H_T^{\boldsymbol{\theta}} = \boldsymbol{0}, \\
& \bar{G}_t^x X_t^{\boldsymbol{\theta}} + \bar{G}_t^u U_t^{\boldsymbol{\theta}} + \bar{G}_t^{\boldsymbol{\theta}} = \boldsymbol{0} \quad \text{and} \quad \bar{G}_T^x X_T^{\boldsymbol{\theta}} + \bar{G}_T^{\boldsymbol{\theta}} = \boldsymbol{0}.
\end{aligned}
$$

(S.21)

Note that in (S.21), we have used the following matrix calculus [86] and trace properties:

$$
\frac{\partial \mathrm{Tr}(AB)}{\partial A} = B', \quad \frac{\partial f(A)}{\partial A'} = \left[\frac{\partial f(A)}{\partial A}\right]', \quad \frac{\partial \mathrm{Tr}(X'HX)}{\partial X} = HX + H'X,
$$
$$
\mathrm{Tr}(A) = \mathrm{Tr}(A'), \quad \mathrm{Tr}(ABC) = \mathrm{Tr}(BCA) = \mathrm{Tr}(CAB), \quad \mathrm{Tr}(A+B) = \mathrm{Tr}(A) + \mathrm{Tr}(B).
$$

Next, we need to show that the second-order condition

$$
\sum_{t=0}^{T-1} \mathrm{Tr}\left(\begin{bmatrix} \Delta X_t \\ \Delta U_t \end{bmatrix}'\underbrace{\begin{bmatrix} \frac{\partial \bar{L}_t^2}{\partial X_t^{\boldsymbol{\theta}} \partial X_t^{\boldsymbol{\theta}}} & \frac{\partial \bar{L}_t^2}{\partial X_t^{\boldsymbol{\theta}} \partial U_t^{\boldsymbol{\theta}}} \\ \frac{\partial \bar{L}_t^2}{\partial U_t^{\boldsymbol{\theta}} \partial X_t^{\boldsymbol{\theta}}} & \frac{\partial \bar{L}_t^2}{\partial U_t^{\boldsymbol{\theta}} \partial U_t^{\boldsymbol{\theta}}} \end{bmatrix}}_{\begin{bmatrix} L_t^{xx} & L_t^{xu} \\ L_t^{ux} & L_t^{uu} \end{bmatrix}}\begin{bmatrix} \Delta X_t \\ \Delta U_t \end{bmatrix}\right) + \mathrm{Tr}\left(\Delta X_T'\underbrace{\left[\frac{\partial \bar{L}_t^2}{\partial X_T^{\boldsymbol{\theta}} \partial X_T^{\boldsymbol{\theta}}}\right]}_{L_T^{xx}}\Delta X_T\right) > 0,
$$

(S.22)

hold for any trajectory $\{\Delta X_{0:T}, \Delta U_{0:T-1}\} \neq \mathbf{0}$ satisfying

$$
\begin{aligned}
\Delta X_{t+1} &= F_t^x \Delta X_t + F_t^u \Delta U_t \quad \text{and} \quad \Delta X_0 = \mathbf{0}, \\
\bar{G}_t^x \Delta X_t + \bar{G}_t^u \Delta U_t &= \mathbf{0} \quad \text{and} \quad \bar{G}_T^x \Delta X_T = \mathbf{0}, \\
H_t^x \Delta X_t + H_t^u \Delta U_t &= \mathbf{0} \quad \text{and} \quad H_T^x \Delta X_T = \mathbf{0},
\end{aligned}
\tag{S.23}
$$

In fact, this is true directly due to (S.6) and (S.7) in Lemma A.2 and the strict complementarity in condition (iii) in Lemma 1 (note that $\check{g}_t(x_t, u_t, \theta) = \bar{g}_t(x_t, u_t, \theta)$ and $\check{g}_T(x_T, \theta) = \bar{g}_T(x_T, \theta)$ because of the strict complementarity). Therefore, with the C-PMP (S.21) and (S.22)-(S.23) holding for $\{X_{0:T}^{\theta}, U_{0:T-1}^{\theta}\}$, we can conclude that $\{X_{0:T}^{\theta}, U_{0:T-1}^{\theta}\}$ is a local unique minimizing trajectory to the auxiliary control system $\bar{\Sigma}(\xi_{\theta})$ according to Lemma A.2.

Second, we prove that the local unique minimizing trajectory $\{X_{0:T}^{\theta}, U_{0:T-1}^{\theta}\}$ is also a *global* one.

We note that any feasible trajectory $\{X_{0:T}, U_{0:T-1}\}$ that satisfies all constraints (dynamics, path and final constraints) in the auxiliary control system $\bar{\Sigma}(\xi_{\theta})$ can be written as

$$
\{X_{0:T}, U_{0:T-1}\} = \{X_{0:T}^{\theta}, U_{0:T-1}^{\theta}\} + \{\Delta X_{0:T}, \Delta U_{0:T-1}\},
\tag{S.24}
$$

with $\{\Delta X_{0:T}, \Delta U_{0:T-1}\}$ satisfying the conditions in (S.23). Let

$$
\begin{aligned}
&\bar{J}(X_{0:T}, U_{0:T-1}) - \bar{J}(X_{0:T}^{\theta}, U_{0:T-1}^{\theta}) \\
&= \text{Tr} \sum_{t=0}^{T-1} \left( \frac{1}{2} \begin{bmatrix} \Delta X_t \\ \Delta U_t \end{bmatrix}' \begin{bmatrix} L_t^{xx} & L_t^{xu} \\ L_t^{ux} & L_t^{uu} \end{bmatrix} \begin{bmatrix} \Delta X_t \\ \Delta U_t \end{bmatrix} + \begin{bmatrix} X_t^{\theta} \\ U_t^{\theta} \end{bmatrix}' \begin{bmatrix} L_t^{xx} & L_t^{xu} \\ L_t^{ux} & L_t^{uu} \end{bmatrix} \begin{bmatrix} \Delta X_t \\ \Delta U_t \end{bmatrix} + \begin{bmatrix} L_t^{x\theta} \\ L_t^{u\theta} \end{bmatrix}' \begin{bmatrix} \Delta X_t \\ \Delta U_t \end{bmatrix} \right) \\
&\quad + \text{Tr} \left( \frac{1}{2} \Delta X_T' L_T^{xx} \Delta X_T + (X_T^{\theta})' L_T^{xx} \Delta X_T + (L_T^{x\theta})' \Delta X_T \right).
\end{aligned}
\tag{S.25}
$$

Based on (S.21), the following term in (S.25) can be simplified to

$$
\begin{aligned}
&\begin{bmatrix} \Delta X_t \\ \Delta U_t \end{bmatrix}' \left( \begin{bmatrix} L_t^{xx} & L_t^{xu} \\ L_t^{ux} & L_t^{uu} \end{bmatrix} \begin{bmatrix} X_t^{\theta} \\ U_t^{\theta} \end{bmatrix} + \begin{bmatrix} L_t^{x\theta} \\ L_t^{u\theta} \end{bmatrix} \right) \\
&= \begin{bmatrix} \Delta X_t \\ \Delta U_t \end{bmatrix}' \begin{bmatrix} -(F_t^x)'\Lambda_{t+1}^{\theta} - (\bar{G}_t^x)'\bar{V}_t^{\theta} - (H_t^x)'W_t^{\theta} + \Lambda_t^{\theta} \\ -(F_t^u)'\Lambda_{t+1}^{\theta} - (\bar{G}_t^u)'\bar{V}_t^{\theta} - (H_t^u)'W_t^{\theta} \end{bmatrix} \\
&= -(\Lambda_{t+1}^{\theta})' F_t^x \Delta X_t - \underbrace{(\bar{V}_t^{\theta})'\bar{G}_t^x \Delta X_t} - \underbrace{(W_t^{\theta})'H_t^x \Delta X_t} + (\Lambda_t^{\theta})'\Delta X_t \\
&\quad - (\Lambda_{t+1}^{\theta})' F_t^u \Delta U_t - \underbrace{(\bar{V}_t^{\theta})'\bar{G}_t^u \Delta U_t} - \underbrace{(W_t^{\theta})'H_t^u \Delta U_t} \\
&= \underbrace{-(\Lambda_{t+1}^{\theta})' F_t^x \Delta X_t - (\Lambda_{t+1}^{\theta})' F_t^u \Delta U_t}_{-(\Lambda_{t+1}^{\theta})'\Delta X_{t+1}} + (\Lambda_t^{\theta})'\Delta X_t = -(\Lambda_{t+1}^{\theta})'\Delta X_{t+1} + (\Lambda_t^{\theta})'\Delta X_t
\end{aligned}
\tag{S.26}
$$

where the cancellations in the last three lines are due to (S.23). Also based on (S.21), the following term in (S.25) can be simplified to

$$
\begin{aligned}
&\left( (X_T^{\theta})' L_T^{xx} + (L_T^{x\theta})' \right) \Delta X_T \\
&= -\underbrace{(\bar{V}_T^{\theta})'\bar{G}_T \Delta X_T} - \underbrace{(W_T^{\theta})'H_T^x \Delta X_T} + (\Lambda_T^{\theta})'\Delta X_T \\
&= (\Lambda_T^{\theta})'\Delta X_T
\end{aligned}
\tag{S.27}
$$

where the cancellation here is due to (S.23).

Then, based on (S.26) and (S.27), (S.25) is simplified to

$$
\begin{aligned}
&\bar{J}(X_{0:T}, U_{0:T-1}) - \bar{J}(X_{0:T}^{\boldsymbol{\theta}}, U_{0:T-1}^{\boldsymbol{\theta}}) \\
&= \mathrm{Tr} \sum_{t=0}^{T-1} \left( \frac{1}{2} \begin{bmatrix} \Delta X_t \\ \Delta U_t \end{bmatrix}' \begin{bmatrix} L_t^{xx} & L_t^{xu} \\ L_t^{ux} & L_t^{uu} \end{bmatrix} \begin{bmatrix} \Delta X_t \\ \Delta U_t \end{bmatrix} + \begin{bmatrix} X_t^{\boldsymbol{\theta}} \\ U_t^{\boldsymbol{\theta}} \end{bmatrix}' \begin{bmatrix} L_t^{xx} & L_t^{xu} \\ L_t^{ux} & L_t^{uu} \end{bmatrix} \begin{bmatrix} \Delta X_t \\ \Delta U_t \end{bmatrix} + \begin{bmatrix} L_t^{x\theta} \\ L_t^{u\theta} \end{bmatrix}' \begin{bmatrix} \Delta X_t \\ \Delta U_t \end{bmatrix} \right) \\
&\quad + \mathrm{Tr} \left( \frac{1}{2} \Delta X_T' L_T^{xx} \Delta X_T + (X_T^{\boldsymbol{\theta}})' L_T^{xx} \Delta X_T + (L_T^{x\theta})' \Delta X_T \right) \\
&= \mathrm{Tr} \sum_{t=0}^{T-1} \left( \frac{1}{2} \begin{bmatrix} \Delta X_t \\ \Delta U_t \end{bmatrix}' \begin{bmatrix} L_t^{xx} & L_t^{xu} \\ L_t^{ux} & L_t^{uu} \end{bmatrix} \begin{bmatrix} \Delta X_t \\ \Delta U_t \end{bmatrix} - (\Lambda_{t+1}^{\boldsymbol{\theta}})' \Delta X_{t+1} + (\Lambda_t^{\boldsymbol{\theta}})' \Delta X_t \right) \\
&\quad + \mathrm{Tr} \left( \frac{1}{2} \Delta X_T' L_T^{xx} \Delta X_T + (\Lambda_T^{\boldsymbol{\theta}})' \Delta X_T \right) \\
&= \mathrm{Tr} \sum_{t=0}^{T-1} \left( \frac{1}{2} \begin{bmatrix} \Delta X_t \\ \Delta U_t \end{bmatrix}' \begin{bmatrix} L_t^{xx} & L_t^{xu} \\ L_t^{ux} & L_t^{uu} \end{bmatrix} \begin{bmatrix} \Delta X_t \\ \Delta U_t \end{bmatrix} \right) + \mathrm{Tr} \left( \frac{1}{2} \Delta X_T' L_T^{xx} \Delta X_T \right),
\end{aligned}
$$
(S.28)

where the last line is because (note $\Delta X_0 = \mathbf{0}$ in (S.23))

$$
\mathrm{Tr} \sum_{t=0}^{T-1} \left( -(\Lambda_{t+1}^{\boldsymbol{\theta}})' \Delta X_{t+1} + (\Lambda_t^{\boldsymbol{\theta}})' \Delta X_t \right) + \mathrm{Tr} \left( (\Lambda_T^{\boldsymbol{\theta}})' \Delta X_T \right) = \mathrm{Tr} \left( (\Lambda_0^{\boldsymbol{\theta}})' \Delta X_0 \right) = 0.
$$

Since

$$
\mathrm{Tr} \sum_{t=0}^{T-1} \left( \frac{1}{2} \begin{bmatrix} \Delta X_t \\ \Delta U_t \end{bmatrix}' \begin{bmatrix} L_t^{xx} & L_t^{xu} \\ L_t^{ux} & L_t^{uu} \end{bmatrix} \begin{bmatrix} \Delta X_t \\ \Delta U_t \end{bmatrix} \right) + \mathrm{Tr} \left( \frac{1}{2} \Delta X_T' L_T^{xx} \Delta X_T \right) \geq 0 \qquad \text{(S.29)}
$$

due to (S.22) for all $\{\Delta X_{0:T}, \Delta U_{0:T-1}\}$ satisfying (S.23), therefore

$$
\bar{J}(X_{0:T}, U_{0:T-1}) - \bar{J}(X_{0:T}^{\boldsymbol{\theta}}, U_{0:T-1}^{\boldsymbol{\theta}}) \geq 0. \qquad \text{(S.30)}
$$

for any feasible trajectory $\{X_{0:T}, U_{0:T-1}\}$ in (S.24). This concludes that the local unique minimizing trajectory $\{X_{0:T}^{\boldsymbol{\theta}}, U_{0:T}^{\boldsymbol{\theta}}\}$ is also a *global* one.

In sum of the two proof steps, the assertion that the trajectory in (4), i.e.,

$$
\frac{\partial \boldsymbol{\xi}_{\boldsymbol{\theta}}}{\partial \boldsymbol{\theta}} = \left\{ X_{0:T}^{\boldsymbol{\theta}}, U_{0:T-1}^{\boldsymbol{\theta}} \right\},
$$

is a globally unique minimizing trajectory to the auxiliary control system $\overline{\boldsymbol{\Sigma}}(\boldsymbol{\xi}_{\boldsymbol{\theta}})$ in (3) follows. This completes the proof of Theorem 1.

$\square$

## C  Proof of Theorem 2

For the unconstrained optimal control system $\boldsymbol{\Sigma}(\boldsymbol{\theta}, \gamma)$ in (5), we define its Hamiltonian below:

$$
\hat{L}_t = c_t(\boldsymbol{x}_t, \boldsymbol{u}_t, \boldsymbol{\theta}) + \boldsymbol{\lambda}_{t+1}' \boldsymbol{f}(\boldsymbol{x}_t, \boldsymbol{u}_t, \boldsymbol{\theta}) - \gamma \sum_{i=1}^{q_t} \ln \left( -g_{t,i}(\boldsymbol{x}_t, \boldsymbol{u}_t, \boldsymbol{\theta}) \right) + \frac{1}{2\gamma} \sum_{i=1}^{s_t} \left( h_{t,i}(\boldsymbol{x}_t, \boldsymbol{u}_t, \boldsymbol{\theta}) \right)^2
$$

$$
\hat{L}_T = c_T(\boldsymbol{x}_T, \boldsymbol{\theta}) - \gamma \sum_{i=1}^{q_T} \ln \left( -g_{T,i}(\boldsymbol{x}_T, \boldsymbol{\theta}) \right) + \frac{1}{2\gamma} \sum_{i=1}^{s_T} \left( h_{T,i}(\boldsymbol{x}_T, \boldsymbol{\theta}) \right)^2.
$$
(S.31)

with $t = 0, 1, \cdots, T-1$.

## C.1  Proof of Claim (a)

We first modify the C-PMP condition (S.5) for the constrained optimal control system $\boldsymbol{\Sigma}(\boldsymbol{\theta})$ into the following set of equations:

$$
\begin{aligned}
&\boldsymbol{x}_{t+1} = \boldsymbol{f}(\boldsymbol{x}_t, \boldsymbol{u}_t, \boldsymbol{\theta}) \quad \text{and} \quad \boldsymbol{x}_0 = \boldsymbol{x}_0(\boldsymbol{\theta}), \\
&\boldsymbol{\lambda}_t = L_t^x(\boldsymbol{x}_t, \boldsymbol{u}_t, \boldsymbol{\lambda}_{t+1}, \boldsymbol{v}_t, \boldsymbol{w}_t, \boldsymbol{\theta}) \quad \text{and} \quad \boldsymbol{\lambda}_T = L_T^x(\boldsymbol{x}_t, \boldsymbol{v}_t, \boldsymbol{w}_t, \boldsymbol{\theta}), \\
&\boldsymbol{0} = L_t^u(\boldsymbol{x}_t, \boldsymbol{u}_t, \boldsymbol{\lambda}_{t+1}, \boldsymbol{v}_t, \boldsymbol{w}_t, \boldsymbol{\theta}), \\
&h_{t,i}(\boldsymbol{x}_t, \boldsymbol{u}_t, \boldsymbol{\theta}) = w_{t,i}\gamma, \quad i = 1, 2, \cdots, s_t, \\
&h_{T,i}(\boldsymbol{x}_T, \boldsymbol{\theta}) = w_{T,i}\gamma, \quad i = 1, 2, \cdots, s_T, \\
&v_{t,i}\, g_{t,i}(\boldsymbol{x}_t, \boldsymbol{u}_t, \boldsymbol{\theta}) = -\gamma, \quad i = 1, 2, \cdots, q_t, \\
&v_{T,i}\, g_{T,i}(\boldsymbol{x}_T, \boldsymbol{\theta}) = -\gamma, \quad i = 1, 2, \cdots, q_T,
\end{aligned}
\tag{S.32}
$$

where the first three equations are the same with the those in (S.5) and only the last two lines of equations are modified by adding some perturbation terms related to $\gamma$.

Now, one can view that the parameters $(\boldsymbol{\theta}, \gamma)$ jointly determine $\boldsymbol{\xi} = \{\boldsymbol{x}_{0:T}, \boldsymbol{u}_{0:T-1}\}$, $\boldsymbol{\lambda}_{1:T}$, $\boldsymbol{v}_{0:T}$, and $\boldsymbol{w}_{0:T}$ through the implicit equations in (S.32). Also, one can note that by letting $\gamma = 0$ and $\boldsymbol{\theta} = \bar{\boldsymbol{\theta}}$, the above equations in (S.32) coincide with the C-PMP condition (S.5) for $\boldsymbol{\Sigma}(\bar{\boldsymbol{\theta}})$. Thus, given that the conditions (i)-(iii) in Lemma 1 hold for $\boldsymbol{\Sigma}(\bar{\boldsymbol{\theta}})$, one can readily apply the implicit function theorem [75] to (S.32) in a neighborhood of $(\bar{\boldsymbol{\theta}}, 0)$ and make the following assertion (its proof can directly follow the proof for Lemma 1 (i.e., the first-order sensitivity result) with little change):

*For any $(\boldsymbol{\theta}, \gamma)$ within a neighborhood of $(\bar{\boldsymbol{\theta}}, 0)$, there exists a unique once-continuously differentiable function $\left( \boldsymbol{\xi}_{(\boldsymbol{\theta},\gamma)}, \boldsymbol{\lambda}_{0:T}^{(\boldsymbol{\theta},\gamma)}, \boldsymbol{v}_{0:T}^{(\boldsymbol{\theta},\gamma)}, \boldsymbol{w}_{0:T}^{(\boldsymbol{\theta},\gamma)} \right)$, which satisfies (S.32) and*

$$
\left( \boldsymbol{\xi}_{(\boldsymbol{\theta},\gamma)}, \boldsymbol{\lambda}_{0:T}^{(\boldsymbol{\theta},\gamma)}, \boldsymbol{v}_{0:T}^{(\boldsymbol{\theta},\gamma)}, \boldsymbol{w}_{0:T}^{(\boldsymbol{\theta},\gamma)} \right) = \left( \boldsymbol{\xi}_{\bar{\boldsymbol{\theta}}}, \boldsymbol{\lambda}_{1:T}^{\bar{\boldsymbol{\theta}}}, \boldsymbol{v}_{0:T}^{\bar{\boldsymbol{\theta}}}, \boldsymbol{w}_{0:T}^{\bar{\boldsymbol{\theta}}} \right) \quad \text{when} \quad (\boldsymbol{\theta}, \gamma) = (\bar{\boldsymbol{\theta}}, 0). \tag{S.33}
$$

With the above claim, in what follows, we will prove that for any $(\boldsymbol{\theta}, \gamma)$ near $(\bar{\boldsymbol{\theta}}, 0)$ additionally with $\gamma > 0$, $\boldsymbol{\xi}_{(\boldsymbol{\theta},\gamma)}$ is a local isolated minimizing trajectory to the unconstrained optimal control system $\boldsymbol{\Sigma}(\boldsymbol{\theta}, \gamma)$ in (5). First, we need to show that such $\boldsymbol{\xi}_{(\boldsymbol{\theta},\gamma)}$ will make $\boldsymbol{\Sigma}(\boldsymbol{\theta}, \gamma)$ well-defined, which is the second part of Claim (a), rewritten below

$$
\begin{aligned}
&g_{t,i}\left( \boldsymbol{x}_t^{(\boldsymbol{\theta},\gamma)}, \boldsymbol{u}_t^{(\boldsymbol{\theta},\gamma)}, \boldsymbol{\theta} \right) < 0, \quad i = 1, 2, \cdots, q_t, \quad \text{and} \\
&g_{T,i}\left( \boldsymbol{x}_T^{(\boldsymbol{\theta},\gamma)}, \boldsymbol{\theta} \right) < 0, \quad i = 1, 2, \cdots, q_T.
\end{aligned}
\tag{S.34}
$$

In fact, such an assertion always holds because the strict complementary for $\boldsymbol{\Sigma}(\bar{\boldsymbol{\theta}})$ from Lemma 1. Specifically, for any $i = 1, 2, \cdots, q_t$, if $g_{t,i}(\boldsymbol{x}_t^{\bar{\boldsymbol{\theta}}}, \boldsymbol{u}_t^{\bar{\boldsymbol{\theta}}}, \bar{\boldsymbol{\theta}}) < 0$, from continuity of $g_{t,i}$ and $\boldsymbol{\xi}_{(\boldsymbol{\theta},\gamma)}$

$$
g_{t,i}(\boldsymbol{x}_t^{(\boldsymbol{\theta},\gamma)}, \boldsymbol{u}_t^{(\boldsymbol{\theta},\gamma)}, \boldsymbol{\theta}) \to g_{t,i}(\boldsymbol{x}_t^{\bar{\boldsymbol{\theta}}}, \boldsymbol{u}_t^{\bar{\boldsymbol{\theta}}}, \bar{\boldsymbol{\theta}}) < 0 \quad as \quad (\boldsymbol{\theta}, \gamma) \to (\bar{\boldsymbol{\theta}}, 0),
$$

thus $g_{t,i}(\boldsymbol{x}_t^{(\boldsymbol{\theta},\gamma)}, \boldsymbol{u}_t^{(\boldsymbol{\theta},\gamma)}, \boldsymbol{\theta}) < 0$ for any $(\boldsymbol{\theta}, \gamma)$ near $(\bar{\boldsymbol{\theta}}, 0)$ with $\gamma > 0$; if $g_{t,i}(\boldsymbol{x}_t^{\bar{\boldsymbol{\theta}}}, \boldsymbol{u}_t^{\bar{\boldsymbol{\theta}}}, \bar{\boldsymbol{\theta}}) = 0$ and $v_{t,i}^{\bar{\boldsymbol{\theta}}} > 0$ (due to strict complementarity), from continuity of $\boldsymbol{v}_t^{(\boldsymbol{\theta},\gamma)}$,

$$
v_{t,i}^{(\boldsymbol{\theta},\gamma)} \to v_{t,i}^{\bar{\boldsymbol{\theta}}} > 0 \quad as \quad (\boldsymbol{\theta}, \gamma) \to (\bar{\boldsymbol{\theta}}, 0), \tag{S.35}
$$

thus $v_{t,i}^{(\boldsymbol{\theta},\gamma)} > 0$ for $(\boldsymbol{\theta}, \gamma)$ near $(\bar{\boldsymbol{\theta}}, 0)$ with $\gamma > 0$, and also due to (S.32), $g_{t,i}(\boldsymbol{x}_t^{(\boldsymbol{\theta},\gamma)}, \boldsymbol{u}_t^{(\boldsymbol{\theta},\gamma)}, \boldsymbol{\theta}) = -\frac{\gamma}{v_{t,i}^{(\boldsymbol{\theta},\gamma)}} < 0$ for $(\boldsymbol{\theta}, \gamma)$ near $(\bar{\boldsymbol{\theta}}, 0)$ with $\gamma > 0$. So, for either case, the first inequality in (S.34) always holds. Similar proof procedure also applies to prove the second inequality in (S.34). In sum, we conclude that $\boldsymbol{\xi}_{(\boldsymbol{\theta},\gamma)}$ satisfies (S.34) and thus makes the $\boldsymbol{\Sigma}(\boldsymbol{\theta}, \gamma)$ well-defined for any $(\boldsymbol{\theta}, \gamma)$ near $(\bar{\boldsymbol{\theta}}, 0)$ with $\gamma > 0$. This completes the second part of Claim (a).

From now on, we prove that for any $(\boldsymbol{\theta}, \gamma)$ near $(\bar{\boldsymbol{\theta}}, 0)$ with $\gamma > 0$, $\boldsymbol{\xi}_{(\boldsymbol{\theta}, \gamma)}$ is a local isolated minimizing trajectory to the unconstrained optimal control system $\boldsymbol{\Sigma}(\boldsymbol{\theta}, \gamma)$ in (5). From the last four equations in (S.32), we solve

$$
\begin{aligned}
w_{t,i}^{(\boldsymbol{\theta}, \gamma)} &= \frac{h_{t,i}(\boldsymbol{x}_t^{(\boldsymbol{\theta}, \gamma)}, \boldsymbol{u}_t^{(\boldsymbol{\theta}, \gamma)}, \boldsymbol{\theta})}{\gamma}, \qquad & w_{T,i}^{(\boldsymbol{\theta}, \gamma)} &= \frac{h_{T,i}(\boldsymbol{x}_T^{(\boldsymbol{\theta}, \gamma)}, \boldsymbol{\theta})}{\gamma}, \\
v_{t,i}^{(\boldsymbol{\theta}, \gamma)} &= -\frac{\gamma}{g_{t,i}(\boldsymbol{x}_t^{(\boldsymbol{\theta}, \gamma)}, \boldsymbol{u}_t^{(\boldsymbol{\theta}, \gamma)}, \boldsymbol{\theta})}, \qquad & v_{T,i}^{(\boldsymbol{\theta}, \gamma)} &= -\frac{\gamma}{g_{T,i}(\boldsymbol{x}_T^{(\boldsymbol{\theta}, \gamma)}, \boldsymbol{\theta})},
\end{aligned}
\tag{S.36}
$$

and plug them into the first three equations in (S.32), then one will find that the obtained equations are exactly the Pontryagin Maximum/Minimum Principle (PMP) for the unconstrained optimal control system $\boldsymbol{\Sigma}(\boldsymbol{\theta}, \gamma)$ with its Hamiltonian already defined in (S.31), that is to say,

$$
\begin{aligned}
\boldsymbol{x}_{t+1}^{(\boldsymbol{\theta}, \gamma)} &= \boldsymbol{f}(\boldsymbol{x}_t^{(\boldsymbol{\theta}, \gamma)}, \boldsymbol{u}_t^{(\boldsymbol{\theta}, \gamma)}, \boldsymbol{\theta}) \quad \text{and} \quad \boldsymbol{x}_0^{(\boldsymbol{\theta}, \gamma)} = \boldsymbol{x}_0(\boldsymbol{\theta}), \\
\boldsymbol{\lambda}_t^{(\boldsymbol{\theta}, \gamma)} &= \hat{L}_t^x(\boldsymbol{x}_t^{(\boldsymbol{\theta}, \gamma)}, \boldsymbol{u}_t^{(\boldsymbol{\theta}, \gamma)}, \boldsymbol{\lambda}_{t+1}^{(\boldsymbol{\theta}, \gamma)}, (\boldsymbol{\theta}, \gamma)), \\
\boldsymbol{\lambda}_T^{(\boldsymbol{\theta}, \gamma)} &= \hat{L}_T^x(\boldsymbol{x}_t^{(\boldsymbol{\theta}, \gamma)}, (\boldsymbol{\theta}, \gamma)), \\
\boldsymbol{0} &= \hat{L}_t^u(\boldsymbol{x}_t^{(\boldsymbol{\theta}, \gamma)}, \boldsymbol{u}_t^{(\boldsymbol{\theta}, \gamma)}, \boldsymbol{\lambda}_{t+1}^{(\boldsymbol{\theta}, \gamma)}, (\boldsymbol{\theta}, \gamma)),
\end{aligned}
\tag{S.37}
$$

indicating that $\boldsymbol{\xi}_{(\boldsymbol{\theta}, \gamma)} = \left\{ \boldsymbol{x}_{0:T}^{(\boldsymbol{\theta}, \gamma)}, \boldsymbol{u}_{0:T-1}^{(\boldsymbol{\theta}, \gamma)} \right\}$ already satisfies the PMP condition for unconstrained optimal control system $\boldsymbol{\Sigma}(\boldsymbol{\theta}, \gamma)$. To show $\boldsymbol{\xi}_{(\boldsymbol{\theta}, \gamma)} = \left\{ \boldsymbol{x}_{0:T}^{(\boldsymbol{\theta}, \gamma)}, \boldsymbol{u}_{0:T-1}^{(\boldsymbol{\theta}, \gamma)} \right\}$ is a local isolated minimizing trajectory to $\boldsymbol{\Sigma}(\boldsymbol{\theta}, \gamma)$ for any $(\boldsymbol{\theta}, \gamma)$ near $(\bar{\boldsymbol{\theta}}, 0)$ with $\gamma > 0$, we only need to verify its second-order condition as stated in (S.6)-(S.7) in Lemma A.2, which is presented next. In the remainder of proof, for convenience of notation, all derivatives are evaluated at $(\boldsymbol{\theta}, \gamma)$ (or $\boldsymbol{\xi}_{(\boldsymbol{\theta}, \gamma)}$) unless otherwise stated.

Before proceeding, we show two facts (easy to prove) about the second-order derivatives of Hamiltonian $\hat{L}_t$ and $\hat{L}_T$ in (S.31). First,

$$
\begin{bmatrix} \hat{L}_t^{xx} & \hat{L}_t^{xu} \\ \hat{L}_t^{ux} & \hat{L}_t^{uu} \end{bmatrix} = \begin{bmatrix} L_t^{xx} & L_t^{xu} \\ L_t^{ux} & L_t^{uu} \end{bmatrix} +
$$

$$
\begin{bmatrix} \sum_{i=1}^{q_t} \frac{\gamma}{g_{t,i}^2} \frac{\partial g_{t,i}'}{\partial \boldsymbol{x}_t} \frac{\partial g_{t,i}}{\partial \boldsymbol{x}_t} + \sum_{i=1}^{s_t} \frac{1}{\gamma} \frac{\partial h_{t,i}'}{\partial \boldsymbol{x}_t} \frac{\partial h_{t,i}}{\partial \boldsymbol{x}_t} & \sum_{i=1}^{q_t} \frac{\gamma}{g_{t,i}^2} \frac{\partial g_{t,i}'}{\partial \boldsymbol{x}_t} \frac{\partial g_{t,i}}{\partial \boldsymbol{u}_t} + \sum_{i=1}^{s_t} \frac{1}{\gamma} \frac{\partial h_{t,i}'}{\partial \boldsymbol{x}_t} \frac{\partial h_{t,i}}{\partial \boldsymbol{u}_t} \\ \sum_{i=1}^{q_t} \frac{\gamma}{g_{t,i}^2} \frac{\partial g_{t,i}'}{\partial \boldsymbol{u}_t} \frac{\partial g_{t,i}}{\partial \boldsymbol{x}_t} + \sum_{i=1}^{s_t} \frac{1}{\gamma} \frac{\partial h_{t,i}'}{\partial \boldsymbol{u}_t} \frac{\partial h_{t,i}}{\partial \boldsymbol{x}_t} & \sum_{i=1}^{q_t} \frac{\gamma}{g_{t,i}^2} \frac{\partial g_{t,i}'}{\partial \boldsymbol{u}_t} \frac{\partial g_{t,i}}{\partial \boldsymbol{u}_t} + \sum_{i=1}^{s_t} \frac{1}{\gamma} \frac{\partial h_{t,i}'}{\partial \boldsymbol{u}_t} \frac{\partial h_{t,i}}{\partial \boldsymbol{u}_t} \end{bmatrix}, \tag{S.38}
$$

and

$$
\hat{L}_T^{xx} = L_T^{xx} + \sum_{i=1}^{q_T} \frac{\gamma}{g_{t,i}^2} \frac{\partial g_{T,i}'}{\partial \boldsymbol{x}_T} \frac{\partial g_{T,i}}{\partial \boldsymbol{x}_T} + \sum_{i=1}^{s_T} \frac{1}{\gamma} \frac{\partial h_{T,i}'}{\partial \boldsymbol{x}_T} \frac{\partial h_{T,i}}{\partial \boldsymbol{x}_T}, \tag{S.39}
$$

respectively. Second, given any $\boldsymbol{x}$ and $\boldsymbol{u}$ with appropriate dimensions, one has

$$
\begin{bmatrix} \boldsymbol{x} \\ \boldsymbol{u} \end{bmatrix}' \begin{bmatrix} \hat{L}_t^{xx} & \hat{L}_t^{xu} \\ \hat{L}_t^{ux} & \hat{L}_t^{uu} \end{bmatrix} \begin{bmatrix} \boldsymbol{x} \\ \boldsymbol{u} \end{bmatrix} = \begin{bmatrix} \boldsymbol{x} \\ \boldsymbol{u} \end{bmatrix}' \begin{bmatrix} L_t^{xx} & L_t^{xu} \\ L_t^{ux} & L_t^{uu} \end{bmatrix} \begin{bmatrix} \boldsymbol{x} \\ \boldsymbol{u} \end{bmatrix} +
$$

$$
\begin{bmatrix} \boldsymbol{x} \\ \boldsymbol{u} \end{bmatrix}' \begin{bmatrix} \sum_{i=1}^{q_t} \frac{\gamma}{g_{t,i}^2} \frac{\partial g_{t,i}'}{\partial \boldsymbol{x}_t} \frac{\partial g_{t,i}}{\partial \boldsymbol{x}_t} + \sum_{i=1}^{s_t} \frac{1}{\gamma} \frac{\partial h_{t,i}'}{\partial \boldsymbol{x}_t} \frac{\partial h_{t,i}}{\partial \boldsymbol{x}_t} & \sum_{i=1}^{q_t} \frac{\gamma}{g_{t,i}^2} \frac{\partial g_{t,i}'}{\partial \boldsymbol{x}_t} \frac{\partial g_{t,i}}{\partial \boldsymbol{u}_t} + \sum_{i=1}^{s_t} \frac{1}{\gamma} \frac{\partial h_{t,i}'}{\partial \boldsymbol{x}_t} \frac{\partial h_{t,i}}{\partial \boldsymbol{u}_t} \\ \sum_{i=1}^{q_t} \frac{\gamma}{g_{t,i}^2} \frac{\partial g_{t,i}'}{\partial \boldsymbol{u}_t} \frac{\partial g_{t,i}}{\partial \boldsymbol{x}_t} + \sum_{i=1}^{s_t} \frac{1}{\gamma} \frac{\partial h_{t,i}'}{\partial \boldsymbol{u}_t} \frac{\partial h_{t,i}}{\partial \boldsymbol{x}_t} & \sum_{i=1}^{q_t} \frac{\gamma}{g_{t,i}^2} \frac{\partial g_{t,i}'}{\partial \boldsymbol{u}_t} \frac{\partial g_{t,i}}{\partial \boldsymbol{u}_t} + \sum_{i=1}^{s_t} \frac{1}{\gamma} \frac{\partial h_{t,i}'}{\partial \boldsymbol{u}_t} \frac{\partial h_{t,i}}{\partial \boldsymbol{u}_t} \end{bmatrix} \begin{bmatrix} \boldsymbol{x} \\ \boldsymbol{u} \end{bmatrix}
$$

$$
= \begin{bmatrix} \boldsymbol{x} \\ \boldsymbol{u} \end{bmatrix}' \begin{bmatrix} L_t^{xx} & L_t^{xu} \\ L_t^{ux} & L_t^{uu} \end{bmatrix} \begin{bmatrix} \boldsymbol{x} \\ \boldsymbol{u} \end{bmatrix} + \sum_{i=1}^{q_t} \frac{\gamma}{g_{t,i}^2} \Big( \frac{\partial g_{t,i}}{\partial \boldsymbol{x}_t} \boldsymbol{x} + \frac{\partial g_{t,i}}{\partial \boldsymbol{u}} \boldsymbol{u} \Big)^2 + \sum_{i=1}^{s_t} \frac{1}{\gamma} \Big( \frac{\partial h_{t,i}}{\partial \boldsymbol{x}_t} \boldsymbol{x} + \frac{\partial h_{t,i}}{\partial \boldsymbol{u}} \boldsymbol{u} \Big)^2, \tag{S.40}
$$

and

$$
\boldsymbol{x}' \hat{L}_T^{xx} \boldsymbol{x} = \boldsymbol{x}' L_T^{xx} \boldsymbol{x} + \boldsymbol{x}' \Big( \sum_{i=1}^{q_T} \frac{\gamma}{g_{t,i}^2} \frac{\partial g_{T,i}'}{\partial \boldsymbol{x}_T} \frac{\partial g_{T,i}}{\partial \boldsymbol{x}_T} + \sum_{i=1}^{s_T} \frac{1}{\gamma} \frac{\partial h_{T,i}'}{\partial \boldsymbol{x}_T} \frac{\partial h_{T,i}}{\partial \boldsymbol{x}_T} \Big) \boldsymbol{x}
$$

$$
= \boldsymbol{x}' L_T^{xx} \boldsymbol{x} + \sum_{i=1}^{q_T} \frac{\gamma}{g_{T,i}^2} \Big( \frac{\partial g_{T,i}}{\partial \boldsymbol{x}_T} \boldsymbol{x} \Big)^2 + \sum_{i=1}^{s_T} \frac{1}{\gamma} \Big( \frac{\partial h_{T,i}}{\partial \boldsymbol{x}_T} \boldsymbol{x} \Big)^2. \tag{S.41}
$$

For the second-order condition of the unconstrained optimal control system $\Sigma(\boldsymbol{\theta}, \gamma)$ with any $(\boldsymbol{\theta}, \gamma)$ near $(\bar{\boldsymbol{\theta}}, 0)$ with $\gamma > 0$, we need to prove that

$$\sum_{t=0}^{T-1} \begin{bmatrix} \boldsymbol{x}_t \\ \boldsymbol{u}_t \end{bmatrix}' \begin{bmatrix} \hat{L}_t^{xx}(\boldsymbol{\theta}, \gamma) & \hat{L}_t^{xu}(\boldsymbol{\theta}, \gamma) \\ \hat{L}_t^{ux}(\boldsymbol{\theta}, \gamma) & \hat{L}_t^{uu}(\boldsymbol{\theta}, \gamma) \end{bmatrix} \begin{bmatrix} \boldsymbol{x}_t \\ \boldsymbol{u}_t \end{bmatrix} + \boldsymbol{x}_T' \hat{L}_T^{xx}(\boldsymbol{\theta}, \gamma) \boldsymbol{x}_T > 0, \tag{S.42}$$

for any $\{\boldsymbol{x}_{0:T}, \boldsymbol{u}_{0:T-1}\} \neq \boldsymbol{0}$ satisfying

$$\boldsymbol{x}_{t+1} = F_t^x(\boldsymbol{\theta}, \gamma) \boldsymbol{x}_t + F_t^u(\boldsymbol{\theta}, \gamma) \boldsymbol{u}_t \quad \text{and} \quad \boldsymbol{x}_0 = \boldsymbol{0}. \tag{S.43}$$

Here, for convenience, the dependence in $F_t^x(\boldsymbol{\theta}, \gamma)$, $F_t^u(\boldsymbol{\theta}, \gamma)$, $H_t^{xx}(\boldsymbol{\theta}, \gamma)$, $H_t^{xu}(\boldsymbol{\theta}, \gamma)$, $H_t^{uu}(\boldsymbol{\theta}, \gamma)$, and $H_T^{xx}(\boldsymbol{\theta}, \gamma)$ means that these first- and second-order derivatives are evaluated at trajectory $\boldsymbol{\xi}_{(\boldsymbol{\theta}, \gamma)}$ (the same notation convention applies below).

Proof by contradiction: suppose that the above second-order condition in (S.42)-(S.43) is false. Then, there must exist a sequence of parameters $(\boldsymbol{\theta}^k, \gamma^k)$ with $\gamma^k > 0$ and a sequence of trajectories $\{\boldsymbol{x}_{0:T}^k, \boldsymbol{u}_{0:T-1}^k\} \neq \boldsymbol{0}$ such that $(\boldsymbol{\theta}^k, \gamma^k) \to (\bar{\boldsymbol{\theta}}, 0)$, $\boldsymbol{x}_{t+1}^k = F_t^x(\boldsymbol{\theta}^k, \gamma^k) \boldsymbol{x}_t^k + F_t^u(\boldsymbol{\theta}^k, \gamma^k) \boldsymbol{u}_t^k$ with $\boldsymbol{x}_0^k = \boldsymbol{0}$, and

$$\sum_{t=0}^{T-1} \begin{bmatrix} \boldsymbol{x}_t^k \\ \boldsymbol{u}_t^k \end{bmatrix}' \begin{bmatrix} \hat{L}_t^{xx}(\boldsymbol{\theta}^k, \gamma^k) & \hat{L}_t^{xu}(\boldsymbol{\theta}^k, \gamma^k) \\ \hat{L}_t^{ux}(\boldsymbol{\theta}^k, \gamma^k) & \hat{L}_t^{uu}(\boldsymbol{\theta}^k, \gamma^k) \end{bmatrix} \begin{bmatrix} \boldsymbol{x}_t^k \\ \boldsymbol{u}_t^k \end{bmatrix} + \boldsymbol{x}_T^{k\prime} \hat{L}_T^{xx}(\boldsymbol{\theta}^k, \gamma^k) \boldsymbol{x}_T^k \leq 0, \tag{S.44}$$

for $k = 1, 2, 3, \cdots$. Here, the dependence $(\boldsymbol{\theta}^k, \gamma^k)$ means that these first- and second-order derivatives are evaluated at trajectory $\boldsymbol{\xi}_{(\boldsymbol{\theta}^k, \gamma^k)}$ for notation convenience. Without loss of generality, assume $\|\text{col}\{\boldsymbol{x}_{0:T}^k, \boldsymbol{u}_{0:T-1}^k\}\| = 1$ for all $k$. Select a convergent sub-sequence $\{\boldsymbol{x}_{0:T}^k, \boldsymbol{u}_{0:T-1}^k\}$, relabel the sequence $\{\boldsymbol{x}_{0:T}^k, \boldsymbol{u}_{0:T-1}^k\}$ for convenience, and call its limit $\{\boldsymbol{x}_{0:T}^*, \boldsymbol{u}_{0:T-1}^*\}$, that is, $\{\boldsymbol{x}_{0:T}^k, \boldsymbol{u}_{0:T-1}^k\} \to \{\boldsymbol{x}_{0:T}^*, \boldsymbol{u}_{0:T-1}^*\}$ and $(\boldsymbol{\theta}^k, \gamma^k) \to (\bar{\boldsymbol{\theta}}, 0)$ as $k \to +\infty$ and $\boldsymbol{x}_{t+1}^* = F_t^x(\bar{\boldsymbol{\theta}}, 0) \boldsymbol{x}_t^* + F_t^u(\bar{\boldsymbol{\theta}}, 0) \boldsymbol{u}_t^*$ with $\boldsymbol{x}_0^* = \boldsymbol{0}$. Then, the limit $\{\boldsymbol{x}_{0:T}^*, \boldsymbol{u}_{0:T-1}^*\}$ must fall into either of two cases discussed below.

*Case 1:* $\|\text{col}\{\boldsymbol{x}_{0:T}^*, \boldsymbol{u}_{0:T-1}^*\}\| = 1$ and at least one of the following holds:

$$\begin{aligned} \bar{G}_t^x(\bar{\boldsymbol{\theta}}, 0) \boldsymbol{x}_t^* + \bar{G}_t^u(\bar{\boldsymbol{\theta}}, 0) \boldsymbol{u}_t^* \neq \boldsymbol{0} \quad \exists t \quad \text{or} \quad H_t^x(\bar{\boldsymbol{\theta}}, 0) \boldsymbol{x}_t^* + H_t^u(\bar{\boldsymbol{\theta}}, 0) \boldsymbol{u}_t^* \neq \boldsymbol{0} \quad \exists t \\ \text{or} \quad \bar{G}_T^x(\bar{\boldsymbol{\theta}}, 0) \boldsymbol{x}_T^* \neq \boldsymbol{0} \quad \text{or} \quad H_T^x(\bar{\boldsymbol{\theta}}, 0) \boldsymbol{x}_T^* \neq \boldsymbol{0}. \end{aligned} \tag{S.45}$$

In this case, as $k \to 0$, $\{\boldsymbol{x}_{0:T}^k, \boldsymbol{u}_{0:T-1}^k\} \to \{\boldsymbol{x}_{0:T}^*, \boldsymbol{u}_{0:T-1}^*\}$, $(\boldsymbol{\theta}^k, \gamma^k) \to (\bar{\boldsymbol{\theta}}, 0)$, we will have

$$\sum_{t=0}^{T-1} \left( \sum_{i=1}^{q_t} \frac{\gamma^k}{(g_{t,i}^k)^2} \left( \frac{\partial g_{t,i}^k}{\partial \boldsymbol{x}_t} \boldsymbol{x}_t^k + \frac{\partial g_{t,i}^k}{\partial \boldsymbol{u}_t} \boldsymbol{u}_t^k \right)^2 + \sum_{i=1}^{s_t} \frac{1}{\gamma^k} \left( \frac{\partial h_{t,i}^k}{\partial \boldsymbol{x}_t} \boldsymbol{x}_t^k + \frac{\partial h_{t,i}^k}{\partial \boldsymbol{u}_t} \boldsymbol{u}_t^k \right)^2 \right)$$
$$+ \sum_{i=1}^{q_T} \frac{\gamma^k}{(g_{T,i}^k)^2} \left( \frac{\partial g_{T,i}^k}{\partial \boldsymbol{x}_T} \boldsymbol{x}_T^k \right)^2 + \sum_{i=1}^{s_T} \frac{1}{\gamma^k} \left( \frac{\partial h_{T,i}^k}{\partial \boldsymbol{x}_T} \boldsymbol{x}_T^k \right)^2 \to +\infty, \tag{S.46}$$

where $\frac{\partial g_{t,i}^k}{\partial \boldsymbol{x}_t}$, $\frac{\partial g_{T,i}^k}{\partial \boldsymbol{x}_T}$, $g_{t,i}^k$, $g_{T,i}^k$, $\frac{\partial h_{t,i}^k}{\partial \boldsymbol{x}_t}$, $\frac{\partial h_{T,i}^k}{\partial \boldsymbol{x}_T}$ are with superscript $k$ to denote their values are evaluated at $\boldsymbol{\xi}_{(\boldsymbol{\theta}^k, \gamma^k)}$ for notation convenience. (S.46) is because at least one of the terms in the summation is $+\infty$. Here, we have used the following facts from the last two equations in (S.32):

$$\frac{\gamma}{\left( g_{t,i}(\boldsymbol{x}_t^{(\boldsymbol{\theta}, \gamma)}, \boldsymbol{u}_t^{(\boldsymbol{\theta}, \gamma)}, \boldsymbol{\theta}) \right)^2} = -\frac{v_{t,i}^{(\boldsymbol{\theta}, \gamma)}}{g_{t,i}(\boldsymbol{x}_t^{(\boldsymbol{\theta}, \gamma)}, \boldsymbol{u}_t^{(\boldsymbol{\theta}, \gamma)}, \boldsymbol{\theta})} \to 0 \quad \text{or} \quad \to +\infty \quad \text{as} \quad (\boldsymbol{\theta}, \gamma) \to (\bar{\boldsymbol{\theta}}, 0),$$

where $\to 0$ corresponds to the inactive inequalities $g_{t,i}(\boldsymbol{x}_t^{\bar{\boldsymbol{\theta}}}, \boldsymbol{u}_t^{\bar{\boldsymbol{\theta}}}, \bar{\boldsymbol{\theta}}) < 0$ and $\to +\infty$ corresponds to the active inequalities $g_{t,i}(\boldsymbol{x}_t^{\bar{\boldsymbol{\theta}}}, \boldsymbol{u}_t^{\bar{\boldsymbol{\theta}}}, \bar{\boldsymbol{\theta}}) = 0$ ($v_{t,i}^{\boldsymbol{\theta}} > 0$ due to strict complementarity); and also

$$\frac{\gamma}{\left( g_{T,i}(\boldsymbol{x}_T^{(\boldsymbol{\theta}, \gamma)}, \boldsymbol{\theta}) \right)^2} = -\frac{v_{T,i}^{(\boldsymbol{\theta}, \gamma)}}{g_{T,i}(\boldsymbol{x}_T^{(\boldsymbol{\theta}, \gamma)}, \boldsymbol{\theta})} \to 0 \quad \text{or} \quad \to +\infty \quad \text{as} \quad (\boldsymbol{\theta}, \gamma) \to (\bar{\boldsymbol{\theta}}, 0).$$

where $\to 0$ corresponds to the inactive inequalities $g_{T,i}(\boldsymbol{x}_T^{\bar{\boldsymbol{\theta}}}, \bar{\boldsymbol{\theta}}) < 0$ and $\to +\infty$ corresponds to the active inequalities $g_{T,i}(\boldsymbol{x}_T^{\bar{\boldsymbol{\theta}}}, \bar{\boldsymbol{\theta}}) = 0$ ($v_{T,i}^{\boldsymbol{\theta}} > 0$ due to strict complementarity).

By extending the left side of (S.44) based on the facts (S.40) and (S.41), (S.46) immediately leads to

$$\lim_{k\to+\infty}\left(\sum_{t=0}^{T-1}\begin{bmatrix}\boldsymbol{x}_t^k\\\boldsymbol{u}_t^k\end{bmatrix}'\begin{bmatrix}\hat{L}_t^{xx}(\boldsymbol{\theta}^k,\gamma^k)&\hat{L}_t^{xu}(\boldsymbol{\theta}^k,\gamma^k)\\\hat{L}_t^{ux}(\boldsymbol{\theta}^k,\gamma^k)&\hat{L}_t^{uu}(\boldsymbol{\theta}^k,\gamma^k)\end{bmatrix}\begin{bmatrix}\boldsymbol{x}_t^k\\\boldsymbol{u}_t^k\end{bmatrix}+\boldsymbol{x}_T^{k\,\prime}\hat{L}_T^{xx}(\boldsymbol{\theta}^k,\gamma^k)\boldsymbol{x}_T^k\right)\to+\infty,\quad\text{(S.47)}$$

which obviously contradicts (S.44).

*Case 2:* $\|\mathrm{col}\,\{\boldsymbol{x}_{0:T}^*,\boldsymbol{u}_{0:T-1}^*\}\|=1$ and all of the following holds:

$$\bar{G}_t^x(\bar{\boldsymbol{\theta}},0)\boldsymbol{x}_t^*+\bar{G}_t^u(\bar{\boldsymbol{\theta}},0)\boldsymbol{u}_t^*=\boldsymbol{0}\quad\forall t\quad\text{and}\quad H_t^x(\bar{\boldsymbol{\theta}},0)\boldsymbol{x}_t^*+H_t^u(\bar{\boldsymbol{\theta}},0)\boldsymbol{u}_t^*=\boldsymbol{0}\quad\forall t$$
$$\text{and}\quad\bar{G}_T^x(\bar{\boldsymbol{\theta}},0)\boldsymbol{x}_T^*=\boldsymbol{0}\quad\text{and}\quad H_T^x(\bar{\boldsymbol{\theta}},0)\boldsymbol{x}_T^*=\boldsymbol{0}.\tag{S.48}$$

In this case, we have

$$\lim_{k\to+\infty}\left(\sum_{t=0}^{T-1}\begin{bmatrix}\boldsymbol{x}_t^k\\\boldsymbol{u}_t^k\end{bmatrix}'\begin{bmatrix}\hat{L}_t^{xx}(\boldsymbol{\theta}^k,\gamma^k)&\hat{L}_t^{xu}(\boldsymbol{\theta}^k,\gamma^k)\\\hat{L}_t^{ux}(\boldsymbol{\theta}^k,\gamma^k)&\hat{L}_t^{uu}(\boldsymbol{\theta}^k,\gamma^k)\end{bmatrix}\begin{bmatrix}\boldsymbol{x}_t^k\\\boldsymbol{u}_t^k\end{bmatrix}+\boldsymbol{x}_T^{k\,\prime}\hat{L}_T^{xx}(\boldsymbol{\theta}^k,\gamma^k)\boldsymbol{x}_T^k\right)$$
$$\geq\lim_{k\to+\infty}\left(\sum_{t=0}^{T-1}\begin{bmatrix}\boldsymbol{x}_t^k\\\boldsymbol{u}_t^k\end{bmatrix}'\begin{bmatrix}L_t^{xx}(\boldsymbol{\theta}^k,\gamma^k)&L_t^{xu}(\boldsymbol{\theta}^k,\gamma^k)\\L_t^{ux}(\boldsymbol{\theta}^k,\gamma^k)&\hat{L}_t^{uu}(\boldsymbol{\theta}^k,\gamma^k)\end{bmatrix}\begin{bmatrix}\boldsymbol{x}_t^k\\\boldsymbol{u}_t^k\end{bmatrix}+\boldsymbol{x}_T^{k\,\prime}L_T^{xx}(\boldsymbol{\theta}^k,\gamma^k)\boldsymbol{x}_T^k\right)$$
$$=\sum_{t=0}^{T-1}\begin{bmatrix}\boldsymbol{x}_t^*\\\boldsymbol{u}_t^*\end{bmatrix}'\begin{bmatrix}L_t^{xx}(\bar{\boldsymbol{\theta}},0)&L_t^{xu}(\bar{\boldsymbol{\theta}},0)\\L_t^{ux}(\bar{\boldsymbol{\theta}},0)&\hat{L}_t^{uu}(\bar{\boldsymbol{\theta}},0)\end{bmatrix}\begin{bmatrix}\boldsymbol{x}_t^*\\\boldsymbol{u}_t^*\end{bmatrix}+\boldsymbol{x}_T^{*\,\prime}L_T^{xx}(\bar{\boldsymbol{\theta}},0)\boldsymbol{x}_T^*>0.\quad\text{(S.49)}$$

Here, the first inequality is based on the fact that the residual term is always non-negative, i.e.,

$$\sum_{t=0}^{T-1}\left(\sum_{i=1}^{q_t}\frac{\gamma^k}{(g_{t,i}^k)^2}\Big(\frac{\partial g_{t,i}^k}{\partial\boldsymbol{x}_t}\boldsymbol{x}_t^k+\frac{\partial g_{t,i}^k}{\partial\boldsymbol{u}_t^k}\boldsymbol{u}_t^k\Big)^2+\sum_{i=1}^{s_t}\frac{1}{\gamma^k}\Big(\frac{\partial h_{t,i}^k}{\partial\boldsymbol{x}_t^k}\boldsymbol{x}_t^k+\frac{\partial h_{t,i}^k}{\partial\boldsymbol{u}_t^k}\boldsymbol{u}_t^k\Big)^2\right)$$
$$+\sum_{i=1}^{q_T}\frac{\gamma^k}{(g_{T,i}^k)^2}\Big(\frac{\partial g_{T,i}^k}{\partial\boldsymbol{x}_T}\boldsymbol{x}_T^k\Big)^2+\sum_{i=1}^{s_T}\frac{1}{\gamma^k}\Big(\frac{\partial h_{T,i}^k}{\partial\boldsymbol{x}_T}\boldsymbol{x}_T^k\Big)^2\geq0,\quad\text{(S.50)}$$

the last inequality is directly from the second-order condition in (S.6)-(S.7) in Lemma A.2. Obviously, (S.49) also contracts (S.44).

Combining the above two cases, we can conclude that for any $(\boldsymbol{\theta},\gamma)$ near $(\bar{\boldsymbol{\theta}},0)$ with $\gamma>0$, the trajectory $\boldsymbol{\xi}_{(\boldsymbol{\theta},\gamma)}$ to the unconstrained optimal control system $\Sigma(\boldsymbol{\theta},\gamma)$ satisfies both its PMP condition in (S.31) and the second-order condition in (S.42)-(S.43). Thus, one can assert that $\boldsymbol{\xi}_{(\boldsymbol{\theta},\gamma)}$ is a local isolated minimizing trajectory to $\Sigma(\boldsymbol{\theta},\gamma)$. This completes the proof of Claim (a) in Theorem 2.

## C.2 Proof of Claim (b)

Given that the conditions (i)-(iii) in Lemma 1 hold for $\Sigma(\bar{\boldsymbol{\theta}})$, we have the following conclusions:

(1) From Claim (a) and its proof, we know that for any $(\boldsymbol{\theta},\gamma)$ in the neighborhood of $(\bar{\boldsymbol{\theta}},0)$, there exists a unique once-continuously differentiable function $\left(\boldsymbol{\xi}_{(\boldsymbol{\theta},\gamma)},\boldsymbol{\lambda}_{0:T}^{(\boldsymbol{\theta},\gamma)},\boldsymbol{v}_{0:T}^{(\boldsymbol{\theta},\gamma)},\boldsymbol{w}_{0:T}^{(\boldsymbol{\theta},\gamma)}\right)$, which satisfies (S.32). Additionally provided $\gamma>0$, such $\boldsymbol{\xi}_{(\boldsymbol{\theta},\gamma)}$ is also a local isolated minimizing trajectory for the well-defined unconstrained optimal control system $\Sigma(\boldsymbol{\theta},\gamma)$.

(2) Additionally let $\gamma=0$ in (S.32), and (S.32) becomes the C-PMP condition for the constrained optimal control system $\Sigma(\boldsymbol{\theta})$. From Lemma 1, for any $\boldsymbol{\theta}$ near $\bar{\boldsymbol{\theta}}$, $\boldsymbol{\xi}_{\boldsymbol{\theta}}=\boldsymbol{\xi}_{(\boldsymbol{\theta},\gamma=0)}$ is a differentiable local isolated minimizing trajectory for $\Sigma(\boldsymbol{\theta})$, associated with the unique once-continuously differentiable function $\left(\boldsymbol{\lambda}_{1:T}^{(\boldsymbol{\theta},\gamma=0)},\boldsymbol{v}_{0:T}^{(\boldsymbol{\theta},\gamma=0)},\boldsymbol{w}_{0:T}^{(\boldsymbol{\theta},\gamma=0)}\right)$.

Therefore, due to the uniqueness and once-continuous differentiability of $\boldsymbol{\xi}_{(\boldsymbol{\theta},\gamma)}$ with respect to $(\boldsymbol{\theta},\gamma)$ near $(\bar{\boldsymbol{\theta}},0)$, one can obtain

$$\boldsymbol{\xi}_{(\boldsymbol{\theta},\gamma)}\to\boldsymbol{\xi}_{(\boldsymbol{\theta},0)}=\boldsymbol{\xi}_{\boldsymbol{\theta}}\quad\text{as}\quad\gamma\to0,\tag{S.51}$$

and

$$\frac{\partial \boldsymbol{\xi}_{(\boldsymbol{\theta},\gamma)}}{\partial \boldsymbol{\theta}} \to \frac{\partial \boldsymbol{\xi}_{(\boldsymbol{\theta},0)}}{\partial \boldsymbol{\theta}} = \frac{\boldsymbol{\xi}_{\boldsymbol{\theta}}}{\partial \boldsymbol{\theta}} \quad \text{as} \quad \gamma \to 0. \tag{S.52}$$

Here (S.51) is due to that $\boldsymbol{\xi}_{(\boldsymbol{\theta},\gamma)}$ is unique and continuous at $(\boldsymbol{\theta}, \gamma = 0)$, and (S.51) is because $\boldsymbol{\xi}_{(\boldsymbol{\theta},\gamma)}$ is unique and once-continuously differentiable at $(\boldsymbol{\theta}, \gamma = 0)$. This completes the proof of Claim (b) in Theorem 2.

## C.3 Proof of Claim (c)

For the unconstrained optimal control system $\boldsymbol{\Sigma}(\boldsymbol{\theta}, \gamma)$ with any $(\boldsymbol{\theta}, \gamma)$ near $(\bar{\boldsymbol{\theta}}, 0)$, $\gamma > 0$, in order to show that its trajectory derivative $\frac{\partial \boldsymbol{\xi}_{(\boldsymbol{\theta},\gamma)}}{\partial \boldsymbol{\theta}}$ is a globally unique minimizing trajectory to its corresponding auxiliary control system $\overline{\boldsymbol{\Sigma}}(\boldsymbol{\xi}_{(\boldsymbol{\theta},\gamma)})$, similarly to the claim of Theorem 1, we need to verify if the following three conditions hold for $\boldsymbol{\Sigma}(\boldsymbol{\theta}, \gamma)$ at $(\boldsymbol{\theta}, \gamma)$.

(i) The second-order condition holds for $\boldsymbol{\xi}_{(\boldsymbol{\theta},\gamma)}$ to be a local isolated minimizing trajectory for $\boldsymbol{\Sigma}(\boldsymbol{\theta}, \gamma)$. In fact, this has been proved in the proof of Claim (a).

(ii) The gradients of all binding constraints (i.e., all equality and active inequality constraints) are linearly independent at $\boldsymbol{\xi}_{(\boldsymbol{\theta},\gamma)}$. Since we do not have inequality constraints in $\boldsymbol{\Sigma}(\boldsymbol{\theta}, \gamma)$, we only need to show the gradients of the dynamics constraint are linearly independent at $\boldsymbol{\xi}_{(\boldsymbol{\theta},\gamma)}$. Specifically, we need to show that the following linear equations are independent

$$\boldsymbol{x}_{t+1} = F_t^x(\boldsymbol{\theta}, \gamma)\boldsymbol{x}_t + F_t^u(\boldsymbol{\theta}, \gamma)\boldsymbol{u}_t, \quad \text{and} \quad \boldsymbol{x}_0 = \boldsymbol{0}, \quad t = 0, 1, \cdots, T. \tag{S.53}$$

where the dependence $(\boldsymbol{\theta}, \gamma)$ means that the derivative matrices are evaluated at trajectory $\boldsymbol{\xi}_{(\boldsymbol{\theta},\gamma)}$, $\boldsymbol{x}_{0:T}$ and $\boldsymbol{u}_{0:T-1}$ here are variables. In fact, the above linear equations in (S.53) can be equivalently written as

$$\boldsymbol{F}_x \boldsymbol{x}_{1:T} + \boldsymbol{F}_u \boldsymbol{u}_{0:T-1} = \boldsymbol{0}, \tag{S.54}$$

with

$$\boldsymbol{F}_u = \begin{bmatrix} -F_0^u(\boldsymbol{\theta}, \gamma) & 0 & \cdots & 0 \\ 0 & -F_1^u(\boldsymbol{\theta}, \gamma) & \cdots & 0 \\ \vdots & \vdots & \ddots & \vdots \\ 0 & 0 & \cdots & -F_{T-1}^x(\boldsymbol{\theta}, \gamma) \end{bmatrix}, \tag{S.55}$$

and

$$\boldsymbol{F}_x = \begin{bmatrix} I & 0 & \cdots & 0 & 0 \\ -F_1^x(\boldsymbol{\theta}, \gamma) & I & \cdots & 0 & 0 \\ \vdots & \vdots & \ddots & \vdots & \vdots \\ 0 & 0 & \cdots & I & 0 \\ 0 & 0 & \cdots & -F_{T-1}^x(\boldsymbol{\theta}, \gamma) & I \end{bmatrix}. \tag{S.56}$$

Obviously, all rows in the concatenation matrix $[\boldsymbol{F}_u, \boldsymbol{F}_x]$ are linear-independent because $[\boldsymbol{F}_u, \boldsymbol{F}_x]$ is already in its the reduced echelon form and has full row rank. Thus, one can conclude that the linear equations in (S.53) are linearly independent.

(iii) Strict complementarity does not apply because there are no inequality constraints in $\boldsymbol{\Sigma}(\boldsymbol{\theta}, \gamma)$ at $(\boldsymbol{\theta}, \gamma)$.

With the above three conditions satisfied, by applying Theorem 1, we can conclude that $\frac{\partial \boldsymbol{\xi}_{(\boldsymbol{\theta},\gamma)}}{\partial \boldsymbol{\theta}}$ is a globally unique minimizing trajectory to the auxiliary control system $\overline{\boldsymbol{\Sigma}}(\boldsymbol{\xi}_{(\boldsymbol{\theta},\gamma)})$. This completes the Claim (c) in Theorem 2.

With the Claims (a), (b), and (c) proved, we have completed the proof of Theorem 2. $\qquad \square$

# D Proof of Theorem 3

We know from the proof of Claim (a) of Theorem 2 in Appendix C.1 that given the conditions in Theorem 3,

- for any $(\boldsymbol{\theta}, \gamma)$ in the neighborhood of $(\boldsymbol{\theta}^*, 0)$, there exists a unique once-continuously differentiable function $\left(\boldsymbol{\xi}_{(\boldsymbol{\theta}, \gamma)}, \boldsymbol{\lambda}_{0:T}^{(\boldsymbol{\theta}, \gamma)}, \boldsymbol{v}_{0:T}^{(\boldsymbol{\theta}, \gamma)}, \boldsymbol{w}_{0:T}^{(\boldsymbol{\theta}, \gamma)}\right)$, which satisfies (S.32), and

$$\left(\boldsymbol{\xi}_{(\boldsymbol{\theta}, \gamma)}, \boldsymbol{\lambda}_{0:T}^{(\boldsymbol{\theta}, \gamma)}, \boldsymbol{v}_{0:T}^{(\boldsymbol{\theta}, \gamma)}, \boldsymbol{w}_{0:T}^{(\boldsymbol{\theta}, \gamma)}\right) = \left(\boldsymbol{\xi}_{\boldsymbol{\theta}^*}, \boldsymbol{\lambda}_{1:T}^{\boldsymbol{\theta}^*}, \boldsymbol{v}_{0:T}^{\boldsymbol{\theta}^*}, \boldsymbol{w}_{0:T}^{\boldsymbol{\theta}^*}\right) \text{ when } (\boldsymbol{\theta}, \gamma) = (\boldsymbol{\theta}^*, 0);$$

- additionally, if all functions defining $\boldsymbol{\Sigma}(\boldsymbol{\theta})$ are *three-times* continuously differentiable, it immediately follows that $\boldsymbol{\xi}_{(\boldsymbol{\theta}, \gamma)}$ is then *twice* continuously differentiable near $(\boldsymbol{\theta}^*, 0)$. This is a direct result by applying the $C^k$ implicit function theorem [87], to the C-PMP condition (S.32) in the neighborhood of $(\boldsymbol{\theta}^*, 0)$.

- additionally provided $\gamma > 0$, such $\boldsymbol{\xi}_{(\boldsymbol{\theta}, \gamma)}$ is also a local isolated minimizing trajectory for the well-defined unconstrained optimal control system $\boldsymbol{\Sigma}(\boldsymbol{\theta}, \gamma)$ in Problem SB$(\boldsymbol{\theta}, \gamma)$.

Thus, in the following, we will ignore the computation process for obtaining $\boldsymbol{\xi}_{(\boldsymbol{\theta}, \gamma)}$ and simply view that $\boldsymbol{\xi}_{(\boldsymbol{\theta}, \gamma)}$ is the twice continuously differentiable function of $(\boldsymbol{\theta}, \gamma)$ near $(\boldsymbol{\theta}^*, 0)$ and $\boldsymbol{\xi}_{\boldsymbol{\theta}^*} = \boldsymbol{\xi}_{(\boldsymbol{\theta}=\boldsymbol{\theta}^*, \gamma=0)}$. The following proof of Theorem 3 follows the procedure of the general interior-point minimization methods, which are systematically studied in [48] (see Theorem 14, p. 80).

Recall the optimization in Problem SP$(\epsilon, \gamma)$, re-write it below for easy reference,

$$\boldsymbol{\theta}^*(\epsilon, \gamma) = \arg\min_{\boldsymbol{\theta}} \quad W(\boldsymbol{\theta}, \epsilon, \gamma) \tag{S.57}$$

with

$$W(\boldsymbol{\theta}, \epsilon, \gamma) = \ell(\boldsymbol{\xi}_{(\boldsymbol{\theta}, \gamma)}, \boldsymbol{\theta}) - \epsilon \sum_{i=1}^{l} \ln\left(-R_i(\boldsymbol{\xi}_{(\boldsymbol{\theta}, \gamma)}, \boldsymbol{\theta})\right) \tag{S.58}$$

Given in Theorem 3 that $\boldsymbol{\theta}^*$ satisfies the second-order sufficient condition for a local isolated minimizer to Problem P (recall the general second-order sufficient condition in Lemma A.1), one can say that there exists a multiplier $\boldsymbol{u}^* \in \mathbb{R}^l$ such that

$$\begin{aligned} \nabla L(\boldsymbol{\theta}^*, \boldsymbol{u}^*) &= \nabla\ell(\boldsymbol{\xi}_{\boldsymbol{\theta}^*}, \boldsymbol{\theta}^*) + \sum_{i=1}^{l} u_i^* \nabla R_i(\boldsymbol{\xi}_{\boldsymbol{\theta}^*}, \boldsymbol{\theta}^*) = \mathbf{0}, \\ u_i^* R_i(\boldsymbol{\xi}_{\boldsymbol{\theta}^*}, \boldsymbol{\theta}^*) &= 0, \quad i = 1, 2, ..., l, \\ R_i(\boldsymbol{\xi}_{\boldsymbol{\theta}^*}, \boldsymbol{\theta}^*) &\leq 0, \quad u_i^* \geq 0, \quad i = 1, 2, ..., l, \end{aligned} \tag{S.59}$$

with the Lagrangian defined as

$$L(\boldsymbol{\theta}, \boldsymbol{u}) = \ell(\boldsymbol{\xi}_{\boldsymbol{\theta}}, \boldsymbol{\theta}) + \sum_{i=1}^{l} u_i R_i(\boldsymbol{\xi}_{\boldsymbol{\theta}}, \boldsymbol{\theta}), \tag{S.60}$$

and further for any $\boldsymbol{\theta} \neq \mathbf{0}$ satisfying $\boldsymbol{\theta}' \nabla R_i(\boldsymbol{\xi}_{\boldsymbol{\theta}^*}, \boldsymbol{\theta}^*) = 0$ with $u_i^* > 0$ and $\boldsymbol{\theta}' \nabla R_i(\boldsymbol{\xi}_{\boldsymbol{\theta}^*}, \boldsymbol{\theta}^*) \leq 0$ with $u_i^* \geq 0$, it follows

$$\boldsymbol{\theta}' \nabla^2 L(\boldsymbol{\theta}^*, \boldsymbol{u}^*) \boldsymbol{\theta} > 0. \tag{S.61}$$

Here, $\nabla L$ and $\nabla^2 L$ denote the first- and second-order derivatives of $L$ with respect to $\boldsymbol{\theta}$, respectively; and $\boldsymbol{\xi}_{\boldsymbol{\theta}^*} = \boldsymbol{\xi}_{(\boldsymbol{\theta}=\boldsymbol{\theta}^*, \gamma=0)}$.

## D.1 Proof of Claim (a)

We modify the first two equations in (S.59) into

$$\begin{aligned} \nabla\ell\left(\boldsymbol{\xi}_{(\boldsymbol{\theta}^*(\epsilon, \gamma), \gamma)}, \boldsymbol{\theta}^*(\epsilon, \gamma)\right) + \sum_{i=1}^{l} u_i^*(\epsilon, \gamma) \nabla R_i\left(\boldsymbol{\xi}_{(\boldsymbol{\theta}^*(\epsilon, \gamma), \gamma)}, \boldsymbol{\theta}^*(\epsilon, \gamma)\right) &= \mathbf{0}, \\ u_i^*(\epsilon, \gamma) R_i\left(\boldsymbol{\xi}_{(\boldsymbol{\theta}^*(\epsilon, \gamma), \gamma)}, \boldsymbol{\theta}^*(\epsilon, \gamma)\right) &= -\epsilon, \quad i = 1, 2, ..., l, \end{aligned} \tag{S.62}$$

respectively, and consider both $\boldsymbol{\theta}^*(\epsilon, \gamma)$ and $\boldsymbol{u}^*(\epsilon, \gamma)$ are implicitly determined by $\epsilon$ and $\gamma$ through the above equations.

Look at (S.62) and note that when $\epsilon = 0$ and $\gamma = 0$, (S.62) is identical to the first two equations in (S.59). Given in Theorem 3 that all binding constraint gradients $\nabla R_i(\boldsymbol{\xi}_{\boldsymbol{\theta}^*}, \boldsymbol{\theta}^*)$ are linearly independent at $\boldsymbol{\theta}^*$ and the strict complementary holds at $\boldsymbol{\theta}^*$, similar to the proof of Theorem 2, one can apply the well-known implicit function theorem [75] to (S.62) in a neighborhood of $(\epsilon, \gamma) = (0, 0)$, leading to the following claim (i.e., the first-order sensitivity result in Theorem 14 in [48]):

*In a neighborhood of $(\epsilon, \gamma) = (0, 0)$, there exists a unique once continuously differentiable function $\big(\boldsymbol{\theta}^*(\epsilon, \gamma), \boldsymbol{u}^*(\epsilon, \gamma)\big)$, which satisfies (S.62) and $\big(\boldsymbol{\theta}^*(\epsilon, \gamma), \boldsymbol{u}^*(\epsilon, \gamma)\big) = (\boldsymbol{\theta}^*, \boldsymbol{u}^*)$ when $(\epsilon, \gamma) = (0, 0)$.*

Next, we show that the above $\boldsymbol{\theta}^*(\epsilon, \gamma)$ always respects the constraints $R_i\big(\boldsymbol{\xi}_{(\boldsymbol{\theta}^*(\epsilon, \gamma), \gamma)}, \boldsymbol{\theta}^*(\epsilon, \gamma)\big) < 0$, $i = 1, 2, ..., l$, for any small $\epsilon > 0$ and any small $\gamma > 0$, which is the second-part of Claim (a).

In fact, for any inactive constraint, $R_i(\boldsymbol{\xi}_{\boldsymbol{\theta}^*}, \boldsymbol{\theta}^*) < 0$, due to the continuity of $\big(\boldsymbol{\xi}_{(\boldsymbol{\theta}^*(\epsilon, \gamma), \gamma)}, \boldsymbol{\theta}^*(\epsilon, \gamma)\big)$, one has

$$R_i\big(\boldsymbol{\xi}_{(\boldsymbol{\theta}^*(\epsilon, \gamma), \gamma)}, \boldsymbol{\theta}^*(\epsilon, \gamma)\big) \to R_i(\boldsymbol{\xi}_{\boldsymbol{\theta}^*}, \boldsymbol{\theta}^*) < 0 \quad \text{as} \quad (\epsilon, \gamma) \to (0, 0), \tag{S.63}$$

and thus $R_i\big(\boldsymbol{\xi}_{(\boldsymbol{\theta}^*(\epsilon, \gamma), \gamma)}, \boldsymbol{\theta}^*(\epsilon, \gamma)\big) < 0$ for any small $\epsilon > 0$ and $\gamma > 0$. For any active constraint, $R_i(\boldsymbol{\xi}_{\boldsymbol{\theta}^*}, \boldsymbol{\theta}^*) = 0$, and since the corresponding $u_i^* > 0$ (due to the strict complementarity given in Theorem 3) and the continuity of $\boldsymbol{u}^*(\epsilon, \gamma)$, one has

$$u_i^*(\epsilon, \gamma) \to u_i^* > 0 \quad \text{as} \quad (\epsilon, \gamma) \to (0, 0), \tag{S.64}$$

and thus $u_i^*(\epsilon, \gamma) > 0$ for small $\epsilon > 0$ and consequently

$$R_i\big(\boldsymbol{\xi}_{(\boldsymbol{\theta}^*(\epsilon, \gamma), \gamma)}, \boldsymbol{\theta}^*(\epsilon, \gamma)\big) = -\frac{\epsilon}{u_i^*(\epsilon, \gamma)} < 0 \tag{S.65}$$

because of (S.62). Therefore, we have proved that for any small $\epsilon > 0$ and $\gamma > 0$, $\boldsymbol{\theta}^*(\epsilon, \gamma)$ always respect the constraints $R_i\big(\boldsymbol{\xi}_{(\boldsymbol{\theta}^*(\epsilon, \gamma), \gamma)}, \boldsymbol{\theta}^*(\epsilon, \gamma)\big) < 0$, $i = 1, 2, ..., l$. This prove the second part of Claim (a).

From now on, we show that the above $\boldsymbol{\theta}^*(\epsilon, \gamma)$ with any small $\epsilon > 0$ and $\gamma > 0$ also is a local isolated minimizer to the unconstrained optimization (S.57). From the last equation in (S.62), we solve

$$u_i^*(\epsilon, \gamma) = -\frac{\epsilon}{R_i\big(\boldsymbol{\xi}_{(\boldsymbol{\theta}^*(\epsilon, \gamma), \gamma)}, \boldsymbol{\theta}^*(\epsilon, \gamma)\big)}, \quad i = 1, 2, ..., l, \tag{S.66}$$

and substitute it to the first equation, yielding

$$\nabla \ell\big(\boldsymbol{\xi}_{(\boldsymbol{\theta}^*(\epsilon, \gamma), \gamma)}, \boldsymbol{\theta}^*(\epsilon, \gamma)\big) - \sum_{i=1}^{l} \frac{\epsilon}{R_i\big(\boldsymbol{\xi}_{(\boldsymbol{\theta}^*(\epsilon, \gamma), \gamma)}, \boldsymbol{\theta}^*(\epsilon, \gamma)\big)} \nabla R_i\big(\boldsymbol{\xi}_{(\boldsymbol{\theta}^*(\epsilon, \gamma), \gamma)}, \boldsymbol{\theta}^*(\epsilon, \gamma)\big) = \boldsymbol{0}. \tag{S.67}$$

One can find that the obtained equation in (S.67) is exactly the first-order optimality condition (KKT condition) for the unconstrained optimization in Problem $SP(\epsilon, \gamma)$ in (S.57), and this indicates that $\boldsymbol{\theta}^*(\epsilon, \gamma)$ satisfies the KKT condition for Problem $SP(\epsilon, \gamma)$. To further show that $\boldsymbol{\theta}^*(\epsilon, \gamma)$ is a local isolated minimizing solution to Problem $SP(\epsilon, \gamma)$ in (S.57), we only need to verify the second-order condition, that is, for any nonzero $\boldsymbol{\theta} \neq \boldsymbol{0}$,

$$\boldsymbol{\theta}'\Big(\nabla^2 W\big(\boldsymbol{\theta}^*(\epsilon, \gamma), \epsilon, \gamma\big)\Big)\boldsymbol{\theta} > 0, \tag{S.68}$$

for any small $\epsilon > 0$ and $\gamma > 0$, which will be proved next.

Proof by contradiction: suppose that the second-order condition (S.68) is false. Then, there must exist a sequence of $(\epsilon_k, \gamma_k) > 0$ and a sequence of $\boldsymbol{\theta}_k$ for $k = 1, 2, ...$ such that $(\epsilon_k, \gamma_k) \to (0, 0)$ and

$$\boldsymbol{\theta}_k'\big(\nabla^2 W\big(\boldsymbol{\theta}^*(\epsilon_k, \gamma_k), \epsilon_k, \gamma_k\big)\big)\boldsymbol{\theta}_k \leq 0. \tag{S.69}$$

as $k \to +\infty$. Without loss of generality, assume $\|\boldsymbol{\theta}_k\| = 1$ for all $k$. Select a convergent sub-sequence of $\boldsymbol{\theta}_k$, relabel the sequence $\boldsymbol{\theta}_k$ for convenience, and call the limit $\bar{\boldsymbol{\theta}}$, that is, $\boldsymbol{\theta}_k \to \bar{\boldsymbol{\theta}}$ and $(\epsilon_k, \gamma_k) \to 0$ as $k \to +\infty$. Then,

$$
\lim_{k \to +\infty} \boldsymbol{\theta}_k' \Big( \nabla^2 W \big( \boldsymbol{\theta}^*(\epsilon_k, \gamma_k), \epsilon_k, \gamma_k \big) \Big) \boldsymbol{\theta}_k
$$

$$
= \lim_{k \to +\infty} \left( \boldsymbol{\theta}_k' \Big( \nabla^2 L(\boldsymbol{\theta}^*(\epsilon_k, \gamma_k), \boldsymbol{u}^*(\epsilon_k, \gamma_k)) + \sum_{i=1}^{l} \frac{\epsilon_k}{(R_i(\epsilon_k, \gamma_k))^2} (\nabla R_i(\epsilon_k, \gamma_k) \nabla R_i(\epsilon_k, \gamma_k)') \Big) \boldsymbol{\theta}_k \right)
$$

$$
= \lim_{k \to +\infty} \left( \boldsymbol{\theta}_k' \Big( \nabla^2 L(\boldsymbol{\theta}^*(\epsilon_k, \gamma_k), \boldsymbol{u}^*(\epsilon_k, \gamma_k)) \Big) \boldsymbol{\theta}_k \right) + \lim_{k \to +\infty} \left( \sum_{i=1}^{l} \frac{\epsilon_k}{(R_i(\epsilon_k, \gamma_k))^2} (\nabla R_i(\epsilon_k, \gamma_k)' \boldsymbol{\theta}_k)^2 \right)
$$

$$
= \bar{\boldsymbol{\theta}}' \Big( \nabla^2 L(\boldsymbol{\theta}^*, \boldsymbol{u}^*) \Big) \bar{\boldsymbol{\theta}} + \lim_{k \to +\infty} \left( \sum_{i=1}^{l} \frac{\epsilon_k}{(R_i(\epsilon_k, \gamma_k))^2} (\nabla R_i(\epsilon_k, \gamma_k)' \boldsymbol{\theta}_k)^2 \right),
$$

(S.70)

where we write $R_i(\epsilon_k, \gamma_k) = R_i(\boldsymbol{\xi}_{(\boldsymbol{\theta}^*(\epsilon_k, \gamma_k), \gamma_k)}, \boldsymbol{\theta}^*(\epsilon_k, \gamma_k))$ and $\nabla R_i(\epsilon_k) = \nabla R_i(\boldsymbol{\xi}_{(\boldsymbol{\theta}^*(\epsilon_k, \gamma_k), \gamma_k)}, \boldsymbol{\theta}^*(\epsilon_k, \gamma_k))$ for notation convenience, and the last line is because $L(\boldsymbol{\theta}, \boldsymbol{u})$ in (S.60) is twice-continuously differentiable with respect to $(\boldsymbol{\theta}, \boldsymbol{u})$ near $(\boldsymbol{\theta}^*, \boldsymbol{u}^*)$, and $(\boldsymbol{\theta}^*(\epsilon, \gamma), \boldsymbol{u}^*(\epsilon, \gamma))$ is once-continuously differentiable with respect to $(\epsilon, \gamma)$ near $(0, 0)$. In (S.70), we consider two cases for $\bar{\boldsymbol{\theta}}$:

*Case 1:* $\|\bar{\boldsymbol{\theta}}\| = 1$ and there exists at least an active inequality constraint $R_i(\boldsymbol{\xi}_{\boldsymbol{\theta}^*}, \boldsymbol{\theta}^*) = 0$, such that $\bar{\boldsymbol{\theta}}' \nabla R_i(\boldsymbol{\xi}_{\boldsymbol{\theta}^*}, \boldsymbol{\theta}^*) \neq 0$. Then,

$$
\lim_{k \to +\infty} \left( \sum_{i=1}^{l} \frac{\epsilon_k}{(R_i(\epsilon_k, \gamma_k))^2} (\nabla R_i(\epsilon_k, \gamma_k)' \boldsymbol{\theta}_k)^2 \right)
$$

$$
= \lim_{k \to +\infty} \left( \sum_{i=1}^{l} \frac{-u_i(\epsilon_k, \gamma_k)}{R_i(\epsilon_k, \gamma_k)} (\nabla R_i(\epsilon_k, \gamma_k)' \boldsymbol{\theta}_k)^2 \right) = +\infty.
$$

(S.71)

This is because the following term corresponding to such active constraint has

$$
\lim_{k \to +\infty} \frac{-u_i(\epsilon_k, \gamma_k)}{R_i(\boldsymbol{\xi}_{(\boldsymbol{\theta}^*(\epsilon_k, \gamma_k), \gamma_k)}, \boldsymbol{\theta}^*(\epsilon_k, \gamma_k))} = +\infty.
$$

(S.72)

due to the strict complementarity given in Theorem 3. Therefore, (S.70) will have

$$
\lim_{k \to +\infty} \boldsymbol{\theta}_k' \Big( \nabla^2 W \big( \boldsymbol{\theta}^*(\epsilon_k, \gamma_k), \epsilon_k, \gamma_k \big) \Big) \boldsymbol{\theta}_k = +\infty,
$$

(S.73)

which contradicts the assumption in (S.69) in that

$$
\lim_{k \to +\infty} \boldsymbol{\theta}_k' \Big( \nabla^2 W \big( \boldsymbol{\theta}^*(\epsilon_k, \gamma_k), \epsilon_k, \gamma_k \big) \Big) \boldsymbol{\theta}_k \leq 0.
$$

(S.74)

*Case 2:* $\|\bar{\boldsymbol{\theta}}\| = 1$ and for any active constraint $R_i(\boldsymbol{\xi}_{\boldsymbol{\theta}^*}, \boldsymbol{\theta}^*) = 0$ (and $u_i^* > 0$ due to strict complementarity given in Theorem 3), $\bar{\boldsymbol{\theta}}' \nabla R_i(\boldsymbol{\xi}_{\boldsymbol{\theta}^*}, \boldsymbol{\theta}^*) = 0$. Then, from (S.70),

$$
\lim_{k \to +\infty} \boldsymbol{\theta}_k' \big( \nabla^2 W \big( \boldsymbol{\theta}^*(\epsilon_k, \gamma_k), \epsilon_k, \gamma_k \big) \big) \boldsymbol{\theta}_k \geq \bar{\boldsymbol{\theta}}' \Big( \nabla^2 L(\boldsymbol{\theta}^*, \boldsymbol{u}^*) \Big) \bar{\boldsymbol{\theta}} > 0,
$$

(S.75)

where the last inequality is because of the second-order condition in (S.61) satisfied for $\boldsymbol{\theta}^*$ given in Theorem 3. The obtained (S.75) also contradicts the assumption in (S.69).

Combining the above two cases, we can conclude that given any small $\epsilon > 0$ and $\gamma > 0$, $\boldsymbol{\theta}^*(\epsilon, \gamma)$ satisfies both the KKT condition (S.67) and the second-order condition (S.68) for $W(\boldsymbol{\theta}, \epsilon, \gamma)$. Thus, one can assert that $\boldsymbol{\theta}^*(\epsilon, \gamma)$ is a local isolated minimizer to the unconstrained optimization $W(\boldsymbol{\theta}, \epsilon, \gamma)$ in (S.57), i.e., Problem SP$(\epsilon, \gamma)$. This completes the proof of the Claim (a) in Theorem 3.

## D.2 Proof of Claim (b)

From the previous proof for Claim (a), we have the following conclusions: first, for any $(\epsilon, \gamma)$ in a neighborhood of $(0,0)$, there exists a *unique* once-continuously differentiable function $\big(\boldsymbol{\theta}^*(\epsilon, \gamma), \boldsymbol{u}^*(\epsilon, \gamma)\big)$, satisfying (S.62); second, additionally provided small $\epsilon > 0$ and $\gamma > 0$, such $\boldsymbol{\theta}^*(\epsilon, \gamma)$ is also a local isolated minimizer to the well-defined unconstrained minimization $W(\boldsymbol{\theta}, \epsilon, \gamma)$ in (S.57); and third, when $(\epsilon, \gamma) = (0,0)$, (S.62) becomes the KKT condition for Problem P, whose solution $(\boldsymbol{\theta}^*, \boldsymbol{u}^*)$ must satisfy. Therefore, due to the uniqueness and continuity of the function $(\boldsymbol{\theta}^*(\epsilon, \gamma), \boldsymbol{u}^*(\epsilon, \gamma))$ near $(\epsilon, \gamma) = (0,0)$, one can obtain

$$\boldsymbol{\theta}^*(\epsilon, \gamma) \rightarrow \boldsymbol{\theta}^*(0,0) = \boldsymbol{\theta}^*, \quad \text{as} \quad (\epsilon, \gamma) \rightarrow (0,0). \tag{S.76}$$

This completes the proof of Claim (b) in Theorem 3.

## D.3 Proof of Claim (c)

To prove Claim (c) in Theorem 3, we use the following facts: first, as proved in Claim (a), for any small $\epsilon > 0$ and $\gamma > 0$, $\boldsymbol{\theta}^*(\epsilon, \gamma)$ always respects the constraints $R_i(\boldsymbol{\xi}_{(\boldsymbol{\theta}^*(\epsilon,\gamma),\gamma)}, \boldsymbol{\theta}^*(\epsilon, \gamma)) < 0$, $i = 1, 2, ..., l$; second, as also proved in Claim (a), $\boldsymbol{\theta}^*(\epsilon, \gamma)$ is differentiable with respect to $(\epsilon, \gamma)$ near $(0,0)$ and $\boldsymbol{\theta}^*(\epsilon, \gamma) \rightarrow \boldsymbol{\theta}^*$ as $(\epsilon, \gamma) \rightarrow (0,0)$; and third, as proved in Theorem 2, $\boldsymbol{\xi}_{(\boldsymbol{\theta},\gamma)}$ is a differentiable function of $(\boldsymbol{\theta}, \gamma)$ near $(\boldsymbol{\theta}^*, 0)$. All these facts lead to that for small $\gamma > 0$, $R_i(\boldsymbol{\xi}_{(\boldsymbol{\theta},\gamma)}, \boldsymbol{\theta})$, $i = 1, 2, ..., l$, is also a continuous function of $\boldsymbol{\theta}$ near $\boldsymbol{\theta}^*(\epsilon, \gamma)$, and

$$R_i(\boldsymbol{\xi}_{(\boldsymbol{\theta},\gamma)}, \boldsymbol{\theta}) \rightarrow R_i\big(\boldsymbol{\xi}_{(\boldsymbol{\theta}^*(\epsilon,\gamma),\gamma)}, \boldsymbol{\theta}^*(\epsilon, \gamma)\big) < 0, \quad \text{as} \quad \boldsymbol{\theta} \rightarrow \boldsymbol{\theta}^*(\epsilon, \gamma), \quad \forall i = 1, 2, ..., l. \tag{S.77}$$

Thus $R_i(\boldsymbol{\xi}_{(\boldsymbol{\theta},\gamma)}, \boldsymbol{\theta}) < 0$, $i = 1, 2, ..., l$, holds for any $\boldsymbol{\theta}$ in a small neighborhood of $\boldsymbol{\theta}^*(\epsilon, \gamma)$ with small $\epsilon > 0$ and small $\gamma > 0$. This completes the proof of Claim (c) in Theorem 3.

With the above proofs for Claims (a), (b), and (c), we have completed the proof of Theorem 3. $\quad\square$

# E   Algorithms for Safe PDP

We have implemented Safe PDP in Python and made it as a stand-alone package with friendly interfaces. Please download at https://github.com/wanxinjin/Safe-PDP.

## E.1   Algorithm for Theorem 1

---

**Algorithm 1:** Solving $\frac{\partial \boldsymbol{\xi_\theta}}{\partial \boldsymbol{\theta}}$ by establishing auxiliary control system $\overline{\boldsymbol{\Sigma}}(\boldsymbol{\xi_\theta})$

---

**Input:** $\boldsymbol{\xi_\theta}$, with the costates $\boldsymbol{\lambda}_{1:T}^{\boldsymbol{\theta}}$ and multiplies $\boldsymbol{v}_{0:T}^{\boldsymbol{\theta}}$ and $\boldsymbol{w}_{0:T}^{\boldsymbol{\theta}}$ from solving Problem B($\boldsymbol{\theta}$)

  **def** Identify_Active_Inequality_Constraints (a small threshold $\delta > 0$):

   $\bar{\boldsymbol{g}}_t(\boldsymbol{x}_t, \boldsymbol{u}_t, \boldsymbol{\theta}) = \mathrm{col}\{g_{t,i} \,|\, g_{t,i}(\boldsymbol{x}_t^{\boldsymbol{\theta}}, \boldsymbol{u}_t^{\boldsymbol{\theta}}, \boldsymbol{\theta}) \geq -\delta, \ \ i = 1, 2, ..., q_t\}, \ \ t = 0, 1, ..., T{-}1;$

   $\bar{\boldsymbol{g}}_T(\boldsymbol{x}_T, \boldsymbol{\theta}) = \mathrm{col}\{g_{T,i} \,|\, g_{T,i}(\boldsymbol{x}_T^{\boldsymbol{\theta}}, \boldsymbol{\theta}) \geq -\delta, \ \ i = 1, 2, ..., q_T\};$

   **Return:** $\bar{\boldsymbol{g}}_t(\boldsymbol{x}_t, \boldsymbol{u}_t, \boldsymbol{\theta})$ and $\bar{\boldsymbol{g}}_T(\boldsymbol{x}_T, \boldsymbol{\theta})$

  Compute the derivative matrices $L_t^{xx}, L_t^{xu}, L_t^{uu}, L_t^{x\theta}, L_t^{u\theta}, L_T^{xx}, L_T^{x\theta}, F_t^x, F_t^u, F_t^\theta, H_t^x,$
  $H_t^u, H_t^\theta, H_T^x, H_T^\theta, \bar{G}_t^x, \bar{G}_t^u, \bar{G}_t^\theta, \bar{G}_T^x, \bar{G}_T^\theta$ to establish $\overline{\boldsymbol{\Sigma}}(\boldsymbol{\xi_\theta})$ in (3);

  **def** Equality_Constrained_LQR_ Solver ( $\overline{\boldsymbol{\Sigma}}(\boldsymbol{\xi_\theta})$ ):

   Implementation of the equality constrained LQR algorithm in [80];

   **Return:** $\{X_{0:T}^{\boldsymbol{\theta}}, U_{0:T-1}^{\boldsymbol{\theta}}\}$

**Return:** $\frac{\partial \boldsymbol{\xi_\theta}}{\partial \boldsymbol{\theta}} = \{X_{0:T}^{\boldsymbol{\theta}}, U_{0:T-1}^{\boldsymbol{\theta}}\}$

---

Note that $\boldsymbol{\lambda}_{1:T}^{\boldsymbol{\theta}}$, $\boldsymbol{v}_{0:T}^{\boldsymbol{\theta}}$, and $\boldsymbol{w}_{0:T}^{\boldsymbol{\theta}}$ are normally the by-product outputs of an optimal control solver [77], and can also be obtained by solving a linear equation of C-PMP (S.5) given $\boldsymbol{\xi_\theta}$, as done in [71]. Also note that the threshold $\delta$ to determine the active inequality constraints can be set according to the accuracy of the solver; in our experiments, we use $\delta = 10^{-3}$.

## E.2   Algorithm for Theorem 2

---

**Algorithm 2:** Safe unconstrained approximations for $\boldsymbol{\xi_\theta}$ and $\frac{\partial \boldsymbol{\xi_\theta}}{\partial \boldsymbol{\theta}}$

---

**Input:** The constrained optimal control system $\boldsymbol{\Sigma}(\boldsymbol{\theta})$ and a choice of small $\gamma > 0$

  Convert $\boldsymbol{\Sigma}(\boldsymbol{\theta})$ to an unconstrained optimal control system $\boldsymbol{\Sigma}(\boldsymbol{\theta}, \gamma)$ in (5) by adding all
  constraints in $\boldsymbol{\Sigma}(\boldsymbol{\theta})$ to its control cost through barrier functions with barrier parameter $\gamma$;

  `/* Below is an implmentation of uncsontrained PDP [71]          */`
  **def** Optimal_Control_Solver ( $\boldsymbol{\Sigma}(\boldsymbol{\theta}, \gamma)$ ):

   Implementation of any trajectory optimization algorithms, such as iLQR [81] and
   DDP [82], or use any optimal control solver [77];

   **Return:** $\boldsymbol{\xi}_{(\boldsymbol{\theta}, \gamma)}$

  Use $\boldsymbol{\xi}_{(\boldsymbol{\theta}, \gamma)}$ to compute the derivative matrices $\hat{L}_t^{xu}, \hat{L}_t^{uu}, \hat{L}_t^{x\theta}, \hat{L}_t^{u\theta}, \hat{L}_T^{xx}, \hat{L}_T^{x\theta}, F_t^x, F_t^u,$
  $F_t^\theta$ to establish the auxiliary control system $\overline{\boldsymbol{\Sigma}}(\boldsymbol{\xi}_{(\boldsymbol{\theta}, \gamma)})$ in (S.78) ;

  **def**  LQR_ Solver ( $\overline{\boldsymbol{\Sigma}}(\boldsymbol{\xi}_{(\boldsymbol{\theta}, \gamma)})$ ):

   Implementation of any LQR algorithm such as Lemma 2 in [71];

   **Return:** $\left\{X_{0:T}^{(\boldsymbol{\theta}, \gamma)}, U_{0:T-1}^{(\boldsymbol{\theta}, \gamma)}\right\} = \frac{\partial \boldsymbol{\xi}_{(\boldsymbol{\theta}, \gamma)}}{\partial \boldsymbol{\theta}}$

**Return:** $\boldsymbol{\xi}_{(\boldsymbol{\theta}, \gamma)}$ and $\frac{\partial \boldsymbol{\xi}_{(\boldsymbol{\theta}, \gamma)}}{\partial \boldsymbol{\theta}}$

---

Note that the auxiliary control system $\overline{\Sigma}(\boldsymbol{\xi}_{(\boldsymbol{\theta},\gamma)})$ corresponding to $\Sigma(\boldsymbol{\theta},\gamma)$ is

$$
\overline{\Sigma}(\boldsymbol{\xi}_{(\boldsymbol{\theta},\gamma)}):
\begin{aligned}
\textit{control cost} \quad \bar{W} &= \mathrm{Tr}\sum_{t=0}^{T-1}\left(\frac{1}{2}\begin{bmatrix}X_t\\U_t\end{bmatrix}'\begin{bmatrix}\hat{L}_t^{xx} & \hat{L}_t^{xu}\\\hat{L}_t^{ux} & \hat{L}_t^{uu}\end{bmatrix}\begin{bmatrix}X_t\\U_t\end{bmatrix}+\begin{bmatrix}\hat{L}_t^{x\theta}\\\hat{L}_t^{u\theta}\end{bmatrix}'\begin{bmatrix}X_t\\U_t\end{bmatrix}\right)\\
&\quad + \mathrm{Tr}\left(\frac{1}{2}X_T'\hat{L}_T^{xx}X_T+(\hat{L}_T^{x\theta})'X_T\right)\\
\textit{subject to}&\\
\textit{dynamics}\quad &X_{t+1}=F_t^x X_t+F_t^u U_t+F_t^\theta \quad\text{with}\quad X_0=X_0^{\boldsymbol{\theta}}.
\end{aligned}
\tag{S.78}
$$

Here, $\hat{L}_t$, $t=0,1,...,T-1$, and $\hat{L}_T$ are the Hamiltonian, defined in (S.31), for the unconstrained optimal control system $\Sigma(\boldsymbol{\theta},\gamma)$. The derivative (coefficient) matrices $\hat{L}_t^{xu}$, $\hat{L}_t^{uu}$, $\hat{L}_t^{x\theta}$, $\hat{L}_t^{u\theta}$, $\hat{L}_T^{xx}$, $\hat{L}_T^{x\theta}$, $F_t^x$, $F_t^u$, $F_t^\theta$ in (S.78) are defined in the similar notation convention as in (3).

### E.3 Algorithm for Theorem 3

---

**Algorithm 3:** Safe PDP to solve Problem P

---

**Input:** Small barrier parameter $\epsilon>0$ for outer-level and $\gamma>0$ for inner-level, initialization $\boldsymbol{\theta}_0$

    /* Convert inner-level into its safe unconstrained approximation */

1    Convert the inner-level constrained control system $\Sigma(\boldsymbol{\theta})$ in (1) into an unconstrained system $\Sigma(\boldsymbol{\theta},\gamma)$ in (5) by adding all constraints to its control cost through barrier functions with the inner-level barrier parameter $\gamma>0$;

    /* Convert outer-level into its safe unconstrained approximation */

2    Convert the constrained Problem P to an unconstrained Problem $\mathrm{SP}(\epsilon,\gamma)$ by adding all task constraints $R_i$ to the task loss through barrier functions with the outer-level barrier parameter $\epsilon>0$;

    /* Gradient-based update for $\boldsymbol{\theta}$                          */

    **for** $k=0,1,2,\cdots$ **do**

3        Apply Algorithm 2 to the inner-level safe unconstrained approximation system $\Sigma(\boldsymbol{\theta}_k,\gamma)$ to compute $\boldsymbol{\xi}_{(\boldsymbol{\theta}_k,\gamma)}$ and $\frac{\partial\boldsymbol{\xi}_{(\boldsymbol{\theta},\gamma)}}{\partial\boldsymbol{\theta}}|_{\boldsymbol{\theta}_k}$;

4        For the outer-level unconstrained Problem $\mathrm{SP}(\epsilon,\gamma)$ with objective function $W(\boldsymbol{\theta},\epsilon,\gamma)=\ell(\boldsymbol{\xi}_{(\boldsymbol{\theta},\gamma)},\boldsymbol{\theta})-\epsilon\sum_{i=1}^{l}\ln\left(-R_i(\boldsymbol{\xi}_{(\boldsymbol{\theta},\gamma)},\boldsymbol{\theta})\right)$, compute the partial gradients $\frac{\partial W}{\partial\boldsymbol{\theta}}|_{\boldsymbol{\theta}_k}$ and $\frac{\partial W}{\partial\boldsymbol{\xi}_{(\boldsymbol{\theta},\gamma)}}|_{\boldsymbol{\xi}_{(\boldsymbol{\theta}_k,\gamma)}}$;

5        Apply the chain rule to obtain the gradient of the outer-level unconstrained objective $W(\boldsymbol{\theta},\epsilon,\gamma)$ with respect to $\boldsymbol{\theta}$, i.e., $\frac{dW}{d\boldsymbol{\theta}}|_{\boldsymbol{\theta}_k}=\frac{\partial W}{\partial\boldsymbol{\theta}}|_{\boldsymbol{\theta}_k}+\frac{\partial W}{\partial\boldsymbol{\xi}_{(\boldsymbol{\theta},\gamma)}}|_{\boldsymbol{\xi}_{(\boldsymbol{\theta}_k,\gamma)}}\frac{\partial\boldsymbol{\xi}_{(\boldsymbol{\theta},\gamma)}}{\partial\boldsymbol{\theta}}|_{\boldsymbol{\theta}_k}$;

6        Gradient-based update: $\boldsymbol{\theta}_{k+1}=\boldsymbol{\theta}_k-\boldsymbol{\eta}\frac{dW}{d\boldsymbol{\theta}}|_{\boldsymbol{\theta}_k}$ with $\boldsymbol{\eta}$ being the learning rate;

    **end**

---

**Return:** $\boldsymbol{\theta}^*(\epsilon,\gamma)$ for the given barrier parameters $\epsilon>0$ and $\gamma>0$

---

Note that after obtaining $\boldsymbol{\theta}^*(\epsilon,\gamma)$ from Algorithm 3, one can sequentially refine $\boldsymbol{\theta}^*(\epsilon,\gamma)$ by choosing a sequence of $\{(\epsilon,\gamma)\}$ such that $(\epsilon,\gamma)\to(0,0)$.

Also note that in the case where the original inner-level control system $\Sigma(\boldsymbol{\theta})$ is already unconstrained, such as in the applications of safe policy optimization and safe motion planning in Section 7, please modify lines 1 and 3 in Algorithm 3 and just compute exact $\boldsymbol{\xi}_{\boldsymbol{\theta}_k}$ and $\frac{\partial\boldsymbol{\xi}_{\boldsymbol{\theta}}}{\partial\boldsymbol{\theta}}|_{\boldsymbol{\theta}_k}$ by following PDP [71].

# F  Experiment Details

The proposed Safe PDP has been evaluated in different simulated systems in Table 1, where each system has the immediate constraints $g(\theta_{\mathrm{cstr}})$ on both its state and input during the entire time horizon (around $T = 50$). For the detailed description and physical models of each system in Table 1, we refer the reader to [71] and its accompanying codes. We have developed the Python code of Safe PDP as a stand-alone package, which can be accessed at `https://github.com/wanxinjin/Safe-PDP`.

## F.1  Safe Policy Optimization

In this experiment, we apply Safe PDP to perform safe policy optimization for the systems in Table 1. In Problem P, we set the details of $\Sigma(\theta)$ as (8), where the dynamics $f$ is learned from demonstrations in Section F.3, and the policy $u_t = \pi(x_t, \theta)$ is represented using a neural network (NN) with $\theta$ the NN parameter. In our experiment, we have used a fully-connected feedforward NN to represent the policy; the number of nodes in the NN is $n-n-m$ (meaning that the input layer has $n$ nodes, hidden layer $n$ nodes, and output layer $m$ nodes, with $n$ and $m$ the dimensions of the system state and input, respectively); and the activation function of the NN is $\tanh$. In Problem P, set the task loss $\ell(\xi_\theta, \theta)$ as the control cost $J(\theta_{\mathrm{obj}})$, and set the task constraints $R_i(\xi_\theta, \theta)$ as the system constraints $g(\theta_{\mathrm{cstr}})$, both in Table 1, with both $\theta_{\mathrm{obj}}$ and $\theta_{\mathrm{cstr}}$ known.

Note that since the parameterized $\Sigma(\theta)$ in (8) does not include the control cost $J$ anymore, solving Problem B($\theta$) for $\xi_\theta$ becomes a simple integration of (8) from $t = 0$ to $T-1$, and the auxiliary control system $\overline{\Sigma}(\xi_\theta)$ in (3) to compute $\frac{\partial \xi_\theta}{\partial \theta}$ is simplified to a feedback control system [71] below:

$$\overline{\Sigma}(\xi_\theta): \quad \begin{array}{ll} \text{dynamics:} & X_{t+1}^\theta = F_t^x X_t^\theta + F_t^u U_t^\theta \quad \text{with} \quad X_0 = \mathbf{0}, \\ \text{control policy:} & U_t^\theta = U_t^x X_t^\theta + U_t^e. \end{array} \tag{S.79}$$

Here, $U_t^x = \frac{\partial \pi_t}{\partial x_t^\theta}$ and $U_t^e = \frac{\partial \pi_t}{\partial \theta}$. Integrating (S.79) from $t = 0$ to $T-1$ leads to $\{X_{0:T}^\theta, U_{0:T-1}^\theta\} = \frac{\partial \xi_\theta}{\partial \theta}$.

In our experiments, in order to make sure the initial NN policy is feasible (safe), we initialize the NN policy using supervised learning from a random demonstration trajectory (note that this demonstration trajectory does not have to be optimal but only to be feasible/safe). For each system in Table 1, we apply Safe PDP Algorithm 3 to optimize the NN policy, and the complete experiment results are shown in Fig. S1-S3. More discussions about how to give a safe initialization are presented in Appendix G.5

For each system, at a fixed barrier parameter $\epsilon$, we have applied the vanilla gradient descent to solve Problem SP($\epsilon$) with the step size (learning rate $\eta$ in Algorithm 3) set around $10^{-3}$. We plot the task loss (i.e., control cost) $\ell(\xi_\theta, \theta)$ versus iteration of the gradient descent in the first panel in Fig. S1-S3, where we only show the results for outer-level barrier parameter $\epsilon$ taking from $10^0$ to $10^{-4}$ because the NN policy has already achieved a good convergence when $\epsilon \leq 10^{-2}$. As shown in the first panel in Fig. S1-S3, for each system, the policy achieves a good convergence after a small number of iterations for each $\epsilon$, and obtains a good convergence after $\epsilon \leq 10^{-2}$.

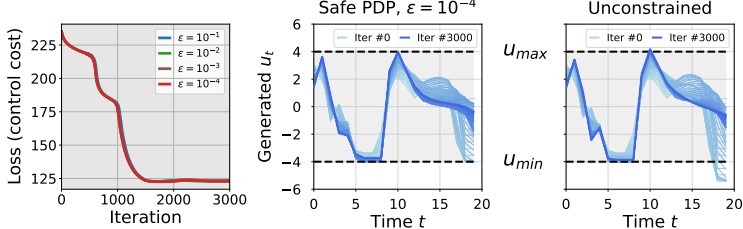

Figure S1: Safe neural policy optimization for cartpole. The first panel plots the loss (control cost) versus gradient-descent iteration under different outer-level barrier parameter $\epsilon$; the second panel plots all intermediate control trajectories generated by the NN policy during the entire gradient-descent iteration ($\epsilon = 10^{-4}$); and the third panel plots all intermediate control trajectories generated by the NN policy for the unconstrained policy optimization under the same experimental conditions. The system constraints are also marked using black dashed lines in the second and third panels.

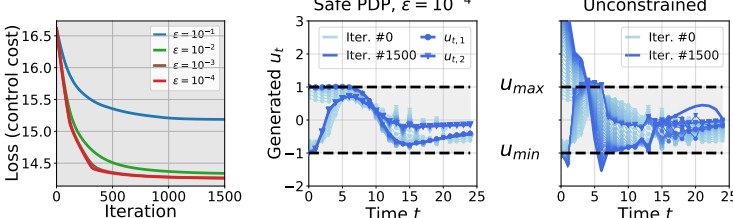

Figure S2: Safe neural policy optimization for robot arm. The first panel plots the loss (control cost) versus gradient-descent iteration under different outer-level barrier parameter $\epsilon$; the second panel plots all intermediate control trajectories generated by the NN policy during the entire gradient-descent iteration ($\epsilon = 10^{-4}$); and the third panel plots all intermediate control trajectories generated by the NN policy for the unconstrained policy optimization under the same experimental conditions. The system constraints are also marked using black dashed lines in the second and third panels.

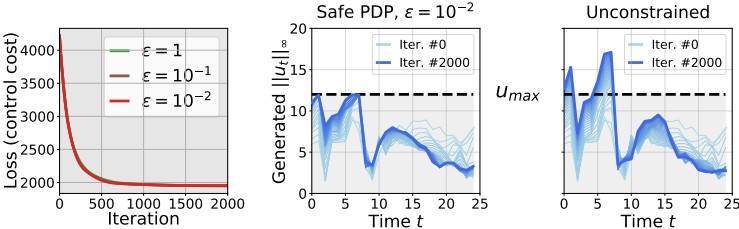

Figure S3: Safe neural policy optimization for 6-DoF maneuvering quadrotor. The first panel plots the loss (control cost) versus gradient-descent iteration under different outer-level barrier parameter $\epsilon$; the second panel plots all intermediate control trajectories generated by the NN policy during the entire gradient-descent iteration ($\epsilon = 10^{-2}$); and the third panel plots all intermediate control trajectories generated by the NN policy for the unconstrained policy optimization under the same experimental conditions. The system constraints are also marked using black dashed lines in the second and third panels.

In order to show the constraint satisfaction of Safe PDP throughout the entire policy optimization process, in the second panel in Fig. S1-S3, respectively, we plot all intermediate control trajectories generated from the NN policy throughout the entire gradient-descent iteration of Safe PDP, as shown from the light to dark blue. From the second panel in Fig. S1-S3, we note that throughout the optimization process, the NN policy is guaranteed safe, meaning that the generated trajectory will never violate the constraints. Under the same experimental conditions (NN configuration, policy initialization, learning rate), we also compare with the unconstrained policy optimization and plot its results in the third panel in Fig. S1-S3, respectively. By comparing the results between Safe PDP and unconstrained policy optimization, we can confirm that Safe PDP enables to achieve an optimal policy while guaranteeing that any intermediate policy throughout optimization is safe, as asserted in Theorem 3.

We have provided the video demonstrations for the above safe policy optimization using Safe PDP; please visit `https://youtu.be/sC81qc2ip8U`. The codes for all experiments here can be downloaded at `https://github.com/wanxinjin/Safe-PDP`.

## F.2 Safe Motion Planning

In this experiment, we apply Safe PDP to solve the safe motion planning problem for the systems in Table 1. In Problem P, we set the details of $\boldsymbol{\Sigma}(\boldsymbol{\theta})$ as follows,

$$\boldsymbol{\Sigma}(\boldsymbol{\theta}): \quad \begin{array}{ll} \textit{dynamics:} & \boldsymbol{x}_{t+1} = \boldsymbol{f}(\boldsymbol{x}_t, \boldsymbol{u}_t) \quad \text{with} \quad \boldsymbol{x}_0, \\ \textit{control input:} & \boldsymbol{u}_t = \boldsymbol{u}(t, \boldsymbol{\theta}), \end{array} \tag{S.80}$$

where the dynamics $\boldsymbol{f}$ is learned from demonstrations in Section F.3, and we parameterize the control

input function $\boldsymbol{u}_t = \boldsymbol{u}(t, \boldsymbol{\theta})$ using the Lagrangian polynomial [83] as follows,

$$\boldsymbol{u}(t, \boldsymbol{\theta}) = \sum_{i=0}^{N} \boldsymbol{u}_i b_i(t) \qquad \text{with} \qquad b_i(t) = \prod_{0 \le j \le N, j \ne i} \frac{t - t_j}{t_i - t_j}. \tag{S.81}$$

Here, $b_i(t)$ is called Lagrange basis, and the policy parameter $\boldsymbol{\theta}$ is defined as

$$\boldsymbol{\theta} = [\boldsymbol{u}_0, \cdots, \boldsymbol{u}_N]' \in \mathbb{R}^{m(N+1)}, \tag{S.82}$$

which is the vector of the pivots of the Lagrange polynomial. The benefit of the above parameterization is that the trajectory of system states, which results from integrating (S.80) given the input polynomial trajectory $\boldsymbol{u}_t = \boldsymbol{u}(t, \boldsymbol{\theta})$, is inherently smooth and dynamics-feasible. In our experiments, the degree $N$ of the Lagrange polynomial is set as $N = 10$. Also in Problem P, we set the task/planning loss $\ell(\boldsymbol{\xi}_{\boldsymbol{\theta}}, \boldsymbol{\theta})$ as the control cost $J(\boldsymbol{\theta}_{\text{obj}})$, and set the task constraints $R_i(\boldsymbol{\xi}_{\boldsymbol{\theta}}, \boldsymbol{\theta})$ as the system constraints $\boldsymbol{g}(\boldsymbol{\theta}_{\text{cstr}})$, both given in Table 1 with $\boldsymbol{\theta}_{\text{obj}}$ and $\boldsymbol{\theta}_{\text{cstr}}$ known.

Since the system $\boldsymbol{\Sigma}(\boldsymbol{\theta})$ in (S.80) now does not include the control cost $J$ anymore, solving Problem B($\boldsymbol{\theta}$) for $\boldsymbol{\xi}_{\boldsymbol{\theta}}$ becomes a simple integration of (S.80) from $t = 0$ to $T-1$, and the auxiliary control system $\overline{\boldsymbol{\Sigma}}(\boldsymbol{\xi}_{\boldsymbol{\theta}})$ in (3) to compute $\frac{\partial \boldsymbol{\xi}_{\boldsymbol{\theta}}}{\partial \boldsymbol{\theta}}$ is simplified to a feedback control system [71] below:

$$\overline{\boldsymbol{\Sigma}}(\boldsymbol{\xi}_{\boldsymbol{\theta}}) : \qquad \begin{aligned} \text{dynamics:} \quad & X_{t+1}^{\boldsymbol{\theta}} = F_t^x X_t^{\boldsymbol{\theta}} + F_t^u U_t^{\boldsymbol{\theta}} \quad \text{with} \quad X_0 = \mathbf{0}, \\ \text{control input:} \quad & U_t^{\boldsymbol{\theta}} = U_t^e, \end{aligned} \tag{S.83}$$

where $U_t^e = \frac{\partial \boldsymbol{\pi}_t}{\partial \boldsymbol{\theta}}$. Integrating (S.83) from $t = 0$ to $T - 1$ leads to $\{X_{0:T}^{\boldsymbol{\theta}}, U_{0:T-1}^{\boldsymbol{\theta}}\} = \frac{\partial \boldsymbol{\xi}_{\boldsymbol{\theta}}}{\partial \boldsymbol{\theta}}$.

For each system in Table 1, we apply Safe PDP Algorithm 3 to perform safe motion planning, and the complete experiment results are shown in Fig. S4-S6. For each system, at a fixed outer-level barrier parameter $\epsilon$, we have applied the vanilla gradient descent to solve Problem SP($\epsilon$) with the step size (learning rate $\eta$ in Algorithm 3) set to $10^{-2}$ or $10^{-1}$. We plot the planning loss $\ell(\boldsymbol{\xi}_{\boldsymbol{\theta}}, \boldsymbol{\theta})$ versus gradient descent iteration in Fig. S4a-S6a, respectively; here we only show the results for $\epsilon$ taking from $10^0$ to $10^{-2}$ because the trajectory has already achieved a good convergence when $\epsilon \le 10^{-2}$. As shown in Fig. S4a-S6a, for each system, the trajectory achieves a good convergence after a small number of iterations given a fixed $\epsilon$, and obtains a good convergence after $\epsilon \le 10^{-2}$.

To demonstrate that Safe PDP can guarantee safety throughout the optimization process, we plot all intermediate trajectories during the entire iteration of Safe PDP in S4b-S6b. At the same time, we also show the results of the ALTRO method [21], which is a state-of-the-art method for constrained trajectory optimization. By comparing the results in Fig. S4b-S6b, we can observe that Safe PDP enables to find the optimal trajectory while guaranteeing strict constraint satisfaction throughout the entire optimization process; while for ALTRO, although the trajectory satisfies the constraints at convergence, the intermediate trajectories during optimization may violate the constraints, making it not suitable to handle safety-critical motion planning tasks.

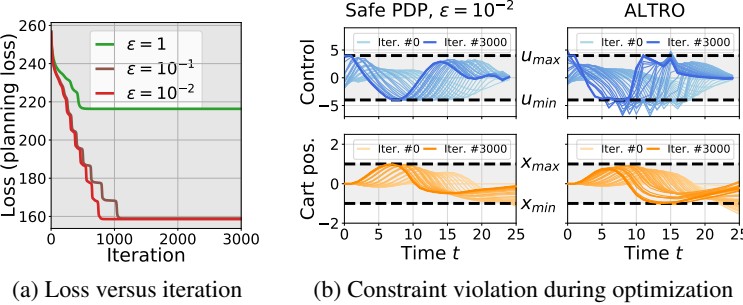

(a) Loss versus iteration      (b) Constraint violation during optimization

Figure S4: Safe motion planning for cartpole. (a) plots the loss (i.e., planning loss) versus gradient-descent iteration under different outer-level barrier parameter $\epsilon$. The left figure in (b) shows all intermediate trajectories during the entire iteration of Safe PDP ($\epsilon = 10^{-2}$), and the right figure in (b) shows all intermediate trajectories during the entire iteration of the ALTRO algorithm [21]. The state and control constraints are also marked in (b).

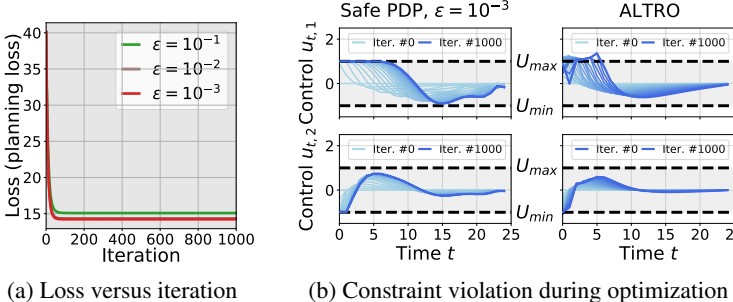

(a) Loss versus iteration      (b) Constraint violation during optimization

Figure S5: Safe motion planning for robot arm. (a) plots the loss (i.e., planning loss) versus gradient-descent iteration under different outer-level barrier parameter $\epsilon$. The left figure in (b) shows all intermediate trajectories during the entire iteration of Safe PDP ($\epsilon = 10^{-3}$), and the right figure in (b) shows all intermediate trajectories during the entire iteration of the ALTRO algorithm [21]. The control constraints are also marked in (b).

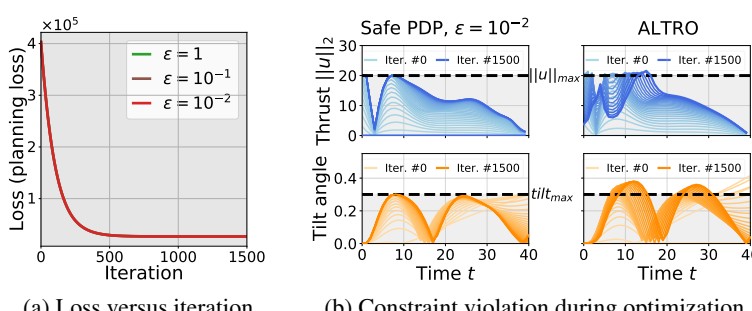

(a) Loss versus iteration      (b) Constraint violation during optimization

Figure S6: Safe motion planning for 6-DoF rocket powered landing. (a) plots the loss (i.e., planning loss) versus gradient-descent iteration under different outer-level barrier parameter $\epsilon$. The left figure in (b) shows all intermediate trajectories during the entire iteration of Safe PDP ($\epsilon = 10^{-2}$), and the right figure in (b) shows all intermediate trajectories during the entire iteration of the ALTRO algorithm [21]. The state and control constraints are also marked in (b).

We have provided the videos for the above safe motion planning using Safe PDP. Please visit the link https://youtu.be/vZVxgo30mDs. The codes for all experiments here can be downloaded at https://github.com/wanxinjin/Safe-PDP.

### F.3 Learning MPCs from Demonstrations

In this experiment, we apply Safe PDP to learn dynamics $\boldsymbol{f}$, constraints $\boldsymbol{g}$, or/and control cost $J$ for the systems in Table 1 from demonstration data. This type of problems has been extensively studied in system identification [88] (neural ODEs [89]), inverse optimal control (inverse reinforcement learning) [90–92], and learning from demonstrations [85, 93]. However, existing methods have the following two technical gaps; first, existing methods are typically developed without considering constraints; second, there are rarely the methods that are capable to *jointly* learn dynamics, state-input constraints, and control cost for continuous control systems. In this part, we will show that the above technical gaps can be addressed by Safe PDP. Throughout this part, we define the task loss in Problem P as the *reproducing loss* as below

$$\ell(\boldsymbol{\xi_\theta}, \boldsymbol{\theta}) = \|\boldsymbol{\xi}^{\text{demo}} - \boldsymbol{\xi_\theta}\|_2^2, \tag{S.84}$$

which is to penalize the distance between the reproduced trajectory $\boldsymbol{\xi_\theta}$ from the learnable model $\boldsymbol{\Sigma}(\boldsymbol{\theta})$ and the given demonstrations $\boldsymbol{\xi}^{\text{demo}}$, and there is no task constraint. For $\boldsymbol{\Sigma}(\boldsymbol{\theta})$ in Problem P, only the unknown parts (dynamics, control cost, or/and constraints) are parameterized by $\boldsymbol{\theta}$. Thus, by solving Problem P, we are able to learn $\boldsymbol{\Sigma}(\boldsymbol{\theta})$ such that its trajectory $\boldsymbol{\xi_\theta}$ has closest distance to the given demonstrations $\boldsymbol{\xi}^{\text{demo}}$.

In our experiment, when dealing with $\boldsymbol{\xi_\theta}$ and $\frac{\partial \boldsymbol{\xi_\theta}}{\partial \boldsymbol{\theta}}$ for $\boldsymbol{\Sigma}(\boldsymbol{\theta})$, we use the following three strategies.

- *Strategy (A)*: use an optimal control solver [77] to solve the constrained optimal control $\Sigma(\boldsymbol{\theta})$ in Problem B($\boldsymbol{\theta}$) to obtain $\boldsymbol{\xi_\theta}$, and use Theorem 1 (i.e., Algorithm 1) to obtain the trajectory derivative $\frac{\partial \boldsymbol{\xi_\theta}}{\partial \boldsymbol{\theta}}$ by solving $\overline{\Sigma}(\boldsymbol{\xi_\theta})$ in (3).

- *Strategy (B)*: by applying Theorem 2 (i.e., Algorithm 2), approximate $\boldsymbol{\xi_\theta}$ and $\frac{\partial \boldsymbol{\xi_\theta}}{\partial \boldsymbol{\theta}}$ using $\boldsymbol{\xi_{(\theta,\gamma)}}$ and $\frac{\partial \boldsymbol{\xi_{(\theta,\gamma)}}}{\partial \boldsymbol{\theta}}$, respectively, with a choice of small barrier parameter $\gamma > 0$.

- *Strategy (C)*: obtain $\boldsymbol{\xi_\theta}$ by solving $\Sigma(\boldsymbol{\theta})$ in Problem B($\boldsymbol{\theta}$) via a solver [77], and apply Theorem 2 (i.e., Algorithm 2) only to approximate $\frac{\partial \boldsymbol{\xi_\theta}}{\partial \boldsymbol{\theta}}$ using $\frac{\partial \boldsymbol{\xi_{(\theta,\gamma)}}}{\partial \boldsymbol{\theta}}$.

In the following experiments, when using Algorithm 2, we choose $\gamma = 10^{-2}$ because the corresponding inner-level approximations $\boldsymbol{\xi_{(\theta,\gamma)}}$ and $\frac{\partial \boldsymbol{\xi_{(\theta,\gamma)}}}{\partial \boldsymbol{\theta}}$ already achieve a good accuracy, as shown in previous experiments. In practice, the choice of $\gamma > 0$ is very flexible depending on the desired accuracy (a smaller $\gamma$ never hurts but would decrease the computational efficiency, as discussed in Appendix G.3).

### F.3.1 Learning Constrained ODEs from Demonstrations

In the first experiment, consider that in $\Sigma(\boldsymbol{\theta})$ the control cost $J$ is known while the dynamics (Ordinary Difference Equation) $\boldsymbol{f}(\boldsymbol{\theta}_{\mathrm{dyn}})$ and constraints $\boldsymbol{g}_t(\boldsymbol{\theta}_{\mathrm{cstr}})$ are unknown and parameterized, as in Table 1, $\boldsymbol{\theta} = \{\boldsymbol{\theta}_{\mathrm{dyn}}, \boldsymbol{\theta}_{\mathrm{cstr}}\}$. We aim to learn $\boldsymbol{\theta}$ from given demonstrations $\boldsymbol{\xi}^{\mathrm{demo}}$ by solving Problem P. Here, the demonstrations are generated by simulating the true system (i.e., expert) with $\boldsymbol{\theta}$ known; the demonstrations for each system contain two episode trajectories with time horizon around $T = 50$.

To solve Problem P, since there are no task constraints, we use the vanilla gradient descent to minimize the reproducing loss (S.84) while using the three strategies as mentioned above to handle the lower-level Problem B($\boldsymbol{\theta}$). The initial condition for the gradient descent is given randomly, and the learning rate for the gradient descent is set as $10^{-5}$. The complete results for all systems in Table 1 are given in Fig. S7.

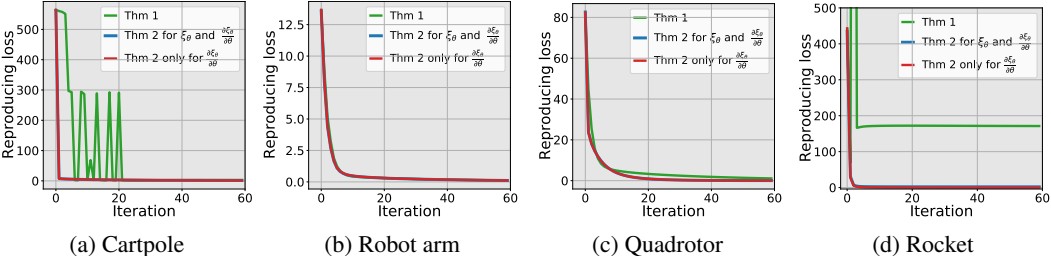

(a) Cartpole        (b) Robot arm        (c) Quadrotor        (d) Rocket

Figure S7: Learning both dynamics and constraints from demonstrations.

Fig. S7a-S7d plot the reproducing loss (S.84) versus gradient-descent iteration. The results show that for Strategies (B) and (C) (in blue and red, respectively), the reproducing loss (S.84) is quickly covering to zeros, indicating that the dynamics and constraints are successfully learned to reproduce the demonstrations. However, we also note that Strategy (A) (in green) suffers from some numerical instability, and this will be discussed later.

### F.3.2 Jointly Learning Dynamics, Constraints, and Control Cost from Demonstrations

In the second experiment, suppose in all systems in Table 1, the control cost $J(\boldsymbol{\theta}_{\mathrm{cost}})$, dynamics $\boldsymbol{f}(\boldsymbol{\theta}_{\mathrm{dyn}})$, and state and input constraints $\boldsymbol{g}_t(\boldsymbol{\theta}_{\mathrm{cstr}})$ are all unknown and parameterized as in Table 1. We aim to jointly learn $\boldsymbol{\theta} = \{\boldsymbol{\theta}_{\mathrm{cost}}, \boldsymbol{\theta}_{\mathrm{dyn}}, \boldsymbol{\theta}_{\mathrm{cstr}}\}$ from given demonstrations $\boldsymbol{\xi}^{\mathrm{demo}}$ by solving Problem P. Here, the demonstrations are generated by simulating the system (i.e., expert) with $\boldsymbol{\theta}$ known, the demonstrations for each system contain two episode trajectories for each system with time horizon around $T = 50$.

To solve Problem P, since there is no task constraints, we use the vanilla gradient descent to minimize the reproducing loss (S.84) while using the three strategies as mentioned above to handle the lower-level Problem B($\boldsymbol{\theta}$). The initial condition for the gradient descent is given randomly, and the learning

rate for the gradient-descent is set as $10^{-5}$. The complete results for all systems in Table 1 are given in Fig. S8 (also see Fig. 4a-4d in the primary text of the paper).

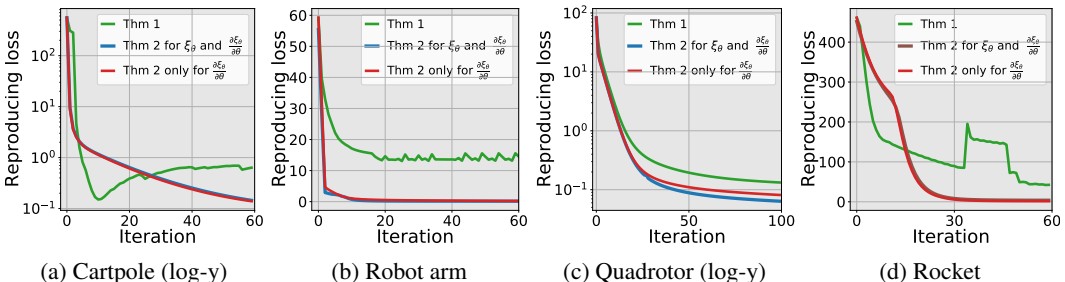

|  (a) Cartpole (log-y) | (b) Robot arm | (c) Quadrotor (log-y) | (d) Rocket |

Figure S8: Jointly learning dynamics, constraints, and control cost from demonstrations.

Fig. S8a - Fig. S8d plot the reproducing loss (S.84) versus gradient-descent iteration. The results show that for Strategies (B) and (C) (in blue and red, respectively), the reproducing loss (S.84) is quickly covering to zeros, indicating that the dynamics, constraints, and control cost function are successfully learned to reproduce the demonstrations. However, we also note that Strategy (A) (in green) suffers from some numerical instability, which will be discussed below.

We have provided some videos for the above learning MPCs from demonstrations using Safe PDP. Please visit the link `https://youtu.be/OBiLYYlWi98`. The codes for all experiments here can be downloaded at `https://github.com/wanxinjin/Safe-PDP`. [1]

**Why implementation of Theorem 1 is not numerically stable?**    In both Fig. S7 and S8, we have noted that Strategy (A) suffers from some numerical instability, and this is due to the following reasons. First, Theorem 1 requires to accurately detect the inactive/active inequalities (i.e., whether an inequality constraint is zero or not), which is always difficult accurately due to computational error (in our experiments, we detect the active constraints by applying a brutal threshold, as described in Algorithm 1). Second, although the differentiability of $\xi_\theta$ holds at the local neighborhood of $\theta$, $\xi_\theta$ might be extremely discontinuous due to the 'jumping switch' of the active and inactive inequality constraints for the large range of $\theta$; thus, such non-smoothness will deteriorate the decrease of loss between iterations.

**Why implementation of Theorem 2 is more numerically stable?**    Theorem 2 has perfectly addressed the above numerical issues of Theorem 1. Specifically, first, there is no need to distinguish the active and inactive inequality constraints in Theorem 2; and second, in Theorem 2, by adding all constraints to the control cost function, it introduces the 'softness' of the hard constraints and potentially eliminates the discontinuous 'jumping switch' between inactive and active inequalities over a large range of $\theta$, enabling a stable decrease of loss when applying gradient descent.

---

[1]All experiments in this paper have been performed on a personal computer with 3.5 GHz Dual-Core Intel Core i7 and macOS Big Sur system.

# G  Further Discussion

In this section, we will provide further experiments and discussion on the performance of Safe PDP.

## G.1  Comparison Between Safe PDP and PDP

In this part, we compare Safe PDP and non-safe PDP [71] to show the performance trade-offs between the constraint enforcement of Safe PDP and its resulting computational expense. We use the example of learning MPCs from expert demonstrations for the cartpole system (in Table 1) to show this, and the experiment settings are the same with Appendix F.3. The comparison results between Safe PDP and PDP are given in the following Table S1.

Table S1: Performance comparison between Safe PDP and PDP

| Methods | Loss at convergence | Timing for Forward Pass | Timing for Backward Pass | Learning constraints? | Constraint Guaranteed? |
|---------|---------------------|-------------------------|--------------------------|----------------------|------------------------|
| PDP | 524.02 | 0.10s | 0.046s | No | No |
| Safe PDP | 7.42 | 0.21s | 0.042s | Yes | Yes |

Based on the results in Table S1, we have the following comments and analysis.

(1) We note that Safe PDP achieves lower training loss. This is because compared to PDP, Safe PDP has introduced the inductive bias of constraints within its model architecture, making it more suited to learn from demonstrations which are the results of a constrained control system (expert). In this sense, Safe PDP architecture (with an inductive bias of constraints) can be thought of as having more expressive power than PDP architecture for the above experiments.

(2) For Safe PDP, its ability to learn and guarantee constraints comes at the cost of lower computational efficiency in the forward pass, as shown in the second column in Table S1. Even though Safe PDP handles constraint enforcement by adding them to the control cost using barrier functions, solving the resulting unconstrained approximation still needs more time than solving the unconstrained PDP. This could be because the added log barrier terms can increase the complex/stiff curvature of the cost/loss landscape, thus taking longer to find the minimizer. Further discussion about how barrier parameter influences the computational efficiency of the forward pass will be given in Appendix G.3.

(3) The running time for the backward pass is almost the same for both PDP and Safe PDP because both methods are solving an unconstrained LQR problem (auxiliary control system) of the same size (see Theorem 2), which can be very efficient based on Riccati equation.

## G.2  Strategies to Accelerate Forward Pass of Safe PDP

In the previous experiments in Appendix F, we have used an NLP solver to solve the trajectory optimization (optimal control) in the forward pass. Since the solver blindly treats an optimal control problem as a general non-linear program without leveraging the (sparse) structures in an optimal control problem. Thus, solving the long-horizon trajectory optimization is not very efficient. To accelerate long-horizon trajectory optimization, one can use plenty of strategies, as described below.

- To solve a long-horizon optimal control problem, one effective method is to scale a (continuous) long time horizon into a smaller one (like a unit) by applying a time-warping function to the system dynamics and control cost [85]. After discretizing and solving this short-horizon optimal control problem, re-scale the obtained optimal trajectory back. This time-scaling strategy is common in many commercial optimal control solvers, such as GPOPS [94].

- There are also the 'warm-up' tricks to accelerate the trajectory optimization in the forward pass of Safe PDP. For example, one can initialize the trajectory at the next iteration using the result of the previous iteration.

- One also can use a coarse-to-fine hierarchical strategy to solve long-horizon trajectory optimization. For example, given a long-time horizon optimal control system, first, discretize the trajectory with larger granularity and solve for a coarse-resolution optimal trajectory;

then use the coarse trajectory as initial conditions to solve the trajectory optimization with fine granular discretization.

As an additional experiment based on cartpole system (in Table 1), we tested and compared the above three strategies for accelerating the forward pass of Safe PDP. The timing for each strategy is given in the following Table S2. Here, $t_f$ is the continuous-time horizon of the cartpole system, $\Delta$ is the discretization interval, and the discrete-time horizon is $T = t_f/\Delta$.

Table S2: Running time for different strategies in accelerating the forward pass of Safe PDP

| Strategies | $t_f = 2$s, $\Delta = 0.1$s, $T = t_f/\Delta = 20$ | $t_f = 6$s, $\Delta = 0.1$s, $T = t_f/\Delta = 60$ | $t_f = 10$s, $\Delta = 0.1$s, $T = t_f/\Delta = 100$ | $t_f = 20$s, $\Delta = 0.1$s, $T = t_f/\Delta = 200$ |
|---|---|---|---|---|
| Plain NPL solver | 0.082s | 0.202s | 0.491s | 1.743s |
| Time scaling | 0.014s | 0.033s | 0.055s | 0.083s |
| Warm start | 0.055s | 0.095s | 0.108s | 0.224s |
| Hierarchical | 0.021s | 0.055s | 0.074s | 0.133s |

From the results in Table S2, one can see that time-scaling is the most effective way among others to accelerate long-horizon trajectory optimization. Of course, one can combine some of the above strategies to further improve the running performance of the forward pass of Safe PDP.

Additionally, one can also use iLQR [81] and DDP [82] to solve optimal control problems. iLQR can be viewed as the one-and-half-order method—linearizing dynamics and quadratizing cost function. DDP is a second-order method — quadractizing both dynamics and cost function. Both methods solve a local bellman equation to generate the update of the control sequence. But without coding optimization, both methods are slower than the commercial NPL solver (e.g., CasADi [77]). Some ongoing works are trying to take advantage of GPUs for accelerating trajectory optimization, which is also our future research.

### G.3   Trade-off Between Accuracy and Efficiency using Barrier Penalties

In the paper, we have provided both theoretical guarantees (see Theorem 2 and Theorem 3) and empirical experiments (see Fig. 1, Fig. 2a and 2c, Fig. 3a and 3c, and Fig. 4) for the relationship between the accuracy of a solution to an unconstrained approximation and the choice of the barrier parameter. This subsection further investigates the trade-off between accuracy and computational efficiency under different choices of the barrier parameter.

In the experiment below (based on the cartpole system in Table 1), by choosing different barrier parameters $\gamma$ in the forward pass of Safe PDP, we show the accuracy of the trajectory $\boldsymbol{\xi}(\gamma)$ solved from an unconstrained approximation system $\boldsymbol{\Sigma}(\gamma)$ and the corresponding computation time. The results are presented in Table S3.

Table S3: Accuracy of the trajectory $\boldsymbol{\xi}(\gamma)$ from the unconstrained approximation system $\boldsymbol{\Sigma}(\gamma)$ and its computation time with different choices of barrier parameter $\gamma$

| | choice of $\gamma$ | | | | | |
|---|---|---|---|---|---|---|
| | 1 | $10^{-1}$ | $10^{-2}$ | $10^{-3}$ | $10^{-4}$ | $10^{-5}$ |
| Accuracy of $\boldsymbol{\xi}(\gamma)$ in percentage: $\frac{\|\boldsymbol{\xi}(\gamma)-\boldsymbol{\xi}^*\|_2}{\|\boldsymbol{\xi}^*\|_2} \times 100\%$ [1] | 51.9% | 12.2% | 1.6% | 0.18% | 0.018% | 0.0002% |
| Timing for computing $\boldsymbol{\xi}(\gamma)$ | 0.023s | 0.033s | 0.035s | 0.040s | 0.038s | 0.047s |

[1] Note that in the above table, $\boldsymbol{\xi}^*$ is the ground-truth solution obtained from solving the original constrained trajectory optimization, and the computation time for such a constrained trajectory optimization is 0.062s.

We have the following comments on the above results in Table S3.

- First, the results in the first row of Table S3 show that a smaller barrier parameter leads to higher accuracy of the approximation solution. This again confirms the theoretical guarantee

in Theorem 2 (Claim (b)). The results here are also consistent with the ones in Fig. 1 in the paper.

- Second, the second row of Table S3 shows that a smaller barrier parameter, however, increases the computation time for solving the unconstrained approximation optimization. This could be because using a small barrier parameter, the added barrier terms can increase the complex/stiff curvature of the cost/loss landscape, thus taking Safe PDP longer to find the minimizer. Despite this, the time needed for finding a minimizer is still lower than directly solving a constrained trajectory optimization in the above experiment.

- Third, if one still wants to further increase the computation efficiency of Safe PDP, we have provided some strategies to achieve so, including "time scaling," "warm start," and "coarse-to-fine." Please check the Appendix G.2 for more detailed descriptions and corresponding experiment results.

In summary, we have shown that higher accuracy of the unconstrained approximation solution can be achieved using a smaller barrier parameter, while a smaller barrier parameter would increase the computation time for finding the approximation solution. In practice, one would likely choose an appropriate barrier parameter to balance the trade-off between accuracy and computational efficiency. Also, there are multiple strategies available to increase the computational efficiency of Safe PDP, as discussed in the Appendix G.2.

### G.4  Learning MPCs from Non-Optimal Demonstrations

In the application of learning MPCs (including objective, dynamics, constraints), given non-optimal demonstrations, Safe PDP can still learn an MPC such that the trajectory reproduced by the learned MPC has the closest discrepancy to the given non-optimal demonstrations (e.g., when the task loss is defined as $l_2$ norm between the reproduced trajectory and demonstrations). As an illustrative example, the following Table S4 shows learning an MPC from a sub-optimal demonstration for the cartpole system.

Table S4: Safe PDP for learning MPCs from non-optimal demonstrations

|  | Number of iterations | | | | | | | |
|---|---|---|---|---|---|---|---|---|
|  | 0 | 10 | 20 | 50 | 100 | 150 | 200 | 1000 |
| loss with optimal demo | 779.986 | 2.206 | 1.481 | 0.832 | 0.641 | 0.620 | 0.611 | 0.232 |
| loss with non-optimal demo | 1126.820 | 18.975 | 17.771 | 15.602 | 13.690 | 12.469 | 11.620 | 10.923 |

As shown in S4, the only difference between learning from optimal and non-optimal demonstrations is that the converged loss for the non-optimal demonstrations is relatively higher than for the optimal ones. This is because, for non-optimal demonstrations, there might not necessarily exist an exact MPC model in the parameterized model space which perfectly corresponds to the given demonstration. In such a case, however, Safe PDP can still find the best model in the parametrized model space such that its reproduced trajectory has a minimal distance to the given non-optimal demonstrations. For the extended research of the generalization ability of the learned MPCs from the non-optimal demonstrations, please refer to [85].

### G.5  Limitation of Safe PDP

Safe PDP requires a safe (feasible) initialization such that the log-barrier-based objectives (cost or task) are well-defined. While this requirement can be restrictive in some cases, we have the following empirical experiences on how to provide safe initialization for different types of problems.

- In safe policy optimization, one could first use supervised learning to learn a safe policy from some safe trajectories/demonstrations (which could not necessarily be optimal). Then, use the learned safe policy to initialize Safe PDP. We have used this strategy in our previous experiments in Appendix F.1.

- In safe motion planning, one could arbitrarily provide a safe trajectory (not necessarily optimal) to initialize Safe PDP. We have used this strategy in the previous experiments in Appendix F.2.

- In learning MPCs from demonstrations (Appendix F.3), the goal includes learning constraint models, and there is no such requirement.

Also, Safe PDP cannot apply to robust learning and control tasks. The goal of robust learning and control concerns achieving or maintaining good performance (such as stability or optimality) in the case of the worst disturbance or attacks to a system. Methods for handling those types of problems, such as robust control and differential game, have been well-developed in both control and machine learning communities. On the other hand, Safe PDP only focuses on guaranteeing the satisfaction of inequality constraints throughout a learning or control process, and such constraints are defined on the system states and inputs.