# OpenReview forum: "Safe Pontryagin Differentiable Programming"
_NeurIPS.cc/2021/Conference — NeurIPS 2021 Poster_

### Official Review · Reviewer_FvjM · 2021-07-06

**Rating:** 6
**Confidence:** 4

**Summary:**

In order to satisfy both immediate and long-term safety constraints, this paper proposes Safe Pontryagin Differentiable Programming (Safe PDP). The proposed method is based on interior-point methods, and it enables us to incorporate heterogeneous types of constraints. The advantages of Safe PDP lie both theoretically and empirically. In the experiments, the authors demonstrated that their proposed method is better than unconstrained methods (in one experiment, a baseline is ALTRO, though) in toy problems.



**Ethical Concerns:**

I don't have any ethical concerns.

**Limitations And Societal Impact:**

I think limitations have been adequately addressed.

**Main Review:**

This paper extends PDP (Jin et al., 2020) from unconstrained to constrained learning and control problems. The proposed Safe PDP algorithm is a natural extension of the original PDP method, and I find it nice in terms of good theoretical properties. In my opinion, however, this paper is hard to understand and fails to claim the authors' contribution in a proper way.


First, the empirical experiments are not well conducted. Except for Problem 2 (where Safe PDP is compared with ALTRO), the authors simply compared their proposed method with unconstrained methods. What the authors should have claimed is that Safe PDP is better than other "safe" baselines (in terms of constraints' satisfaction, sample complexity, etc). If I understand correctly, the empirical advantage of Safe PDP is that 1) multiple immediate/long-term safety constraints can be handled, 2) safety constraints are strictly guaranteed, and 3) conservativeness can be tuned (i.e., performance should be better than other too conservative baselines). In short, "Paper Contributions" mentioned in Section 1.2 are not well discussed in the empirical experiments. I would like the authors to reconsider what kinds of experiments are needed to support their claims.


Second, the advantages of the proposed method have not fully discussed in a convincing way. In lines 86 - 88, there is the following statement "The above two safe control frameworks favorably consider the pure state constraints and cannot be readily extended to other constraints, such as constraints defined on both state and input, long-term cumulative constraints defined on the system trajectory level, etc". But, is this really true? As far as I know, there is a bunch of literature dealing with safety function (S \times A \rightarrow R) which is defined for state-action pairs. And, regarding lines 70 - 71, there are also several papers that deal with immediate constraint(s) and guarantee safety at every time step as being represented by

- Turchetta, Matteo, Felix Berkenkamp, and Andreas Krause. "Safe exploration in finite Markov decision processes with gaussian processes." Advances in Neural Information Processing Systems. 2016.
- Wachi, Akifumi, and Yanan Sui. "Safe reinforcement learning in constrained Markov decision processes." International Conference on Machine Learning. 2020.

I feel this paper has many questionable statements in an attempt to make itself look good.


Third, it is unclear how this paper is related to offline RL literature. In Problem III (lines 344 - 363), the dynamics are learned from true system expert demonstration. I personally think that demonstrations are not usually such ideal ones, and the authors should have made the problem settings consistent with typical offline RL studies. I think this paper should not totally ignore the offline RL pieces of literature if a problem like Problem III is solved in experiments. As a notable reference, please see

- Le, Hoang, Cameron Voloshin, and Yisong Yue. "Batch policy learning under constraints." *International Conference on Machine Learning*. 2019.


Minor Comments

- I think lines 1099 - 1104 ("Why implementation of Theorem 2 is more numerically stable?" in the Appendix) should be in the main paper (I understand that page limit is very severe, though).
- In RL experiments, the constraints in terms of actions are often implemented by clipping. I wonder what happens if the action is clipped in Figure 2(b).


Questions

- Safe PDP is totally dependent on the assumption of differentiability. How do you justify that the assumptions hold in a real environment? For example, how do you deal with constraints with steep changes?

- What happens if the dynamics are learned from imperfect/noisy/non-expert demonstration?

- Is it possible to conduct a theoretical analysis regarding sample complexity?

**Time Spent Reviewing:**

5 hours

---

> ### Author Response · Authors · 2021-08-10
> **Our response to Reviewer FvjM's comments**
>
> We thank the reviewer for the time and effort on reviewing our paper.
>
> **Comments:** *"First, the empirical experiments are not well conducted. Except for Problem 2 (where Safe PDP is compared with ALTRO), the authors simply compared their proposed method with unconstrained methods. ..... I would like the authors to reconsider what kinds of experiments are needed to support their claims."*
>
> **Our response:**
> We would like to point out that ALTRO is a constrained trajectory optimization method (Augmented Lagrangian based). In this response, we have made a great effort to add the following new experiments to support the theoretical contributions of the Safe PDP paper. Specifically,
>    - In response to Reviewer v3kP (see Table 1), we have compared Safe PDP against non-safe PDP to show the tradeoff of Safe PDP due to constraint enforcement.
>    - In response to Reviewer Xjxt (see Table 2), we have added additional experiments to show different strategies for accelerating the forward pass of Safe PDP.
>    - In response to Reviewer u2xh (see Table 3), we have added the comparison between Safe PDP and CasADi [R1], a state-of-the-art interior-point based solver, to compare the performance in terms of computational cost and constraint violation when solving safe motion planning.
>    - In response to Reviewer u2xh (see Table 5), we have compared Safe PDP against a state-of-the-art sensitivity analysis algorithm [R2], and also against Differentiable MPC [R3], to show the performance for differentiating an optimal control system with respect to parameters. Also, see Table 4 for the running performance comparison between directly solving a constrained optimal control versus solving its unconstrained log-barrier approximation.
>    - In response to Reviewer FvjM (see Table 6 and Table 7), we have added additional experiments to show that Safe PDP can handle non-optimal demonstrations and has small sample complexity.
>
> All the above new experiments will be added to the final paper.
>
> ---
>
> **Comments:** *"Second, the advantages of the proposed method have not fully discussed in a convincing way. ..."*
>
> **Our response:**
> The "two safe control frameworks" in lines 86-88 refer to the reachability methods and control barrier functions. Both frameworks are originally designed for the constraints of states instead of states and inputs. To our best knowledge, applying those frameworks to address state-input constraints is rare if possible. We will revise some overclaims in the related work (e.g., in lines 70-71) in our final paper. We will also cite and discuss more related literature as the reviewer has suggested.
>
> ---
>
> **Comments:** *"Third, it is unclear how this paper is related to offline RL literature...."*
>
> **Our response:**
> We would think that Problem III (learning MPC from demonstrations) is more suited to the topic of `inverse problem’, e.g., inverse optimal control, inverse reinforcement learning, or learning objective from demonstrations, since they mainly focus on learning the aspects of the cost function, dynamics, constraints from the demonstrations. Offline RL, instead, only focuses on learning policies from recorded data. In our final version, we will add brief comments on offline RL and will cite the offline RL work suggested by the reviewer.
>
> ---
> **Comments:** *"Minor Comments: I think lines 1099 - 1104 ("Why implementation of Theorem 2 is more numerically stable?" in the Appendix) should be in the main paper (I understand that page limit is very severe, though)."*
>
> **Our response:**
> Yes, we will move such comments from the Appendix to primary text in the final version.
>
> ---
>
> **Comments:** *"In RL experiments, the constraints in terms of actions are often implemented by clipping. wonder what happens if the action is clipped in Figure 2(b)."*
>
> **Our response:**
> If one clips the control inputs in the right panel of Fig. 2(b) (obtained from unconstrained policy optimization), although satisfying constraints, the resulting control cost in fact is much higher than that of Safe PDP. This means that by simply clipping the control inputs, the manipulated control policy is not optimal and even not stable anymore.  For example, in the quadrotor experiment, simply clipping the input to satisfy the constraint will lead to the crash of the quadrotor.
>
> ---
> **Comments:** *"Safe PDP is totally dependent on the assumption of differentiability. How do you justify that the assumptions hold in a real environment? For example, how do you deal with constraints with steep changes?"*
>
>
> **Our response:**
> In this paper, a differentiability assumption is posed on mathematical models (such as the neural networks). We can use the differentiable models to construct a non-smooth system, while still requiring each component of the system, e.g., dynamics, objective, constraint, to be differentiable. One example is contact dynamics in robotics. The contact dynamics is formulated by the complementarity constraints and the constraint function itself is differentiable, but the generated robot behavior is non-smooth, sometimes even resulting in impulse inputs. In this paper, the proposed Safe PDP only requires that each component, i.e., dynamics, objective, and constraints, of a control system be differentiable, and not requiring the resulting trajectory (behavior) of the system to be differentiable. Safe PDP provides a systematic framework to perform learning or control of a constrained optimal control system.
>
> ---
> **Comments:** *"What happens if the dynamics are learned from imperfect/noisy/non-expert demonstration?"*
>
> **Our response:**
> Given non-optimal demonstrations, Safe PDP can still learn a MPC model (including objective, dynamics, constraints) such that the trajectory generated by this learned MPC model has the closest distance to the given non-optimal demonstrations (provided that the loss is defined as l2 norm between reproduced trajectory and demonstrations). As an illustrative example, the following table shows the results of learning MPC from a sub-optimal demonstration in Cartpole environment.
>
> ***Table 6: Safe PDP for learning MPCs from non-optimal demonstrations.***
>
> |Number of iterations|	#0|	#10|	#20	|#50|	#100|	#150|	#200|	#1000|
> |----------- | ----------- |----------- | ----------- |----------- | ----------- |----------- | ----------- |----------- |
> |Loss with optimal demo|	779.986|	2.206|	1.481|	0.832|	0.641|	0.620|	0.611|	0.232|
> |Loss with non-optimal demo|	1126.820|	18.975	|17.771	|15.602	|13.690	|12.469	|11.620	|10.923|
> ---
>
> As shown in the above results, the only difference between optimal and non-optimal demos is that the converged loss for the non-optimal demos is a relatively higher than the optimal one. This is because, for non-optimal demos, there might not necessarily exist an exact MPC model in the parameterized model space which perfectly corresponds to the given demo. However, Safe PDP can always find the best model in the parametrized model space such that its produced trajectory has a minimal distance to the given demo.
>
> ----
> **Comments:** *"Is it possible to conduct a theoretical analysis regarding sample complexity?"*
>
> **Our response:**
> Yes, it is possible to provide a theoretical analysis regarding the sample complexity. Here, we first give an example (in the below table) to show that Safe PDP can successfully learn a deterministic MPC only from a single optimal trajectory.
>
> ***Table 7: Safe PDP for learning MPCs from a single demonstration.***
>
> |Number of iterations|	#0|	#50	|#100|	#500|	#1000|
> |----------- | ----------- |----------- | ----------- |----------- | ----------- |
> Parameter error $norm(\boldsymbol{\theta}-\boldsymbol{\theta^*})$ ($\boldsymbol{\theta^*}$ is the true parameter)|	17.0|	0.791|	0.711|	0.298|	0.093|
> ---
>
> As shown in the above table, as the iteration proceeds, the learned parameter $\boldsymbol{\theta}$ converges to the true MPC parameter $\boldsymbol{\theta^*}$, indicating the success for learning/identifying the MPC model. In fact, one can follow the analysis in [R6] (see the reference below) to quantitively obtain a necessary lower bound for successfully learning a deterministic MPC. Simple tuition underpinning such analysis is that it converts the learning problem into the problem of solving a set of non-linear equations for the unknown parameters. Thus, if the number of equations induced by the training trajectory data is larger than the number of learnable parameters, it is possible to successfully learn a deterministic MPC model (without over-parameterization).  In our final paper, we will also give a similar analysis in the appendix.
>
> Reference:
>
> [R1] Andersson, J. A., Gillis, J., Horn, G., Rawlings, J. B., & Diehl, M. (2019). CasADi: a software framework for nonlinear optimization and optimal control. Mathematical Programming Computation, 11(1), 1-36.
>
> [R2] Andersson, J. A., & Rawlings, J. B. (2018). Sensitivity analysis for nonlinear programming in CasADi. IFAC-PapersOnLine, 51(20), 331-336.
>
> [R3] Amos, B., Rodriguez, I. D. J., Sacks, J., Boots, B., & Kolter, J. Z., Differentiable MPC for End-to-end Planning and Control. In NeurIPS, 2018.
>
> [R4] Patterson MA, Rao AV. GPOPS-II: A MATLAB software for solving multiple-phase optimal control problems using hp-adaptive Gaussian quadrature collocation methods and sparse nonlinear programming. ACM Transactions on Mathematical Software (TOMS). 2014 Oct 27;41(1):1-37.
>
> [R5] Biegler LT, Zavala VM. Large-scale nonlinear programming using IPOPT: An integrating framework for enterprise-wide dynamic optimization. Computers & Chemical Engineering. 2009 Mar 20;33(3):575-82.
>
> [R6] Jin W, Murphey TD, Kulić D, Ezer N, Mou S. Learning from sparse demonstrations. arXiv preprint arXiv:2008.02159. 2020 Aug 5.

---

> > ### Comment · Reviewer_FvjM · 2021-08-12
> > **Thank you for clarifications!**
> >
> > Thank you for your clarifications and further experiments. I read through other reviews and authors' rebuttal.
> >
> > My concerns at the timing of the initial review have been resolved especially by the additional experimental results. Under the assumption that the authors will surely improve their papers based on the reviewers' comments, I raised my score from 4 to 6.

---

> > > ### Author Response · Authors · 2021-08-14
> > > **Thank you for raising the score!**
> > >
> > > We sincerely thank you for reading through our response and raising the score.
> > > We believe that your valuable comments have improved the paper. Thank you again for your comments.

---

### Official Review · Reviewer_u2xh · 2021-07-07

**Rating:** 6
**Confidence:** 4

**Summary:**

The authors propose a procedure for safe end-to-end learning of parameterized trajectory optimization problems. The entire learning problem is presented as a bi-level optimization. The outer optimization prescribes a loss function and safety constraints, which take as inputs the parameters and the output of an inner trajectory optimization problem using these same parameters. The overall map from parameters to the output of the outer problem is differentiated approximately with the help of: 1) approximating the inner and outer problems with unconstrained versions via log-barrier objective terms, and 2) using the discrete-time constrained PMP to differentiate the inner problem along an optimal trajectory. Safety is enforced by the log-barrier functions in the outer problem approximation.

**Limitations And Societal Impact:**

As discussed above, the authors need to add substantial discussion of the limitations of their work.




**Main Review:**

*** Originality ***

There is value in the clear manner in which the authors combine different methods with a machine learning audience in mind, and the authors' description of using log-barrier functions to enforce "safety" (even if this idea has been presented in the past). In particular, from a machine learning perspective, I enjoyed the authors' careful consideration of the quality of gradient approximations.

However, the novelty of the paper is rather limited. Specifically, the results up to the end of Section 4, including Theorem 1, are directly from:

-- Jin, Wang, Yang, and Mou, "Pontryagin Differentiable Programming: An End-to-End Learning and Control Framework", NeurIPS 2020.

The use of log-barrier functions to travel within the interior of the constraint set is the primary characteristic of Interior Point Methods (IPMs). Viewing this as a notion of "safe learning" has been done in:

-- Usmanova, Krause, and Kamgarpour, "Safe non-smooth black-box optimization with application to policy search", L4DC 2020.

-- Liu, Ding, and Liu, "IPO: Interior-Point Policy Optimization under Constraints", AAAI 2020.

Neither of these papers are cited by the authors.

I appreciate the authors' clear presentation of differentiating solutions to trajectory optimization problems. However, the authors strongly overclaim originality in this regard; e.g., on line 96 they state: "... all existing work has not studied/established the results regarding the differentiablity of general constrained optimal control systems ..." This neglects: 1) all work in sensitivity analysis for nonlinear programming and optimal control (e.g., Andersson and Rawlings, "Sensitivity Analysis for Nonlinear Programming in CasADi", IFAC 2018), 2) study of differentiable solution maps in optimal control (e.g., Trelat, "Some Properties of the Value function and its Level Sets for Affine Control Systems with Quadratic Cost", JDCS 2000), and 3) recent work on differentiable MPC (Amos, Rodriguez, et al, "Differentiable MPC for End-to-end Planning and Control", NeurIPS 2018).

*** Quality ***

I like the bi-level optimization framework the authors have clearly laid out, and I appreciate the effort put into the experiments. However, neglected details in the authors' theoretical development and flaws in the experimental evaluation mar the quality of this paper.

-- In Section 5, the unconstrained approximation of the trajectory optimization problem is not guaranteed to be safe w.r.t. the original nonlinear equality constraints (i.e., h_t and h_T). The quadratic barrier function used by the authors is insufficient in this regard. The authors do not discuss this point. On a minor related note, the authors should mention the interior of the safe set defined by the constraints must be non-empty to apply an IPM via log-barrier functions.

-- In Section 6, Theorem 3 just summarizes characteristics of IPMs, and the authors do not discuss how the approximation of the inner trajectory optimization affects the quality of the outer solution (i.e., if the inner problem is solved with some `gamma > 0`, then `theta^*(epsilon)` does not approach `theta^*` as `epsilon -> 0`). This part is particularly concerning and represents a significant disconnect in the paper.

-- Finding a feasible initial point is a key challenge when using IPMs. Given this, it is disappointing that the authors did not address the issue of finding a feasible initialization of the parameters `theta` in the main paper. In Section 7 ("Problem II: Safe Motion Planning"), this point is exacerbated by the authors' choice to experimentally compare their method with the ALTRO solver for trajectory optimization. ALTRO is not an IPM-based solver, and it can start with an infeasible trajectory guess; this kind of trade-off is common in nonlinear programming. The authors should have compared their method with an IPM-based trajectory optimization solver (e.g., DIRCOL, which uses IPOPT).

-- In Section 5, the authors state: "... solving a constrained optimal control ... is difficult since the popular trajectory optimization methods, such as iLQR and DDP, are designed only for unconstrained systems" There is an entire field of literature and academic community dedicated to trajectory optimization under constraints. The authors even use popular tools from this field, such as CasADi and ALTRO, in their experiments. I urge the authors to clarify their meaning in Section 5 and how this motivates their work.

In general, the authors do not provide any informative evaluation of the limitations of their work. As outlined above, there are a number of clear opportunities for the authors to do this, many of which are rooted in limitations of IPMs.

*** Clarity ***

The paper is well-organized and clearly written. The authors should fix any minor typos (e.g, line 104: "board", line 329: "exiting", line 291: "minimizor"), and improve the formatting and consistency of the references.

*** Significance ***

Overall, I think the idea of using log-barrier functions to enforce "safety" in an Interior Point Method (IPM)-style of optimization is a useful perspective for the machine learning community. While this idea exists in past work, I think the authors did well in communicating this idea through their current work. However, for this work to be complete, the authors need to spend more time evaluating and communicating the limitations of their method. Much of this could be connected to fundamental limitations of IPMs. Fair experimental comparisons to the state-of-the-art in trajectory optimization, optimal control, and sensitivity analysis would also help improve the quality of this paper. Finally, the authors should make an effort to discuss how the approximation of the inner trajectory optimization affects the quality of the outer solution -- an important aspect that the authors seem to have overlooked.

**Time Spent Reviewing:**

10

---

> ### Author Response · Authors · 2021-08-10
> **Our response to Reviewer u2xh's comments**
>
> We sincerely thank the reviewer for the time and effort on reviewing our paper.
>
> ----
> **Comment:** *Originality---"However, the novelty of the paper is rather limited. Specifically, the results up to the end of Section 4, including Theorem 1, are directly from: -- Jin, et al, "Pontryagin Differentiable Programming", NeurIPS 2020."*
>
> **Our response:**
> Section 4 extends Jin et al’s PDP into **constrained** systems. Importantly, we empathize the following **new theoretical results** in Section 4, which are lacking in Jin et al’s PDP. First, we have established Lemma 1 for the differentiability conditions, which are crucial for establishing Theorem 2 and Theorem 3. More importantly, we have provided a formal result in Theorem 1 and its proof to show the existence and uniqueness of the solution to the Auxiliary Control System (Equ. 3) (which is the key to PDP). For more detailed clarification, please check our response in 'our response to all reviewers' at the beginning.
>
> ---
> **Comment:** *Quality---"In Section 5, the unconstrained approximation of the trajectory optimization problem is not guaranteed to be safe w.r.t. the original nonlinear equality constraints (i.e., h_t and h_T)..... On a minor related note, the authors should mention the interior of the safe set defined by the constraints must be non-empty to apply an IPM via log-barrier functions."*
>
> **Our response:**
> We apologize for the confusion. We will clarify at the beginning of the paper that the concept of safety in this paper is only with respect to inequality constraints.
>
> As for the non-empty requirement of the safe set defined by constraints, the differentiability condition (i) in Lemma 1 has already implied it. Specifically, condition (i) in Lemma 1 states that `the second-order conditions are satisfied for the optimal control system’.  This implies that the KKT condition (or Pontryagin’s Minimum Principle) holds (see Lemma A.2 in Appendix). That the KKT condition holds also implies that the first-order constraint qualification is satisfied (see [46], pp. 19), and the constraint qualification then implies that the interior of the constraint set is non-empty. For clarification of the reader, we will add those comments under Lemma 1.
>
> ---
> **Comment:** *"In Section 6, Theorem 3 just summarizes characteristics of IPMs, and the authors do not discuss how the approximation of the inner trajectory optimization affects the quality of the outer solution.... This part is particularly concerning and represents a significant disconnect in the paper."*
>
> **Our response:**
> We thank the reviewer for raising this very important issue. We have updated Theorem 3 to make it inclusive of the approximation result of inner-level trajectory optimization in Theorem 2. Please check the updated Theorem 3  in 'our response to all reviewers ' at the beginning of this page. Please also check our proof at the anonymous link.
>
> ----
> **Comments:** *"Finding a feasible initial point is a key challenge when using IPMs.......The authors should have compared their method with an IPM-based trajectory optimization solver (e.g., DIRCOL, which uses IPOPT)."*
>
> **Our response:**
> We agree with the reviewer that the requirement of a safe initialization is a limitation of IPM-related methods, thus also a limitation of Safe PDP. While this requirement can be restrictive in some cases, we have some empirical experiences of how to provide a safe initialization for different applications in the paper experiment.  For more details, we kindly refer the reviewer to our response to all reviewers at the beginning.
>
> As commented by the reviewer, we have added an additional comparison with an IPM-based solver---CasADi (which uses IPOPT), to test their performance when solving the constrained trajectory optimization (on the 6-DoF rocket in the paper). Note that we have double-checked that DIRCOL actually uses SNOPT (i.e., SQP methods) instead of IPOPT. The comparison results are shown in the following table.
>
> ***Table 3: Comparison between Safe PDP with CasADi [R1] (which uses IPOPT, IPM-based method)***
>
> ||Number of function evaluations per iteration |	Number of gradient evaluations per iteration|	Loss value at convergence |	Constraint violation|	Compute Hessian|
> |----------- | ----------- |----------- | ----------- |----------- | ----------- |
> |CasADi (IPOPT), [R1]|	850	|160|	**26223.5**|	3.28626 e-14|	Yes|
> |Safe PDP | **120**	| **80**|26698.3|	**0**|**No**|
> ---
>
> We have the following comments on the above results.
>    - First, as shown in the third column, CasADi achieves a lower loss at the convergence than Safe PDP. This is because CasADi directly searches for the state/inputs of each step (a direct collocation method), while Safe PDP searches for a control trajectory representation $\boldsymbol{u}_t=u(t,\boldsymbol{\theta})$,  which is parameterized as a Lagrangian polynomial (see Appendix F.2). Thus, a lower-order polynomial has a limited degree of freedom to further lower the loss.
>    - Second, as shown in the first and second columns, Safe PDP requires fewer function/gradient evaluations at each iteration, this potentially shows the computational efficiency of Safe PDP. Note that we have not compared the actual running time because Safe PDP’s codes are not optimized. In the last column, we also notice that Safe PDP, when solving optimal control, does not require Hessian information. This makes Safe PDP be able to be integrated into existing neural networks.
>    - Finally, the fourth column shows that CasADi has a tolerable constraint violation. This could be because although CasADi is based on IPM, the internal process of IPOPT (see [R4]) is mainly to find the root of the perturbed KKT equations, thus allowing the violation of the constraints. In other words, the perturbed KKT equations are still well-defined even when the constraints are violated.
>
> ---
> **Comments:** *"In Section 5, the authors state: "... solving a constrained optimal control ... is difficult since the popular trajectory optimization methods, such as iLQR and DDP, are designed only for unconstrained systems" There is an entire field of literature and academic community dedicated to trajectory optimization under constraints..... I urge the authors to clarify their meaning in Section 5 and how this motivates their work."*
>
>
> **Our response:**
>  We do agree with the reviewer that there are some new tools/algorithms available for constrained trajectory optimization. However, solving a constrained optimal control problem is generally more computationally expensive than an unconstrained one. This has been shown in Fig. 4(e)---- constrained (Theorem 1) versus unconstrained (log-barrier approximation, i.e., Theorem 2). Here, we also provide more examples in the table below to show this.
>
> ***Table 4: Comparison between solving constrained optimal control and solving its unconstrained approximation***
>
> | Varying task time horizon | T=20|	T=60|	T=100|	T=140|	T=180|
> |----------- | ----------- |----------- | ----------- |----------- | ----------- |
> |Running time for solving constrained optimal control|	0.093 s|	0.542s|	1.475s|	2.243s|	3.616s|
> Running Time for solving unconstrained log-barrier approximation|	0.033s|	0.278s	|0.651s	|1.286s|	2.109s|
>
> Both Fig. 4(e) and the above table show that constrained trajectory optimization usually takes 60% longer than the unconstrained approximation. Because of this, we thus consider the computational convenience as a source of motivation to resort to the barrier-based unconstrained approximation. Additionally, the other benefits of the unconstrained formulation include avoiding the computation of the Lagrangian multipliers and attaining more numerical stability (see Fig. 4(a-d) and analysis in Appendix F.3).
>
> --------
> **An additional experiment required by the reviewer:**
> we have compared Safe PDP with CasADi [R2], a state-of-the-art sensitivity analysis software, and Differentiable MPC [R3], a differentiable framework for MPC, to show the computational performance for differentiating an optimal control system with respect to parameters. The results are shown in the following table.
>
> ***Table 5:  Comparison between Safe PDP, sensitivity analysis [R2], and differentiable MPC [R3].***
>
> |Varying task time horizon |	T=20|	T=60|	T=100|	T=140|	T=180|	T=220|	T=260|	T=300|
> |----------- | ----------- |----------- | ----------- |----------- | ----------- |----------- |----------- | ----------- |
> |Runing time for backward pass of Safe PDP |	**0.018**s	|**0.096**s	|**0.084**s|	**0.120**s	|**0.149**s|	**0.183**s|	**0.231**s	|**0.247**s|
> |Runing time for sensitivity analysis	|0.087s|	0.243s|	0.501s	|0.968s	|1.458s|	2.028s	|2.672s	|3.647s|
> |Runing time for Differentiable MPC |	0.102s	|0.257s	|0.607s	|1.208s|	2.011s|	2.837s|	3.808s|	5.169s|
> ---
>
> The above table clearly shows the computational advantage of Safe PDP over CasADi and Differentiable MPC in obtaining the gradient of optimal control systems. Specifically, Safe PDP has a complexity of $\mathcal{O}(T)$, while Differentiable MPC and CasADi have the complexity of $\mathcal{O}(T^2)$. This is because both differentiable MPC and sensitivity analysis need to compute the inverse of a large hessian matrix with the size (proportional) of $T\times T$ when applying the implicit function theorem. Instead, Safe PDP solves the gradient by constructing an Auxiliary Control System, which can be iteratively solved using the Riccati equation (see the comments below Theorem 1).
>
>
> ----
> **Comments:** *"In general, the authors do not provide any informative evaluation of the limitations of their work...."*
>
> **Our response:**
> For discussion about the limitation of Safe PDP, please check our response to all reviewers at the beginning. In our final paper, we will add a dedicated section to discuss the above limitations of Safe PDP.
>
> ----
> We will correct some typos, overclaimed languages, cite and discuss some papers suggested by the reviewer in the final paper.

---

> > ### Comment · Reviewer_u2xh · 2021-08-25
> > **Feedback on the authors' response**
> >
> > I would like to thank the authors for their thorough responses and clarifications. I raised the score to 6 as a key concern I had, namely the incompleteness of Theorem 3, has been adequately addressed in the authors' response. That said, I think this paper still has room for improvement. For example, the new results in Table 3 are not completely convincing. Also, it would be beneficial to provide a deeper investigation of the trade-off between achieving optimality and computational efficiency when approximating the optimization with barrier penalties.

---

> > > ### Author Response · Authors · 2021-08-26
> > > **Our response to Reviewer u2xh's second-round comments**
> > >
> > > We sincerely thank the reviewer for the second-round feedback and for raising the score. Below, we provide further clarifications to clear the remaining comments from the reviewer.
> > >
> > > ---
> > > __Comments:__  _" I raised the score to 6 as a key concern I had, namely the incompleteness of Theorem 3, has been adequately addressed in the authors' response. "_
> > >
> > > __Our Response:__ We again thank the reviewer for reading our response and raising the score.
> > >
> > > ---
> > > __Comments:__  _" That said, I think this paper still has room for improvement. For example, the new results in Table 3 are not completely convincing. "_
> > >
> > > __Our Response:__ Thanks for the comments. Table 3 provides a comparison between Safe PDP and CasADi (using IPOPT). Before providing further clarifications about the results in Table 3, we would like to first comment on the difference between CasADi (using IPOPT) and Safe PDP.
> > > - CasADi uses IPOPT [R2.2] to solve constrained trajectory optimization.  While IPOPT is an interior-point-based method, it does not directly add inequality constraints through barrier penalty to the cost; instead, IPOPT first converts inequality constraints into equality ones by introducing slack variables and then adds the barrier of slack variables into the cost. Thus, IPOPT still needs to explicitly handle the equality constraints during optimization (see Equ 3 in [R2.2]). This is the reason why IPOPT solves a barrier problem (see Equ. 3 in [R2.2]) using the  __primal-dual approach__---the dual variable refers to the Lagrange multipliers corresponding to equality constraints.
> > > - In contrast, Safe PDP directly adds the barrier of all constraints to the cost. Safe PDP holistically treats the cost added with the barrier as a new cost; consequently, Safe PDP only deals with unconstrained optimization and does not need to deal with any constraints during optimization anymore. Because of the resulting unconstrained problem, Safe PDP directly minimizes the barrier-added cost using __gradient descent__, instead of the primal-dual approach used in IPOPT.
> > >
> > > Based on the above differences between CasADi and Safe PDP, we now make the following comments on the results in Table 3.
> > > - The first column of Table 3 shows the number of function evaluations per iteration. For CasADi(IPOPT), the function evaluations per iteration include the evaluations of dynamics, constraints, cost function at each time step (system horizon is 40-time steps). Also note that, IPOPT needs additional function evaluations when performing line-search (see Section 2.3 in [R2.2]). For Safe PDP, at each gradient-descent, it only includes the evaluations of dynamics and barrier cost at each time step, and there is no constraint evaluation and line search process. This is the reason why Safe PDP requires far fewer function evaluations than CasADi (IPOPT).
> > > - The second column shows the number of gradient/jacobian evaluations per iteration for each method. Again, since IPOPT requires additionally to compute jacobians of equality constraints at each time step per iteration. The total number of gradient/jacobian evaluations in CasADi is larger than that in Safe PDP.
> > > - The third column shows the loss/cost value at convergence.  CasADi achieves a lower loss value at convergence than Safe PDP. This is because CasADi directly searches for the sequence of optimal states and inputs, while Safe PDP searches for a control trajectory representation $\boldsymbol{u}_t=\boldsymbol{u}(t,\boldsymbol{\theta})$,  which is parameterized as a Lagrangian polynomial (see Appendix F.2), in other words, Safe PDP searches for $\boldsymbol{\theta}$ (polynomial parameters) instead of $\boldsymbol{u}_t$. Here, we have used a 10-order polynomial, such a lower-order polynomial has a limited degree of freedom (limited expressive power) to further lower the loss.
> > > - The fourth column compares the constraint violation at convergence. It shows that CasADi (IPOPT) has a tolerable constraint violation. One explanation could be that although the internal process of IPOPT uses a fraction-to-the-boundary rule (see Equ. 15 in [R2.2]) to maintain the positivity of slack variables (i.e. the satisfaction of inequality constraints), it could possibly introduce some violation due to some numerical difficulty or computing error in certain problems.
> > > - The fifth column shows that CasADi requires Hessian during optimization. This is because IPOPT uses damped Newton’s method to solve the perturbed KKT equation (Equ. 4 in [R2.2]), which requires to compute of the Hessian of the Lagrangian of the barrier problem (see Equ. 9 in [R2.2]). In contrast, Safe PDP only requires the first-order derivative of barrier cost and dynamics. This makes Safe PDP be able to be more easily integrated as a differentiable layer into a neural network framework.
> > > - Additionally, we need to report that although the number of function/gradient evaluations per iteration in CasADi(IPOPT) is larger than that in Safe PDP, CasADi needs fewer iterations (around 300) to converge, compared to Safe PDP which requires around 400 iterations.  This is because IPOPT is based on Newton's method, which is a second-order method, while Safe PDP only uses vanilla gradient-descent, which is a first-order method. Generally, second-order methods have a faster convergence rate than first-order methods.   Also note, IPOPT uses some strategies to further accelerate convergence, such as a Line-Search Filter Method (see Section 2.3 in [R2.2]) and the Second-Order Corrections (see Section 2.4 in [R2.2]). Of course, one can also accelerate the convergence of Safe PDP by taking advantage of the advanced gradient-descent techniques such as Nesterov’s Accelerated Gradient Descent, Adagrad, Adam, etc. All these new gradient-descent techniques can be readily incorporated into Safe PDP for acceleration.
> > >
> > > ---
> > > __Comments:__  _"Also, it would be beneficial to provide a deeper investigation of the trade-off between achieving optimality and computational efficiency when approximating the optimization with barrier penalties. "_
> > >
> > > __Our Response:__ Thanks for the comments. In the paper, we have provided a theoretical guarantee, see Theorem 2 and Theorem 3 (updated in our response), and experiments, see Figure 1, Figures 2(a) and 2(c), Figures 3(a) and 3(c), to show smaller barrier parameters $\gamma$ and $\epsilon$ lead to higher accuracy of approximation results. In our response, Tables 1, 2, 4, 5 have provided the timing results of Safe PDP in obtaining approximation solutions and comparisons with other methods.
> > >
> > > To more directly address the reviewer's comments, we below provide an additional experiment to show the accuracy of approximation solution and computation time of Safe PDP (corresponding to Theorem 2 in the paper).
> > >
> > > ***Table 8: Accuracy of approximation solution $\boldsymbol{\xi}(\gamma)$ and its computation time with different choices of barrier parameters $\gamma$ in Safe PDP forward pass***
> > >
> > > | barrier parameter $\gamma$ | $1$ |	$10^{-1}$|	$10^{-2}$|$10^{-3}$|$10^{-4}$|$10^{-5}$|
> > > |----------- | ----------- |----------- | ----------- |----------- | ----------- | -----------|
> > > |accuracy percetange for approxiamtion solution, $\frac{\lVert \boldsymbol{\xi}({\gamma})-\boldsymbol{\xi}^* \rVert}{\lVert \boldsymbol{\xi}^* \rVert}$ |	51.9%|12.2%|	1.6%|	0.18%|	0.018%|  0.0002%|
> > > Timing to obtain $\boldsymbol{\xi}({\gamma})$  |	0.023s |	0.033s	|0.035s	|0.040s|	0.038s|  0.047s|
> > >
> > > Note that in the above table, $\boldsymbol{\xi}^*$ is the ground-truth solution obtained from solving a constrained trajectory optimization, and the computation time for constrained trajectory optimization is 0.062s.
> > >
> > > ---
> > >
> > > We have the following comments on the above results in Table 8:
> > >  - First, the results in the first row show that a smaller barrier parameter leads to higher accuracy of the approximation solution of Safe PDP. This again has been theoretically proved in Theorem 2 (claim (b)). The results here are consistent with the ones in Figure 1 in the paper.
> > >  - Second, the second row shows that a smaller barrier parameter, however, leads to an increase in the computational time when solving the unconstrained trajectory optimization in Safe PDP. This could be because using a small barrier parameter, the added barrier terms can increase the complex/stiff curvature of the cost/loss landscape, thus taking Safe PDP longer to find the minimizer. Despite this, the time needed for finding a minimizer is still apparently lower than directly solving a constrained trajectory optimization.
> > >  - Third, if one still wants to further increase the computation efficiency of Safe PDP, we have provided some strategies to achieve so, including "time scaling", "warm start with the previous result", and "coarse-to-fine". Please check Table 2 and our response to Reviewer Xjxt for detailed descriptions and corresponding experiment results.
> > >
> > > In summary, we have shown that a higher accuracy of approximation solution can be achieved using a smaller barrier parameter. Although a smaller barrier parameter would increase the computational time of finding the approximation solution, it is still apparently efficient than solving a constrained trajectory optimization problem. In practice, one would be likely to choose an appropriate barrier parameter according to the required accuracy. Also, there are multiple strategies available to increase the computational efficiency of Safe PDP, as detailed in our response to   Reviewer Xjxt and experimentally shown in Table 2 in our response.
> > >
> > > We will add all the above comments/results to the final version.
> > >
> > > ---
> > > __Reference__
> > >
> > > [R2.1]Andersson JA, Gillis J, Horn G, Rawlings JB, Diehl M. CasADi: a software framework for nonlinear optimization and optimal control. Mathematical Programming Computation. 2019 Mar;11(1):1-36.
> > >
> > > [R2.2] Wächter A, Biegler LT. On the implementation of an interior-point filter line-search algorithm for large-scale nonlinear programming. Mathematical programming. 2006 Mar;106(1):25-57.

---

### Official Review · Reviewer_Xjxt · 2021-07-16

**Rating:** 7
**Confidence:** 3

**Summary:**

This paper proposes a Safe Pontryagin Differentiable Programming (Safe PDP) methodology. The contributions are at both theory and algorithmics for a new safe differentiable framework for various safety-critical learning and control tasks. The proposed approach is inspired by interior-point methods. Both backward and forward pass can be approximated and computed efficient by solving an unconstrained counterpart. The approximation can also be controlled for arbitrary accuracy using a barrier parameter. In addition, the authors can also show that all constraints are respected throughout the course of approximation and optimization, thus guaranteeing safety. Experiment results are shown in many learning and control tasks on different challenging control
systems such as 6-DoF maneuvering quadrotor and 6-DoF rocket powered landing.


**Limitations And Societal Impact:**

Yes

**Main Review:**


I found this paper studies a very interesting and important research problem in control learning. The proposed approach will be very potential for many robot learning problems. The experiments on many robotic tasks are very convincing and non-trivial.

Could the authors comment on how to improve the solving time of the forward pass? Each iteration, e.g. T=100, might take approximately 1s. If this can be improved, I believe Safe PDP could become more useful, e.g. integrated into an end-to-end task network. It might be interesting if the implementation can be leveraged to exploit GPUs, similar to the way OptNet and cvxpylayers do.

If possible for future work, it might be very interesting to see Safe PDP can be used as a layer in an end-to-end network.

Reference [1] might be related.


[1] Priya L. Donti, Melrose Roderick, Mahyar Fazlyab, J. Zico Kolter:
Enforcing robust control guarantees within neural network policies. ICLR 2021


**Time Spent Reviewing:**

6

---

> ### Author Response · Authors · 2021-08-10
> **Our response to Reviewer Xjxt's comments**
>
> We sincerely thank the reviewer for the time and effort on reviewing our paper.
>
> ---
> **Comments:** *"Could the authors comment on how to improve the solving time of the forward pass? Each iteration, e.g. T=100, might take approximately 1s. If this can be improved, I believe Safe PDP could become more useful, e.g. integrated into an end-to-end task network. It might be interesting if the implementation can be leveraged to exploit GPUs, similar to the way OptNet and cvxpy layers do."*
>
> **Our response:**
> The forward pass of Safe PDP is to solve an optimal control problem (trajectory optimization). In the paper experiments, we used an NLP solver (based on IPOPT [R5]) to solve the trajectory optimization. Since the solver blindly treats an optimal control problem as general non-linear programming without leveraging the (sparse) structures that exist in optimal control systems. Thus, it is not very efficient and ideal, especially for long-horizon control tasks. To accelerate trajectory optimization, there are plenty of strategies, including
> - To solve longer-time horizon optimal control problems, one most effective method is to scale the (continuous) time horizon into a smaller one (like a unit) by applying a given time-warping function to both the system dynamics and cost function. After discretizing and solving this scaled short-horizon optimal control problem, then re-scale the obtained optimal trajectory back. This is a common strategy used in many commercial optimal control solvers, such as GPOPS by Patterson et al. (2014) [R4].
> - There is also a `warm-up’ trick to accelerate the trajectory optimization in the forward pass. Specifically, one can initialize the trajectory at the next iteration using the result of the previous iteration.
> - One also can use a hierarchical strategy to perform trajectory optimization from coarse to fine resolutions. For example, given a long-time horizon optimal control system, first, discretize the trajectory with larger granularity and solve for a coarse-resolution trajectory; then use the coarse trajectory as initial conditions to solve the trajectory optimization with fine granular discretization.
>
> As an additional experiment, we tested and compared the above three strategies for solving the forward pass of Safe PDP.  The running performance of each strategy is given in the following table. Here, $t_f$ is the continuous-time horizon,  $\Delta$ is the discretization interval, and the discrete-time horizon is   $T=f_f/\Delta$.
>
> ***Table 2: Running time for different strategies to accelerate the forward pass of Safe PDP.***
>
> | |$t_f=2$s, $\Delta=0.1$s, $T=f_f/\Delta$=20|	$t_f=6$s, $\Delta=0.1$s, $T=f_f/\Delta$=60 | $t_f=10$s, $\Delta=0.1$s, $T=f_f/\Delta$=100	| $t_f=20$s, $\Delta=0.1$s, $T=f_f/\Delta$=200	|
> |----------- |----------- | ----------- |----------- | ----------- |
> |Plain NPL solver|	0.082s|	0.202s|	0.491s|	1.743s|
> **Time scaling**| 	**0.014**s|	**0.033**s	|**0.055**s|	**0.083**s|
> Warm start with previous result	|0.055s	|0.095s	|0.108s	|0.224s|
> Coarse-to-fine|	0.021s	|0.055s	|0.074s|	0.133s|
> ---
>
>
> From the above table, one can see that time-scaling is the most effective way to accelerate the trajectory optimization for longer time horizon tasks. Of course, one can combine some of the above strategies to further improve the running performance of the forward pass.
>
>  Additionally, one can also use iLQR and DDP to solve optimal control problems. iLQR can be viewed as the one-and-half– order method---linearizing dynamics and quadratizing cost function. DDP is a second-order method ---quadractizing both dynamics and cost function. Both methods solve a local bellman equation to generate the update of the control sequence. But without coding optimization, both methods are slower than the commercial NPL solver (e.g., CasADi [R1]). There are some ongoing works trying to take advantage of GPU for accelerating trajectory optimization, which is also our future research.
>
> We will put the above discussions into our final version paper.
>
>
> ---
> **Comments:** *"If possible for future work, it might be very interesting to see Safe PDP can be used as a layer in an end-to-end network."*
>
> **Our response:**
> Currently, we have provided a standalone python package for Safe PDP (GitHub link will be shown in the final version), which has a computerizable interface to allow a user to connect to a differentiable physical environment for learning and control. Our future work focuses on integrating Safe PDP into regular neural network frameworks; at the same time, we are also working on improving the optimal control solver using GPUs.
>
>
>  ---
>  **Comment:** *"Reference [1] might be related.
> [1] Priya L. Donti, Melrose Roderick, Mahyar Fazlyab, J. Zico Kolter: Enforcing robust control guarantees within neural network policies. ICLR 2021"*
>
> **Our response:** Thank the reviewer for pointing out the reference. In Donti et al’s work, safety is defined as robustness ---- the controller needs to ensure system stability in the case of the worst adversarial disturbance. Here, the set of robust policies is given by classic robust control (solving LMI). Then, an RL neural policy with a differentiable projection layer is learned in an end-to-end manner, such that the action outputted by the RL policy can lie in the robust policy set. Compared to this method, safety in Safe PDP is defined as constraint satisfaction. Safe PDP does not use projection to enforce constraints, instead, it handles the constraints by adding them into the cost/objective function through barrier functions. Thus, Safe PDP does not explicitly handle the hard constraints but guaranteeing their satisfaction through the barrier function. In the revised paper, we will cite the above work and give some discussions.
>
>
> ----
>
> Reference:
>
> [R1] Andersson, J. A., Gillis, J., Horn, G., Rawlings, J. B., & Diehl, M. (2019). CasADi: a software framework for nonlinear optimization and optimal control. Mathematical Programming Computation, 11(1), 1-36.
>
> [R4] Patterson MA, Rao AV. GPOPS-II: A MATLAB software for solving multiple-phase optimal control problems using hp-adaptive Gaussian quadrature collocation methods and sparse nonlinear programming. ACM Transactions on Mathematical Software (TOMS). 2014 Oct 27;41(1):1-37.
>
> [R5] Biegler LT, Zavala VM. Large-scale nonlinear programming using IPOPT: An integrating framework for enterprise-wide dynamic optimization. Computers & Chemical Engineering. 2009 Mar 20;33(3):575-82.

---

> > ### Comment · Reviewer_Xjxt · 2021-08-26
> > **Response**
> >
> > Thanks the authors for a great elaborated response. I found it very insightful and convincing.

---

> > > ### Author Response · Authors · 2021-08-26
> > > **Thanks for your feedback!**
> > >
> > > We sincerely thank the reviewer for reading through our response.  We believe that your valuable comments have improved the paper. Thank you again for your comments.

---

### Official Review · Reviewer_v3kP · 2021-07-18

**Rating:** 9
**Confidence:** 4

**Summary:**

This paper provides an efficient method based on Pontryagin Differentiable Programming (PDP) for the control of safety-critical systems. The method entails formulating the control problem as a bi-level optimization problem, where
* The lower-level problem solves for a trajectory that minimizes some control cost subject to dynamics, terminal constraints, and path constraints associated with the underlying system (all of which are potentially parameterized), and
* The upper-level problem aims to pick the parameters of the lower-level problem to minimize some task-specific loss subject to task-specific constraints.

The authors provide an efficient method to solve problems of this form (e.g., both terminal and path constraints) by employing barrier methods to both rewrite the outer problem as an unconstrained optimization problem, and to efficiently solve for solutions of and implicit gradients through the inner problem. They then prove that under certain assumptions, their method satisfies all problem constraints, as needed for the safety-critical setting. They provide experimental validation for the problems of policy optimization, motion planning, and imitation learning in several dynamical systems settings.

**Limitations And Societal Impact:**

While the authors do address the *assumptions* of their method, they do not really discuss its limitations -- notably, the potential computational cost with respect to other methods, in which settings the stated assumptions might break, and in which settings other methods  (even, e.g., traditional robust control) may be better-suited. It could be good to incorporate some additional discussion of these points in the related work and experiments (in the latter case, particularly by introducing some baselines).

**Main Review:**

This paper provides a novel and interesting extension of PDP for safety-critical settings. The approach is well-motivated, the theory is well-grounded with reasonable assumptions, and the authors do a good job explaining the intuition and implications of the theoretical results even without including the full proofs in the main paper. The empirical evaluation does a good job showing the efficacy and generality of the proposed method. Overall, I think this is a very solid contribution and is well-suited for NeurIPS.

One suggestion for improvement is that non-safe PDP methods should be compared against in the experiments, in order to enable a better discussion of the salient tradeoffs (e.g., does constraint enforcement come at the cost of computational expense?).

Another suggestion is that the nascent literature on safe control via neural networks + differentiable optimization layers should also be discussed and ideally compared against in the experiments. (These methods do attempt to address similar settings as in the present work, and therefore can potentially provide some "safe" baselines.) The idea of this literature is that differentiable optimization layers can be constructed to capture the different types of constraints demanded by the setting (e.g., both immediate and long-term constraints, as addressed in the present work), and then used to constrain the output of a neural network. Some examples here include:
* Pham, Tu-Hoa, Giovanni De Magistris, and Ryuki Tachibana. "OptLayer - Practical constrained optimization for deep reinforcement learning in the real world." 2018 IEEE International Conference on Robotics and Automation (ICRA). IEEE, 2018.
* Donti, P. L., Roderick, M., Fazlyab, M., & Kolter, J. Z. (2021). Enforcing robust control guarantees within neural network policies. ICLR 2021.
* Chen, B., Donti, P., Baker, K., Kolter, J. Z., & Berges, M. (2021). Enforcing Policy Feasibility Constraints through Differentiable Projection for Energy Optimization. ACM e-Energy 2021.

As an additional question: In the experiments for Problem I, are the controllers evaluated on the true $f$, or the learned $f$? This should be made clearer, as evaluating on the learned $f$ could lead to potential over-stating of the results.

Minor points, questions, and typo corrections (not affecting my score):
* Line 113: "arbitrarily" --> "arbitrary"
* Equation 1: Should the dynamics and path constraints have some notion of "$\forall t$" somewhere? (Similar comment for later equations.)
* Line 160: Since challenge 1 already seems to address how to obtain $\partial \xi_\theta / \partial \theta$ efficiently (with $\mathcal{O}(T)$ complexity), I was initially confused by the implicit claim in challenge 2 that  $\partial \xi_\theta / \partial \theta$ is potentially expensive to obtain (and therefore needs to be cheaply obtained via an approximation). Perhaps the wording and presentation around the $\mathcal{O}(T)$ bound could be modified in the first point in order to smooth the narrative.
* Lines 160--169: These points should generally be checked for grammar and typos.
* Line 239: "motive" --> "motivate"
* Line 248, Theorem 2: "differentiability" is misspelled
* Line 291, Theorem 3: "minimizer" is misspelled
* Line 329: "exiting" --> "existing"

**Time Spent Reviewing:**

3.5

---

> ### Author Response · Authors · 2021-08-10
> **Our response to Reviewer v3kP's comments**
>
> We sincerely thank the reviewer for the time and effort on reviewing our paper.
>
> ---
> **Comments:** *"One suggestion for improvement is that non-safe PDP methods should be compared against in the experiments, in order to enable a better discussion of the salient tradeoffs (e.g., does constraint enforcement come at the cost of computational expense?)."*
>
>
> **Our response:**
> As additional experiments in this response, we compare Safe PDP and PDP [61] (see the reference in the paper) in solving Problem III (i.e., learning MPCs from demos, here experiment settings are the same with the paper and appendix). The comparison results are shown in the following table.
>
>
> ***Table 1: Performance comparison between Safe PDP and PDP [61].***
>
> |  | Loss at convergence | Time for forward pass|Time for backward pass|Can learn constraints?| Constraint guarantee?|
> | ----------- | ----------- |----------- | ----------- |----------- | ----------- |
> |PDP [61] |	524.02|	**0.10s**|	0.046s|	No|	No|
> |Safe PDP|	**7.42**|	0.21s|	0.042s|	**Yes**	|**Yes**|
> ------
>
> Based on the above results, we have the following comments and analysis:
>   - We note that Safe PDP achieves lower training loss. This is because compared to PDP, Safe PDP has introduced the inductive bias of constraints within its model architecture, making it more suited to learn from demonstrations which are the results of constrained control systems. In this sense, Safe PDP architecture (with constraints bias) can be thought of having more expressive power than PDP architecture in the above experiments.
>   - For Safe PDP, its ability to learn and guarantee constraints comes at the cost of higher computational expense in forward pass (see the second column in the above table). That is, each iteration in Safe PDP requires solving constrained trajectory optimization. Even though Safe PDP handles constraint enforcement by adding them to cost using log barrier functions, solving the resulting unconstrained approximation still needs more time than solving the unconstrained PDP. This has been shown in the second column of the above table. This could be because the added log barrier terms can increase the complex/stiff curvature of the cost/loss landscape, thus taking longer to find the minimizer.
>   - The running time of backward is almost the same for both PDP and Safe PDP, because both methods are solving an unconstrained LQR problem of the same size (see Theorem 2), which can be very efficient based on Riccati Equation (iteration).
>
> In the final paper, we will add the above discussion as an additional subsection. Please refer to other limitations of Safe PDP in our response to all reviewers at the beginning.
>
> ----
>
> **Comments:** *"Another suggestion is that the nascent literature on safe control via neural networks + differentiable optimization layers should also be discussed and ideally compared against in the experiments...."*
>
>
> **Our response:** we thank the reviewer for suggesting some concurrent works. Since some of them are published very recently and we were not aware of them at the time of the paper submission. We will cite those and give a comparison/analysis in our final paper. We below provide a brief comment on those works. Both Pham et al. (2018) and Chen et al. (2021) ensure safety by constructing a dedicated `projection layer’ --- which projects the actions from a neural policy into the safe region (satisfying the safe constraints). This projection layer is a differentiable convex layer, which can be trained end-to-end.  In work by Donti et al. (2021), safety is defined as robustness ---- controller needs to ensure stability in the case of the worst adversarial disturbance. Here, the set of robust policies is given by classic robust control (solving LMI). Then, an RL neural policy with a differentiable projection layer is learned in an end-to-end manner, such that the action outputted by the RL policy can lie in the robust policy set.
>
> ---------
> **Comments:** *"As an additional question: In the experiments for Problem I, are the controllers evaluated on the true f, or the learned f? This should be made clearer, as evaluating on the learned f could lead to potential over-stating of the results."*
>
> **Our response:** In Problem I, the controller is evaluated on true dynamics. We will make it clear in the final paper.
>
> ----
> **Comment:** *"While the authors do address the assumptions of their method, they do not really discuss its limitations..."*
>
> **Our response:** For discussion about the limitation of Safe PDP, we kindly refer the reviewer to our response to all reviewers at the beginning. In our final paper, we will add a dedicated section to discuss the above limitations of Safe PDP in detail.
>
> ---
> **Comment:** *"Minor points, questions, and typo corrections (not affecting my score)....."*
>
> **Our response:** as for the reviewer's question "Equation 1: Should the dynamics and path constraints have some notion of "∀t" somewhere? (Similar comment for later equations.)", our response is --- yes, dynamics and path constraints are for any time steps. Note that since we allow both the dynamics and constraints to be time-dependent (explicitly depending on $t$), it is flexible to have different constraints/dynamics at different time steps. As for the typos and language in the paper, we will address them in the final version. We thank the reviewer for pointing them out.
>
> ---
>
> We will correct some typos, languages, cite and discuss some papers suggested by the reviewer in the final paper.

---

> > ### Comment · Reviewer_v3kP · 2021-08-24
> > **Re: Our response to Reviewer v3kP's comments**
> >
> > Thanks to the authors for the detailed response, and for the running of additional experiments! Based on the experimental results in (both those presented here and in responses to the other reviewers), I have further raised my score from 8 to 9.

---

> > > ### Author Response · Authors · 2021-08-25
> > > **Thank you for raising the score!**
> > >
> > > We sincerely thank you for reading through our response and raising the score. We believe that your valuable comments have greatly improved the paper. Thank you again for your time and effort in providing the comments.

---

### Author Response · Authors · 2021-08-10
**Our response to all Chairs and Reviewers about the major improvement to the paper**

We sincerely thank all reviewers for their constructive comments. Following all reviewers’ comments, point by point, we have improved the paper. The major improvement includes:

1. **First, as per Reviewers v3kP and u2xh, we add one dedicated section to discuss the limitations of the proposed method**. Specifically, the limitations of Safe PDP are summited as below.

    - Safe PDP requires a safe (feasible) initialization such that the log-barrier based optimization is well-defined. While this requirement can be restrictive in some cases, we have the following empirical experiences on how to provide safe initializations for different types of problems. Specifically,
        - In safe policy optimization (Problem I), one could first use supervised learning to learn a safe policy from certain safe trajectories/demonstrations (which could not necessarily be optimal). Then, use the learned safe policy to initialize the Safe PDP algorithm. We have used this strategy in the experiments.
        - In safe motion planning (Problem II), one could arbitrarily provide a safe trajectory (not necessarily optimal) to initialize the Safe PDP algorithm. We have used this strategy in the experiments.
        - In Problem III (learning MPCs from demonstrations), the goal includes learning of constraint models, and there is no such requirement.

    - Safe PDP cannot address the problem of robust control/learning. The goal of robust control/learning concerns how to maintain or achieve good performance (such as stability or optimality) in the worst case of disturbance or attacks to a system. Methods that handle robustness have been well-developed in robust control theory and differential game in the control/learning community. Safe PDP, on the other end, only focuses on the guarantee of inequality constraints throughout learning/control, and such constraints are defined on the system states and inputs.

2. **Second, as per Reviewer u2xh, we have improved Theorem 3 to make it inclusive of the approximation result of inner-level trajectory optimization in Theorem 2. The updated Theorem 3 is stated below, and its proof is given at this anonymous link: (https://ufile.io/x08iffla)**

   ----------------------------------------
   **Theorem 3.**  *Consider all functions  defining the constrained optimal control system $\boldsymbol{\Sigma}(\boldsymbol{\theta})$ are three-times continuously differentiable,  and let the conditions (i)-(iii) in Lemma 1 for differentiblity of* $\boldsymbol{\xi}_{\boldsymbol{\theta}}$  *hold in a neighborhood of* $\boldsymbol{\theta}^*$.  *Suppose that the second-order conditions  for a  local isolated  minimizer*  $\boldsymbol{\theta}^*$ *to Problem P are satisfied*,  *that the gradients of all binding constraints*  $R_i(\boldsymbol{\xi}_\boldsymbol{\theta^*},\boldsymbol{\theta}^*)=0$, $\nabla_\boldsymbol{\theta} R_i(\boldsymbol{\xi}_\boldsymbol{\theta^*},\boldsymbol{\theta}^*)$, *are linearly* independent at $\boldsymbol{\theta}^*$, *and that  the strict complementary holds at* $\boldsymbol{\theta}^*$. *Then, for any small* $\epsilon>0$ *and any small* $\gamma>0$, *the optimization*

    $$ \qquad\qquad \qquad  \qquad  \boldsymbol{\theta}^*{(\epsilon, \gamma)}=\arg\min_\boldsymbol{\theta}\ell\big(\boldsymbol{\xi}_{(\boldsymbol\theta,\gamma)},\boldsymbol{\theta}\big) - \epsilon
\sum\_{i=1}^l\ln\Big(-R_i\big( \boldsymbol{\xi}\_{(\boldsymbol\theta,\gamma)}, \boldsymbol{\theta} \big) \Big),   \tag*{SP(${\epsilon},\gamma$)}
   $$
   *with $\boldsymbol{\xi}\_{(\boldsymbol{\theta},\gamma)}$ being the trajectory solution to inner-level constrained optimal control system $\boldsymbol{\Sigma}(\boldsymbol{\theta},\gamma)$,* *has the following assertions:*

   - *(a) there  exists a local isolated minimizer* $\boldsymbol{\theta}^*(\epsilon,\gamma)$ *to the above optimization* $SP({\epsilon},\gamma)$,  *and the corresponding system trajectory* $\boldsymbol{\xi}_{(\boldsymbol{\theta}^*(\epsilon,\gamma),\gamma)}$  *from the inner-level system*  $\boldsymbol\Sigma{\big(\boldsymbol{\theta}^*(\epsilon,\gamma),\gamma\big)}$ *is  safe,  i.e.,* $R_i\big(\boldsymbol{\xi}\_{(\boldsymbol{\theta}^*(\epsilon,\gamma),\gamma)}, \boldsymbol{\theta}^*(\epsilon,\gamma)\big)<0$, $i=1,2,...,l$;
   -(b) $\boldsymbol{\theta}^*(\epsilon,\gamma)$ *is  once-continuously differentiable with respect to both $\epsilon$ and $\gamma$, and*
$$ \qquad\qquad \qquad  \qquad
	\boldsymbol{\theta}^*(\epsilon,\gamma)\rightarrow \boldsymbol{\theta}^* \qquad \text{as} \qquad \epsilon\rightarrow 0 \quad \text{and} \quad  \gamma\rightarrow0;
$$
   - (c) *for any* $\boldsymbol{\theta}$ near  $\boldsymbol{\theta}^*(\epsilon,\gamma)$,  *the trajectory*  $\boldsymbol{\xi}\_{(\boldsymbol{\theta},\gamma)}$ *from the inner-level system* $\boldsymbol\Sigma{(\boldsymbol{\theta},\gamma)}$ *is safe, i.e.,*  $R_i(\boldsymbol{\xi}\_{(\boldsymbol{\theta},\gamma)},\boldsymbol{\theta})< 0$,  $i=1,2,...,l$.
   ----------------------------------------

   We emphasize the importance of the above theorem in the following aspects.
	  - Claim (a) affirms that even though the inner-level trajectory $\boldsymbol{\xi}\_{(\boldsymbol{\theta},\gamma)}$  is approximate (recall that the accuracy of $\boldsymbol{\xi}\_{(\boldsymbol{\theta},\gamma)}$ is controlled by the inner-level barrier parameter $\gamma$, as asserted in Theorem 2), the outer-level optimization in $SP({\epsilon},\gamma)$ always has a locally unique solution  $\boldsymbol{\theta}^*(\epsilon,\gamma)$.

	  - Claim (b) asserts that the accuracy of the outer-level solution  $\boldsymbol{\theta}^*(\epsilon,\gamma)$  is controlled by both the inner-level barrier parameter $\gamma$ and outer-level barrier parameter $\epsilon$; and as both such parameters approach zero, the outer-level solution  $\boldsymbol{\theta}^*(\epsilon,\gamma)$   is converging to the true solution  $\boldsymbol{\theta}^*$  to the original Problem P.

	  - Claim (c) says the local search of the outer-level approximate solution $\boldsymbol{\theta}^*(\epsilon,\gamma)$ with inner-level approximate trajectory $\boldsymbol{\xi}\_{(\boldsymbol{\theta},\gamma)}$  always guarantees the safety of inequality constraints.

   The updated Theorem 3 summarizes all the technical results about Safe PDP in the previous sections (Section 4 and Section 5), which is one of the main theoretical contributions of this Safe PDP paper.

3. **Third, we have added the new experiments to support the theoretical contributions of Safe PDP**. Specifically,
    - In response to Reviewer v3kP (see Table 1), we have compared Safe PDP against non-safe PDP to show the tradeoff of Safe PDP due to constraint enforcement.
    - In response to Reviewer Xjxt (see Table 2), we have added additional experiments to show different strategies for accelerating the forward pass of Safe PDP.
    - In response to Reviewer u2xh (see Table 3), we have added the comparison between Safe PDP and CasADi [R1], a state-of-the-art interior-point based solver, to compare the performance of computational efficiency and constraint violation when solving safe motion planning.
    - In response to Reviewer u2xh (see Table 5), we have compared Safe PDP against a state-of-the-art sensitivity analysis algorithm [R2], and also against Differentiable MPC [R3], to show the performance for differentiating an optimal control system with respect to parameters. Also, see Table 4 for the running performance comparison between directly solving a constrained optimal control versus solving its unconstrained log-barrier approximation.
    - In response to Reviewer FvjM (see Table 6 and Table 7), we have added additional experiments to show that Safe PDP can handle non-optimal demonstrations and has small sample complexity.

4. **Finally, as per Reviewer u2xh, we want to remind the reviewers of our theoretical contributions in Section 4**. Section 4 is an extension of Jin et al’s PDP [61] into **constrained** optimal control systems, but we empathize the following **new theoretical results**, which are lacking in Jin et al’s PDP:
   - We have established Lemma 1 for the differentiability conditions of an optimal control system. Such conditions are crucial for establishing the results in Theorem 2 and Theorem 3. Specifically, it warrants (a) the existence uniqueness solution of the Auxiliary Control System, a key concept of PDP for efficient differentiation, in Theorem 2, and (b) the controlled accuracy of the log-barrier approximation in both inner- and outer- levels, claimed in Theorem 3 (updated above).
   - Importantly, we have formally given Theorem 1 and its proof to show the existence and uniqueness of the solution to the Auxiliary Control System (LQR). In Jin et al’s paper, the solution to Auxiliary Control System is only given empirically, and its existence and uniqueness have not been proved. Theorem 1 now formally states that if the original optimal control system (Equ. 1) satisfies the differentiability conditions in Lemma 1, the Auxiliary Control System always has a unique and global solution, which is exactly the gradient of the trajectory. Such assertion and its proof are not trivial.

References (will be used in the following response):

[R1] Andersson, J. A., Gillis, J., Horn, G., Rawlings, J. B., & Diehl, M. (2019). CasADi: a software framework for nonlinear optimization and optimal control. Mathematical Programming Computation, 11(1), 1-36.

[R2] Andersson, J. A., & Rawlings, J. B. (2018). Sensitivity analysis for nonlinear programming in CasADi. IFAC-PapersOnLine, 51(20), 331-336.

[R3] Amos, B., Rodriguez, I. D. J., Sacks, J., Boots, B., & Kolter, J. Z., Differentiable MPC for End-to-end Planning and Control. In NeurIPS, 2018.

[R4] Patterson MA, Rao AV. GPOPS-II: A MATLAB software for solving multiple-phase optimal control problems using hp-adaptive Gaussian quadrature collocation methods and sparse nonlinear programming. ACM Transactions on Mathematical Software (TOMS). 2014 Oct 27;41(1):1-37.

---

### Decision · Program_Chairs · 2021-09-27

**Decision:**

Accept (Poster)

**Comment:**

The authors response has address concerns from the reviewers, and all reviewers agree on an acceptance. I encourage the authors to revise the draft by including the discussion and the additional experiments from the rebuttal.